# Microenvironment-responsive immunoregulatory electrospun fibers for promoting nerve function recovery

Kun Xi [1,5], Yong Gu [1,5], Jincheng Tang [1,5], Hao Chen[2], Yun Xu[3], Liang Wu[1], Feng Cai[1], Lianfu Deng[2], Huilin Yang[1], Qin Shi [1,4✉], Wenguo Cui [1,2✉] & Liang Chen [1✉]

The strategies concerning modification of the complex immune pathological inflammatory environment during acute spinal cord injury remain oversimplified and superficial. Inspired by the acidic microenvironment at acute injury sites, a functional pH-responsive immunoregulation-assisted neural regeneration strategy was constructed. With the capability of directly responding to the acidic microenvironment at focal areas followed by triggered release of the IL-4 plasmid-loaded liposomes within a few hours to suppress the release of inflammatory cytokines and promote neural differentiation of mesenchymal stem cells in vitro, the microenvironment-responsive immunoregulatory electrospun fibers were implanted into acute spinal cord injury rats. Together with sustained release of nerve growth factor (NGF) achieved by microsol core-shell structure, the immunological fiber scaffolds were revealed to bring significantly shifted immune cells subtype to down-regulate the acute inflammation response, reduce scar tissue formation, promote angiogenesis as well as neural differentiation at the injury site, and enhance functional recovery in vivo. Overall, this strategy provided a delivery system through microenvironment-responsive immunological regulation effect so as to break through the current dilemma from the contradiction between immune response and nerve regeneration, providing an alternative for the treatment of acute spinal cord injury.

---

[1] Department of Orthopedics, the First Affiliated Hospital of Soochow University, Orthopedic Institute, Soochow University, 188 Shizi Road, 215006 Suzhou, Jiangsu, P. R. China. [2] Shanghai Key Laboratory for Prevention and Treatment of Bone and Joint Diseases, Shanghai Institute of Traumatology and Orthopaedics, Ruijin Hospital, Shanghai Jiao Tong University School of Medicine, 197 Ruijin 2nd Road, 200025 Shanghai, P. R. China. [3] Shanghai Public Health Clinical Center, Fudan University, 2901 Caolang Road, 200025 Shanghai, P. R. China. [4] Key Laboratory of Stem Cells and Biomedical Materials of Jiangsu Province and Chinese Ministry of Science and Technology, 199 Renai Road, 215123 Suzhou, P. R. China. [5] These authors contributed equally: Kun Xi, Yong Gu, Jincheng Tang. ✉email: shiqin@suda.edu.cn; wgcui80@hotmail.com; chenliang1972@sina.com

 1

Owing to the limited regenerative capability of spinal cord, the sensory and motor function of the patient could be permanently compromised after spinal cord injury (SCI)[1]. After acute SCI, an ischemic contributed to the decrease in tissue pH immediately, forming an acidic environment[2]. Meanwhile, infiltrating macrophages and in situ activated microglia reached their respective peaks at focal area on 3 and 7 days after injury and polarized into "classically activated macrophage (M1)" due to the effect of tumor necrosis factor (TNF)-α and other pro-inflammatory factors responsible for the secretion of more pro-inflammatory factors, which further led to the aggravation of the injury and formation of scar tissue[3,4]. On the other hand, when macrophages and microglia bind to specific factors, they can be polarized into M2 subtype, also known as "alternatively activated macrophage," which secretes anti-inflammatory factors and reduces the inflammatory responses[5]. In the stage of acute SCI, the balance between M1 and M2 subtypes could be one of the key factors in determining the prognosis of SCI[6]. Neural tissue engineering, as a promising alternative therapeutic strategy for biomimetic regulation and promoting neural regeneration, has attracted considerable attention[7]. In the study of conventional tissue engineering using biomaterials to transmit biological information to promote neural differentiation of stem cells, nerve repair is at risk of failure under the influence of severe immune inflammation exerted by acute SCI[8]. Therefore, the construction of the bioinspired material that can rapidly respond to local microenvironment during the acute stage and accurately regulate macrophage and microglia cell polarization to reduce inflammatory response and provide sustained neurogenic platform for stem cells in the later stage may break through the bottleneck of the current treatment of acute SCI.

As one of the pivotal elements concerning tissue engineering, scaffolds made of biomaterials has been employed as the vehicle for stem cells or bioactive factor delivery as well as the core template for cellular activity and tissue regeneration[9]. The oriented electrospun fibers with biomimetic structure have been extensively reported in the repairing of specific tissues due to its capability in contacting guidance tissue regeneration and loading bioactive drugs[10]. However, before qualified as a suitable drug loading vehicle, electrospinning technique still had some problems to solve such uncontrolled burst release, short sustained release period, and low focal concentration[11]. Previously, the microsol electrospinning technology developed by our group has exhibited significant advantages over emulsion electrospinning and coaxial electrospinning due to its simple equipment, stable process, high drug loading rate >80%, and prolonged controlled release for >6 weeks, which opens up a way for the sustained release of water-soluble drugs and protein molecules[12]. More importantly, the core–shell structure could shield the vulnerable biological factors against unpleasant microenvironment, quantifying it as a suitable structure for controlled biofactor release in acidic environment after SCI[13]. In addition, cationic liposomes have provided another option for the drug loading in electrospun scaffold[14]. More importantly, they could potentially endow the electrospun fibers with the capability of non-viral gene transduction due to their membrane-like structure and promoted lipid exchange, adsorption, and endocytosis with cell membrane[15]. Loading the cationic liposome with non-viral gene transfection vectors was reported to be an effective while safe approach due to its reduced toxicity and non-involvement of host genome[16]. Studies have shown that combining electrospinning with cationic liposomes would be more effective in antibacterial, inducing bone tissue regeneration and promoting vascular repair[17–19]. However, few studies were reported to employ similar strategy to simultaneously control the inflammatory response after acute SCI and promote nerve regeneration continuously.

In addition to biomaterials, biological factors comprise another key element in tissue engineering to endow the biomaterial with the functionality of immunoregulation under spinal cord micro-environment and neurogenic activity for the guidance of stem cell differentiation. Interleukin-4 (IL-4), a member of the chemokine family, is a cytokine secreted by white blood cells and plays a regulatory role between white blood cells[20]. It can bind to macrophage surface receptors to phosphorylate signal transducer and activator of transcription factor 6, induce T helper type 2 immune response through the phosphoinositide-3 kinase/AKT signaling pathway, and promote tissue repair and reduce inflammatory immune response[21]. It has also been reported that IL-4 can play a protective role in the central nervous system and promote the migration of endogenous stem cells to the injured site to create a more favorable microenvironment for nerve repair[22]. Moreover, IL-4 can indirectly promote Schwann cell migration and synthesis and release of nerve growth factor (NGF) in the nervous system[23]. NGF has the dual biological effect of nourishing neurons and promoting axon growth and plays an important role in promoting the differentiation of endogenous neural progenitor cells into neuron-like cells[24]. Under pathological conditions caused by SCI, the depletion of local NGF leads to the damage of protecting neurons and promoting nerve regeneration, so it is imperative to supply NGF continuously in the repair of SCI[25]. In spite of multiple candidate biological factors in the treatment of SCI, integrating them with biomaterials in a manner suited for the biological and pathological feature of SCI would be the key in successful recovery of compromised functionality.

In this work, inspired by the inflammation and acid-enriched feature of the microenvironment after acute SCI, a biomimetic fiber scaffold with both immunoregulation function and neurogenic potential is designed to satisfy the specific demand during the acute-stage suppression of inflammation and later-stage neural regeneration in the scenario of SCI. The aldehyde-modified cationic liposomes loading IL-4 plasmid (pDNA) are grafted onto the surface of amino-modified oriented microsol electrospun fiber scaffolds through Schiff base bond that could break under acidic environment, leading to the release of pDNA-loaded liposome as well as its transfection effect that induced the M2 polarization and anti-inflammatory factor secretion of macrophages and microglia, so as to achieve the triggered and rapid immunoregulation effect on the suppression of inflammation during the acute stage and pave the way for later-stage neural regeneration (Fig. 1). The construction of oriented fiber scaffold as well as the loading of pDNA-liposome and NGF are fully investigated through physiochemical methodologies, followed by the evaluation of the fiber scaffolds' in vitro effect on immunological regulation and neurogenic stimulation using macrophages and bone marrow mesenchymal stem cells (BMSCs). A spinal cord hemi-section model on Sprague Dawley (SD) rat is employed to investigate the in vivo performance on inflammation suppression and nerve regeneration together with the promoted functional recovery.

## Results and Discussion

**Screening of aldehyde cationic liposomes (aLs).** The bioinspired microenvironment-responsive fiber has been constructed step by step as shown above (Fig. 1). In order to verify the effect of materials, physiochemical property and in vitro and in vivo biological characteristics of different combinations were characterized comprehensively in this study with the group setting depicted in Supplementary Table 1.

In order to provide the non-viral transfection vectors for regulation of immunological microenvironment, liposomes were prepared and further modified with aldehyde phospholipids and

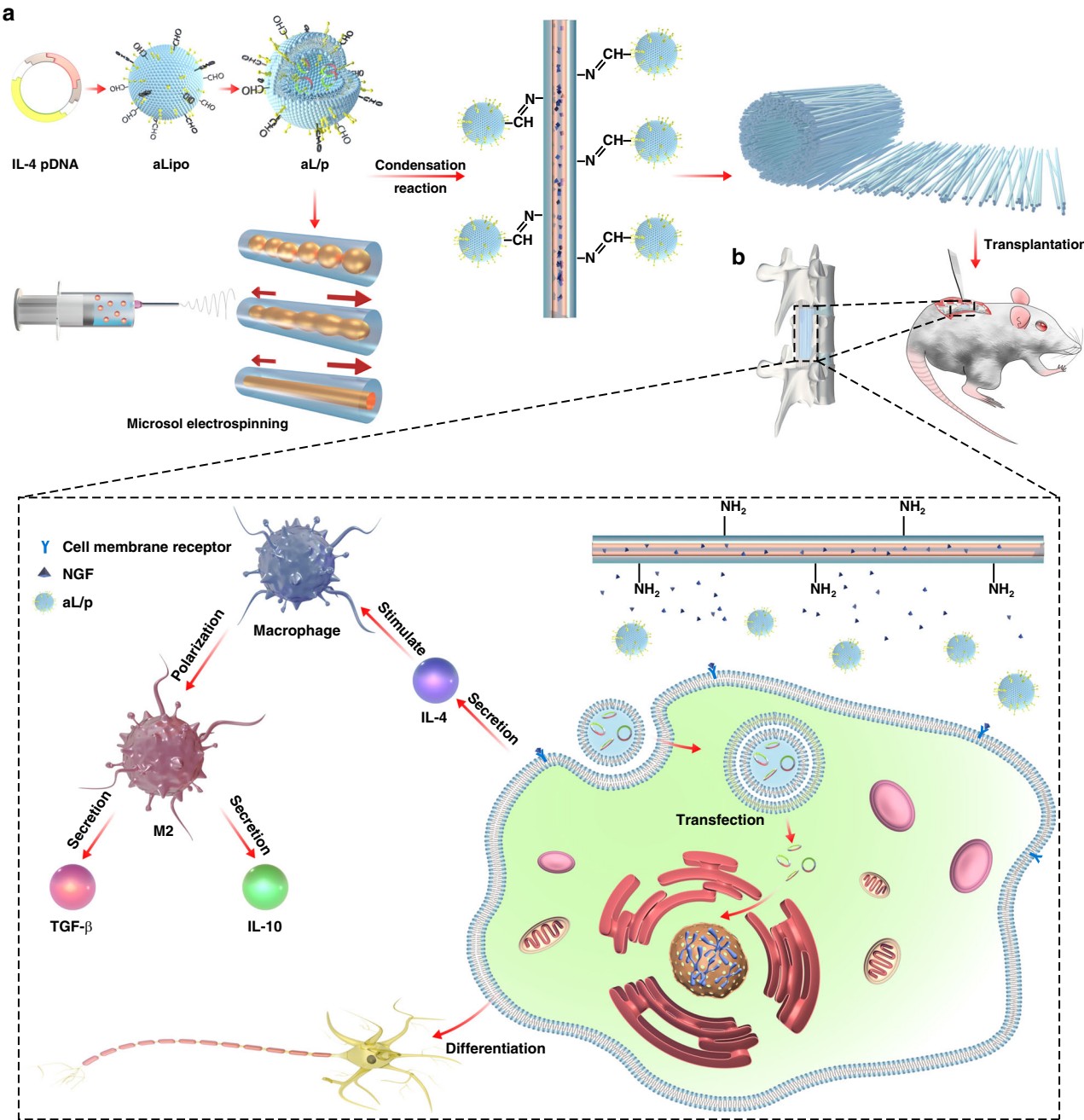

**Fig. 1 Construction and effect mechanism of composite fiber.** Scheme illustration of **a** the construction of bioinspired composite scaffold for the treatment of spinal cord injury along with **b** its microenvironment-responsive immune regulation and nerve regeneration effect.

octadecylamine for the grafting reaction. First of all, the prepared blank aLs were observed by transmission electron microscope (TEM; Fig. 2a), showing typical phospholipid bilayer membrane structure. Since the stability of liposomes could directly affect the results of grafting and transfection, the stability of liposomes was detected at different time points within 24 h after preparation (Supplementary Fig. 1b), and it was found that the concentration of phospholipids began to decrease after 1 h. However, there was no significant difference in the concentration of liposomes at each time point, proving the stability of liposomes.

After configuring different proportion of pDNA/aLiposome, it was found by dynamic light scattering (DLS) particle size analyzer (Zeta sizer, Malvern, Nano-ZS90, UK) that the particle size of liposomes decreased while the surface charge increased upon the decrease of pDNA proportion. Yet, the polydispersity index (PDI)

did not change significantly (Fig. 2b, c). This phenomenon could be attributed to the electrostatic adsorption between positively charged liposomes and negatively charged pDNA, which resulted in the electrostatic absorption of loaded pDNA onto the membrane of cationic liposomes[15]. Consequently, loading of pDNA increased particle size while reduced the surface charge of liposomes, just like the electrostatic interaction between cationic liposomes and negatively charged cell membranes or accounting. The molecular weight (Mw) of pDNA used in this study is 10.4 kb. Therefore, the pDNA/aLiposome encapsulation efficiency (Fig. 2d) of different proportion was determined by dialysis method with 500 kDa ultrafiltration centrifuge tube. The results showed that, upon the decrease of pDNA/aLiposome ratio from 1:1 to 1:3, the encapsulation efficiency increased from 50.79 ± 0.69% to 75.77 ± 1.56%. It was speculated that this result was due

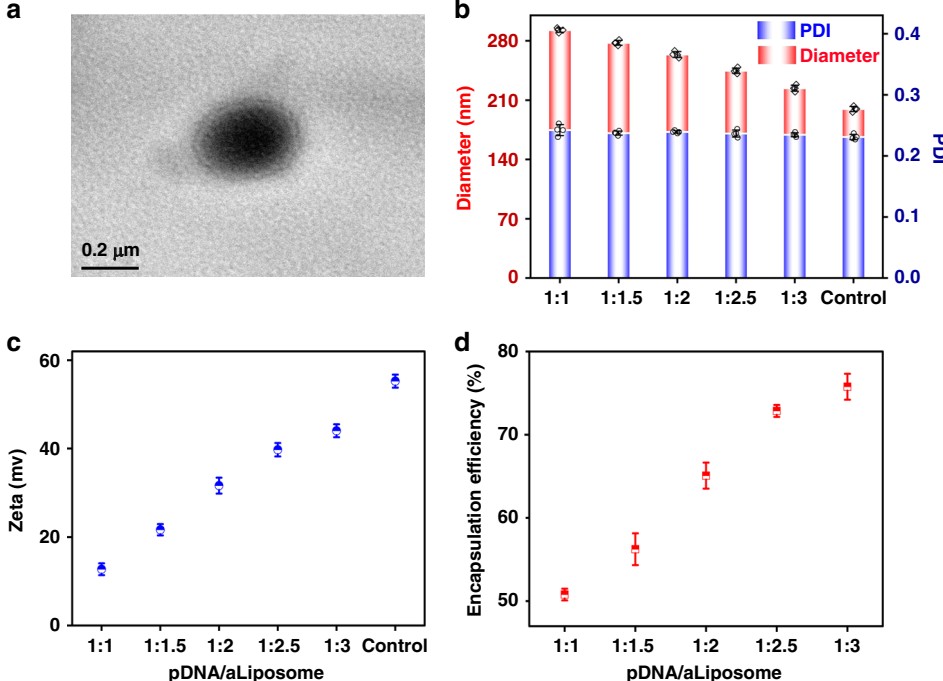

**Fig. 2 Aldehyde cationic liposomes loading different ratios of pDNA. a** The TEM image of blank aldehyde cationic liposomes (scale bar was 0.2 μm, 5 independent experiments). **b** Particle size distribution and PDI of aLiposome loading different ratio of pDNA ($n = 4$, size distribution and PDI values were mean ± std. dev.). **c** The surface charge of aLiposome loading different ratio of pDNA ($n = 3$, surface charge values were mean ± std. dev.). **d** The encapsulation efficiency of different ratio of pDNA/aLiposome ($n = 3$, encapsulation efficiency values were mean ± std. dev.).

to the decrease of the charge number on the surface of cationic liposomes and the weakening of electrostatic adsorption when the pDNA was at a high level.

Despite of the advantages such as low toxicity, ignorable immunogenicity, and superior genomic safety, non-viral vectors were still significantly bothered by their limited encapsulation efficiency and low transfection efficiency[26]. Herein, the surface of the liposomes was modified with octadecylamine to endow them with positive charge. The benefit of octadecylamine modification lay not only in increased encapsulation efficiency but also in higher transfection efficiency due to the electrostatic interaction between positively charged liposome and negatively charged cell membrane[27]. In order to characterize the transfection effect of pDNA/aLiposome, the fluorescence intensity of enhanced green fluorescent protein (eGFP) in BMSCs was observed after 72 h of transfection and the fluorescence overlap coefficient rate of nucleus and eGFP was analyzed (Fig. 3a, b). The results indicated significant differences among the groups with different pDNA/aLiposome ratios, among which the 1:2.5 group exhibited the highest fluorescence overlap coefficient rate, which was 69.32 ± 0.5%. Meanwhile, the secretion of IL-4 (Fig. 3c) was measured at different time points, and the maximum secretion of IL-4 was obtained at the 1:2.5 group after transfection for 5 days (922.12 ± 63.99 pg ml$^{-1}$). The results were consistent with fluorescence overlap coefficient rate results. In addition, according to reports, liposomes with a diameter of 200 nm–1 μm can enter the cell slowly through endocytosis, so that DNA can be released from the liposome before reaching the lysosome, which can prevent its degradation in the lysosome[28]. After screening the pDNA/aLiposome complex at various ratios in this study, it was found that the transfection efficiency was the highest at 1:2.5, and the average size was 224.1 ± 4.4 nm, so it could be used in subsequent studies.

**Physiochemical of immunological fiber scaffold.** Oriented electrospun fiber served as the template for tissue regeneration as

well as the reservoir for controlled release of neural growth factor[29]. The core–shell structure shaped in microsol electrospun fiber directly affects the regenerative effect of scaffold. In order to verify the successful construction of core–shell structure, the particle size and distribution of hyaluronic acid (HA) microsol in dichloromethane (DCM) were measured by DLS. The results showed the average particle size of 332.7 nm (Supplementary Fig. 2a) and the PDI of 0.214 ± 0.03, which indicated the uniform particle distribution. In addition, the particle size distribution did not change significantly within 2 h, indicating that the microsol particles were stable in DCM.

Through the induction of acid-sensitive Schiff base bond, liposomes were grafted on the surface of oriented fibers, which provide a platform for axonal directional growth and guide nerve fiber extension (Fig. 4a). The morphology of fibers from different groups was observed by scanning electron microscope (SEM). The results showed that, after collecting with parallel electrode receiving device, fibers presented more regular and oriented distribution (Fig. 4b–d). Among them, obvious grafted liposomes on the surface of amino polylactic acid (aPLA) microsol fibers carrying aLs loading IL-4 plasmid fibers (MSaP-aL/p) could be captured, as well as the fusion of some fibers. Such phenomenon could be explained by the lipotropic property of both aPLA fibers (aP) and phospholipids, which led to adjacent fiber fusion during the process of grafting. In addition, 100 fibers were randomly selected from the SEM images and analyzed on the distribution of fiber diameter (Fig. 4e–g). It was found that the fiber diameter was 0.554 ± 0.17, 0.56 ± 0.13, and 0.57 ± 0.19 μm for aP, aPLA microsol fibers (MSaP), and MSaP-aL/p, respectively. Meanwhile, the comparative analysis of fiber diameter distribution (Supplementary Fig. 2b) showed no significant difference in between each group. Nano-topology of neural tissue engineering biomaterials is one of the important factors affecting cell behavior. Biomimetic-oriented electrospinning fiber scaffold can provide contact guidance in vivo or in vitro for the growth and extension of

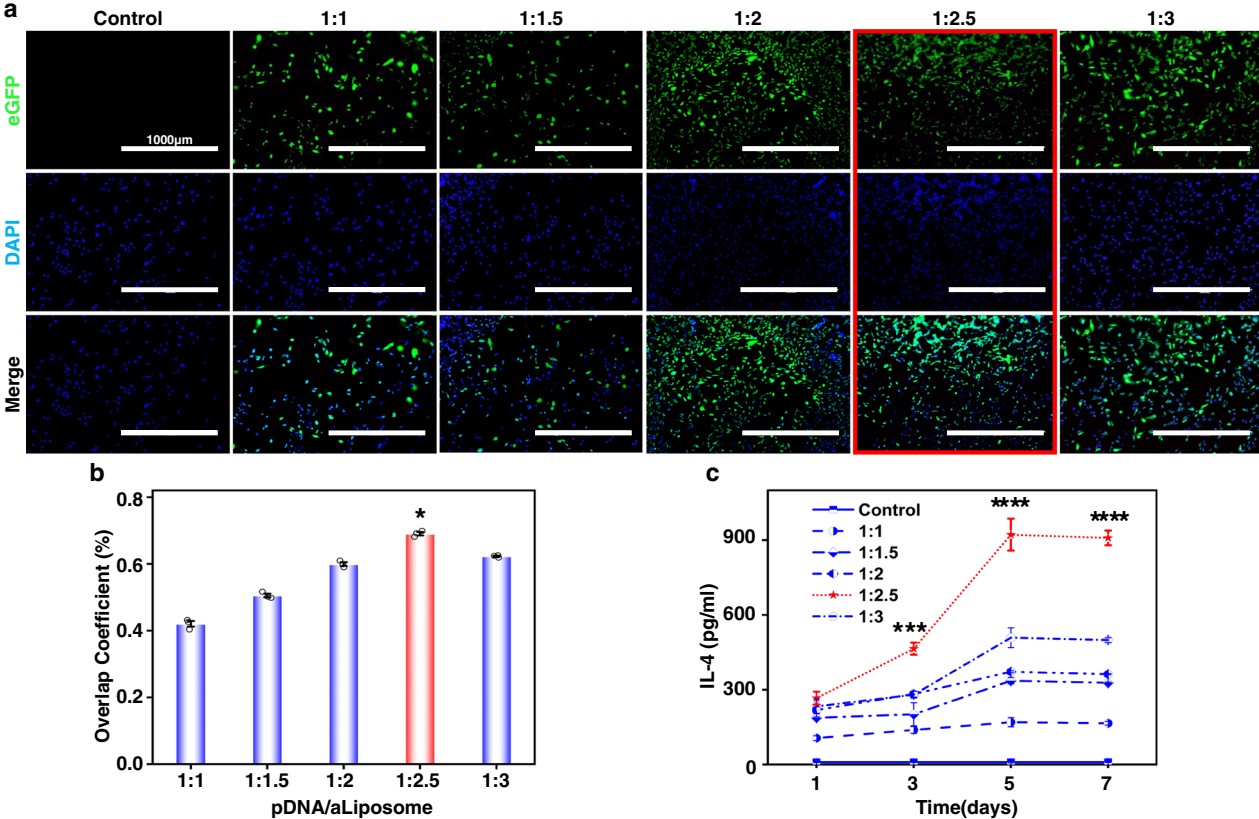

**Fig. 3 Transfection efficiency of aldehyde cationic liposomes. a** GFP expression indicated by green fluorescence in BMSCs after transfecting 72 h (scale bars were 1000 μm, 2 independent experiments). **b** Analysis of nuclear and eGFP fluorescence overlap coefficient rate ($n = 3$, values were mean ± std. dev., *$p < 0.05$ when comparing 1:2.5 and other ratios of pDNA/aLiposome via one-way analysis of variance (ANOVA) with Tukey's post hoc test). **c** Quantification of the IL-4 secretion after 1, 3, 5, and 7 days of transfection ($n = 3$, values were mean ± std. dev., ***$p < 0.001$, ****$p < 0.0001$ when comparing 1:2.5 and other ratios of pDNA/aLiposome via one-way analysis of variance (ANOVA) with Tukey's post hoc test at the same time point). The red box, column chart, and dashed line represented the highest group of transfection efficiency.

axons of in situ neurons or induced differentiated neuron-like cells. The orientations of the three groups of scaffold fibers at 0° were 10,799.59 ± 1152.65, 30,435.12 ± 3440.99, and 49,274.01 ± 2292.76, which indicated that the fiber orientation of each group is highly consistent (Supplementary Fig. 2c).

Since the controlled release capability of microsol eletrospun fibers significantly relied on its core–shell structure, TEM was employed to observe the internal structure of microsol fibers. MSaP and MSaP-aL/p groups possessed typical core–shell structure compared with bare fiber structure of aP. Also, liposomes on the surface of MSaP-aL/p could also be captured under TEM view indicating successful grafting (Fig. 4h–j).

In addition, success construction of core–shell structure and grafting of liposome could be traced by the alteration of element composition on fiber surface. In order to identify and quantify the chemical elements on the surface of each fiber scaffold, X-ray photoelectron spectroscopy was conducted and showed no significant difference in carbon peak (C 284.9 eV), nitrogen peak (N 398.8 eV), and oxygen peak (O 533.4 eV) on the surface of aP and MSaP fibers (Fig. 4k–m), suggesting no HA microsol particles on the fiber surface. However, significant increases on the carbon peak, nitrogen peak, and oxygen peak, as well as the appearance of phosphorus peak (P 133.0 eV), suggested the existence of grafted aLs on the fiber surface.

The average surface roughness (Ra, roughness) of atomic force microscopic images was measured by the Nanoscope Analysis 1.7 software (Fig. 4n–p). The surface Ra of aP and MSaP fibers were 249 and 440 nm, respectively. On the other hand, the Ra of

MSaP-aL/p fiber is 648 nm, and the phase diagram showed round protruding liposomes on the surface of MSaP-aL/p fibers, indicating stable integration of liposomes on the fiber surface.

Owing to the physiological activity of spine as well as the influence of surrounding tissue, superior mechanical properties such as elasticity were required to some extent for the maintenance of biomimetic scaffold after implantation[30]. The stress–strain curves of all fiber scaffolds show certain degrees of tensile strength. Owing to the existence of aminated PLA and HA core–shell, the maximum tensile strength of MSaP decreased slightly compared with PLA fibers (PLA) and aP (Supplementary Fig. 3a). However, the maximum tensile strength of MSaP-aL/p was increased, and Young's modulus was 0.33 ± 0.004 Mpa, which was significantly higher than that of other scaffolds (0.27 ± 0.004, 0.26 ± 0.003, 0.19 ± 0.005 Mpa; Fig. 5a). The improvement on mechanical property could be attributed to the chemical bond between liposomes and fibers, which caused partial fusion between these two hydrophobic materials, bringing higher maximum tensile strength and Young's modulus.

Biomaterials should be degraded gradually after providing support for tissue repair and regeneration and should not hinder tissue regeneration, so degradability is an important property of biomaterials[31,32]. In the study of degradation in vitro (Fig. 5b), it was found that the weight of MSaP-aL/p decreased to 89.48 ± 0.59% on the sixth day, indicating a significantly faster degradation rate than other scaffolds. This is due to the fact that the hydrophilic amino group improves the hydrophobic properties of PLA and speeds up the hydrolysis of ester bonds on the

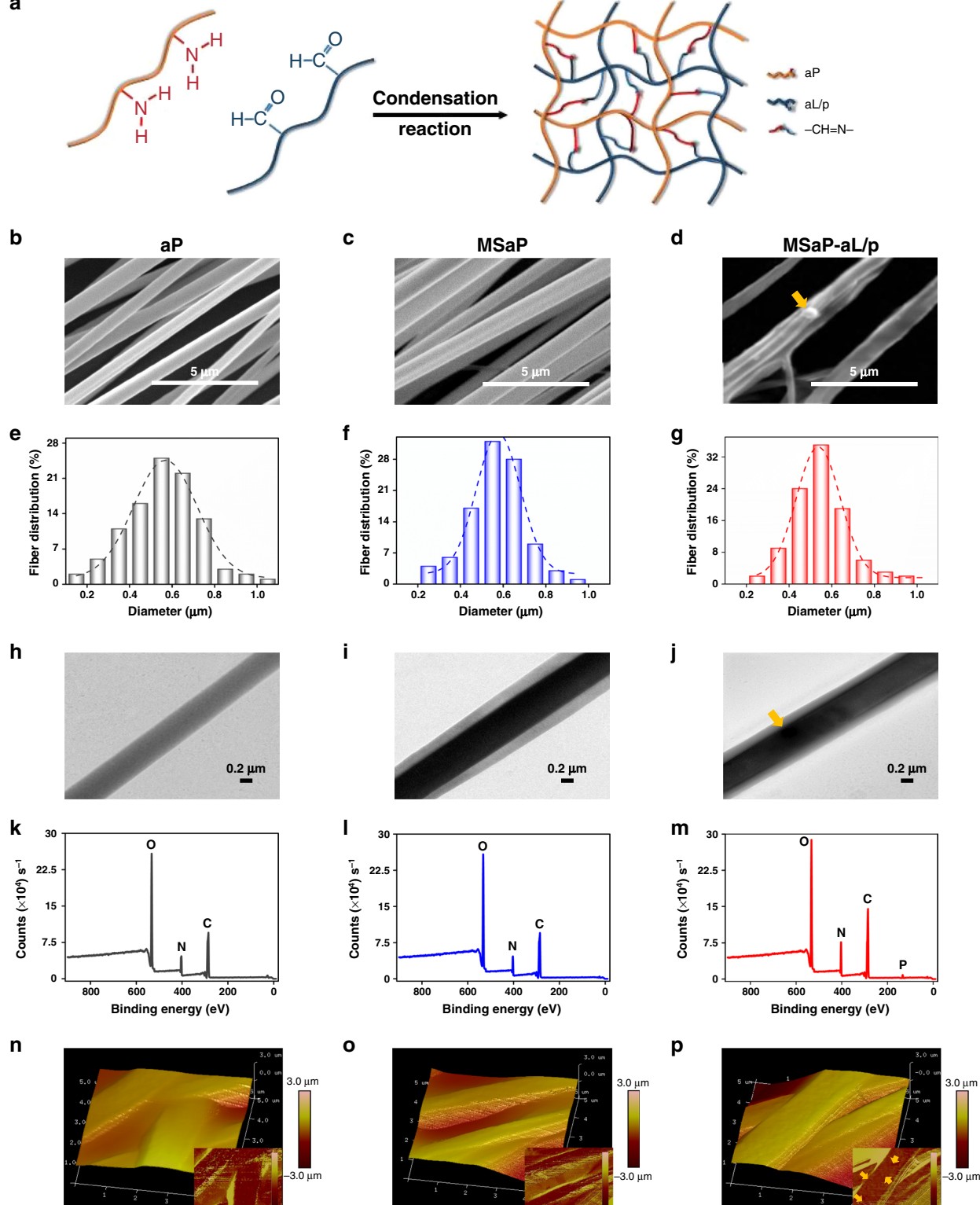

**Fig. 4 The morphology of different fiber scaffolds. a** Schematic diagram of amino polylactic acid microsol electrospun fiber and aldehyde cationic liposomes connected with Schiff base. **b–d** SEM images of different fiber scaffolds (three independent experiments). **e–g** Histogram of frequency distribution of different fiber scaffolds diameters (n = 100 fibers from each sample). **h–j** TEM images of different fiber scaffolds (four independent experiments). **k–m** XPS indicating chemical elements on different fiber scaffold surfaces. **n–p** AFM images of scaffolds (three independent experiments). The yellow arrows in the images indicate the location of liposome (n = 3, all SEM scale bars were 5 μm, all TEM scale bars were 0.2 μm).

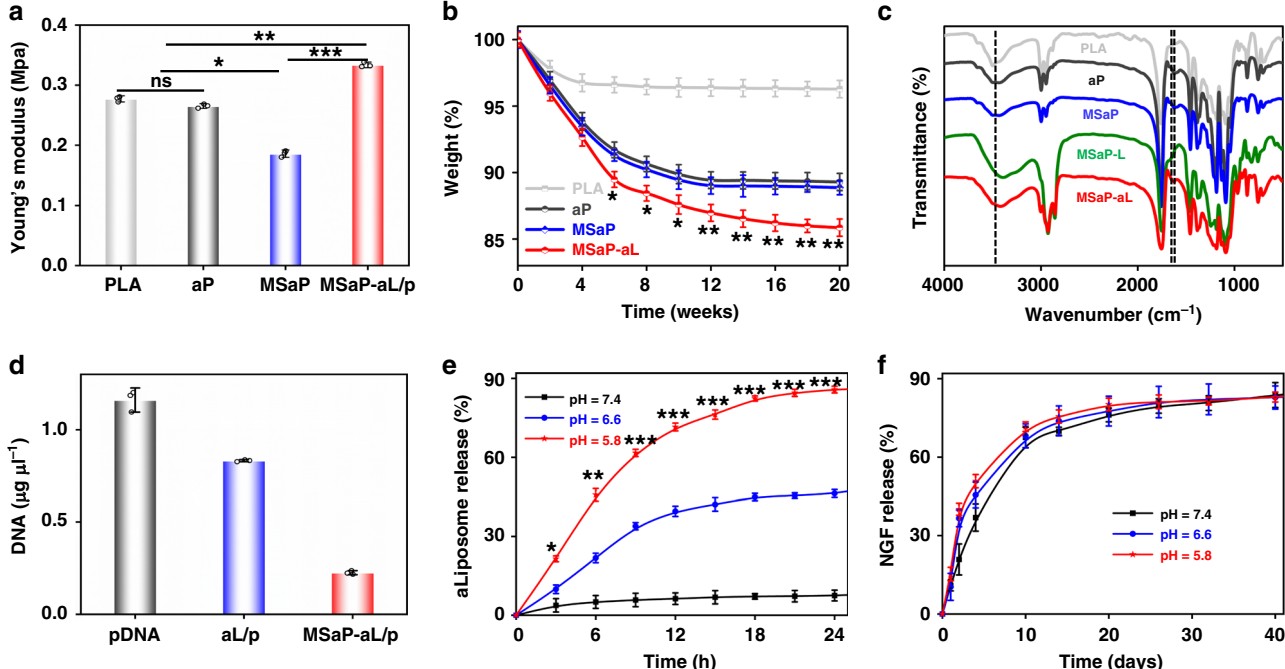

**Fig. 5 The physiochemical performance of nanofibrous scaffolds. a** Young's modulus of different fiber scaffolds ($n = 3$, values were mean ± std. dev., *$p <$ 0.05 when comparing MSaP and PLA (0.021) and aP (0.015) groups, respectively, **$p < 0.01$ when comparing MSaP-aL/p and PLA (0.0062) and aP (0.0043) groups, respectively, and ***$p < 0.001$ when comparing MSaP-aL/p and MSaP (0.00054) via one-way analysis of variance (ANOVA) with Tukey's post hoc test). **b** Degradation curve of different fiber scaffolds ($n = 3$, degradation values were mean ± std. dev., *$p < 0.05$, **$p < 0.01$ when comparing MSaP-aL/p and three other samples at the same time point via two-way analysis of variance (ANOVA) with Tukey's post hoc test). **c** FTIR spectra of different fiber scaffolds. **d** Concentration of pDNA in the aLiposome solutions before and after grafting at the surface of scaffold ($n = 3$, concentration of pDNA values were mean ± std. dev.). **e** Cumulative release curve of aLiposomes in different pH environments ($n = 3$, aLiposome release values were mean ± std. dev., *$p < 0.05$, **$p < 0.01$, ***$p < 0.001$ when comparing pH 5.8 and the other two pH at the same time via two-way analysis of variance (ANOVA) with Tukey's post hoc test). **f** Cumulative release curve of NGF in microsol electrospinning in different pH environments ($n = 3$, NGF release values were mean ± std. dev., statistical analysis evaluated by two-way analysis of variance (ANOVA) with Tukey's post hoc test).

main chain[32]. In addition, liposomes with hydrophilic properties may accelerate this effect. As one of the most important factors in tissue engineering, biocompatibility was significantly affected by the hydrophobicity/hydrophilicity of materials. Therefore, measurement of water contact angle was employed, which showed the contact angles of 120.37 ± 0.61°, 86.76°±0.27°, 86.17 ± 0.15°, and 80.37 ± 0.41° for PLA, aP, MSaP, and MSaP-aL/p fibers, respectively (Supplementary Fig. 3b). It was found that the hydrophobicity of the scaffold surface was significantly reduced by introducing hydrophilic amino group onto PLA, and contact angle of MSaP-aL/p fiber decreased most significantly due to the hydrophilicity of surface liposome phospholipids, which extended into the aqueous phase to enhance their hydrophilicity. One of the advantages of electrospinning fiber scaffolds becoming a hot spot in tissue engineering research is that it has large porosity, which can provide more growth space for cells. As mentioned above, there is no statistically significant difference in the diameter of the three groups of fiber scaffolds, so the factors that change the porosity due to different diameters can be excluded. The experimental group MSaP-aL/p was compared with the two aP and MSaP groups, and the porosity increased significantly (Supplementary Fig. 3c). This phenomenon can also be explained by the lipophilicity of aminated PLA and phospholipids; during the grafting process, some adjacent fibers fused, and the internal bulk density decreased, resulting in increased porosity.

**The microenvironment-responsive release effect.** Grafting of liposomes onto fiber with Schiff base bond provided the fundament for the microenvironment-responsive release of transfecting

vector. In order to verify the successful grafting, Fourier transform infrared spectroscopy was employed to study the alteration of chemical bond on fibers (Fig. 5c). Compared with bare PLA fiber, aminated PLA showed obvious $NH_2$ band at 3420 cm$^{-1}$. No significant differences could be found between infrared spectras of aPLA, MSaP, and aPLA microsol fibers carrying blank cationic liposomes (MSaP-L) groups, verifying the absence of HA characteristic band in MSaP. It is reasonable to conclude that all the microsol particles were wrapped in the core part of the fiber during the spinning process. After the introduction of the aldehyde-free liposomes, since the liposomes had the composition of lecithin, there was a C=O band at 1720 cm$^{-1}$, and a band appeared at 1655 cm$^{-1}$ in aPLA microsol fibers carrying blank aLs group (MSaP-aL) indicating the formation of -C=N- bond. It could be confirmed that the aLs were stably grafted on the surface of aPLA microsol fiber scaffold by acid-sensitive chemical bond Schiff base.

Since the grafting of liposome was directly associated with the transfection activity of the loaded vector, grafting efficiency was measured through quantification of loaded pDNA. After the grafting of pDNA-loaded aLs, the initial concentration of pDNA was quantified to be 1.16 ± 0.06 µg µl$^{-1}$. On the other hand, the concentration of pDNA integrated on the surface of the fiber scaffolds was found to be 0.22 ± 0.01 µg µl$^{-1}$, indicating that the fiber scaffold carried about 19% of the initial concentration of pDNA (Fig. 5d).

Severe injury of local blood vessels caused by external mechanical damage of spinal cord will lead to a sharp decrease of pH, resulting in an acidic microenvironment in the injured area[2,33]. Based on this pathological characteristic, the focal pH

was simulated in vitro for the characterization of microenvironment-responsive releasing functionality. Specifically, the releasing rates of liposome were tested through immersing the liposome-grafted fiber scaffold in phosphate-buffered saline (PBS) solutions with different pH (Fig. 5e). In the acidic environments with pH of 5.8 and 6.6, the release of liposomes increased gradually with time. The release rate of liposomes under the pH of 5.8 reached 61.66 ± 1.38% within 10 h, which was significantly higher than that of other pHs. Therefore, quick response to acidic environment of the responsive fiber scaffold could be verified in vitro, which provided basis for further study.

As measured, the drug loading efficiency of all NGF-loaded microsol electrospun scaffolds were about 79.53 ± 1.44% in this study. Unlike the quick responsive release of liposome for acute-stage immunoregulation, the therapeutic effect of NGF required a controlled and prolonged release that is not sensitive to environment. Hence, the release characteristics of NGF from microsol electrospun fibers under different pH were also evaluated (Fig. 5f). The results showed that the release rate under the pH of 5.8 was slightly increased in acidic environment, reaching 71.64 ± 1.82% (544.21 ± 21.41 ng) on the tenth day, while it was 68.14 ± 4.59% (545.09 ± 36.72 ng) and 59.54 ± 1.93% (476.29 ± 15.41 ng) under pH of 6.6 and 7.4, respectively. All scaffolds under different pH exhibited sustained releases of NGF, which lasted for >40 days. We could conclude that the microsol fibers with core–shell structure had a mild response to the acidic microenvironment and was able to provide a stable platform for controlled release of NGF, which was pivotal for the regeneration of injured spinal cord.

**Evaluation of cell biological characteristics**. In this study, BMSCs and bone marrow macrophages (BMMs) were co-cultured through Transwell to simulate the effects of fiber membranes on cell-to-cell interaction in the microenvironment of SCI and to evaluate the biological characteristics of nerve fiber membranes.

As the template for cell adhesion and proliferation, superior cytocompatibility was required for the fiber membrane. Cellular survival and proliferation characteristics were also evaluated via seeding BMSCs on different fiber membranes. Live and dead cells were stained after culture for 3 days, and three different visual fields were selected to count the number of living and dead cells. Because of the hydrophobic property and relatively low mechanical hardness of aP, the fluorescence of living cells in each fiber membrane was lower than that in the control group, and the fluorescence semi-quantitative results were similar to those in the picture (Supplementary Fig. 4a, b). Cell proliferation assay was conducted with the CCK8 Kit, which showed no significant difference between different groups in absorbance on 1 and 3 days. However, on the fifth day, due to superior mechanical hardness and no biological factor interference, the cell proliferation rate in the control group was significantly higher than the other groups, while the absorbance in the MSaP-aL/p group was slightly lower than other fiber groups, which could be attributed to the transfection effect of liposomes on the fibers and the biological factors loaded in microsol fibers. But there was no significant difference among the membrane groups (Supplementary Fig. 4c).

Integrin is a transmembrane protein receptor that plays a key role in signal transduction between extracellular matrix and cytoskeleton[34]. It was shown that, compared with the control group, the extended morphology of BMSCs on oriented fiber membranes was significantly affected, consistently showing the long axis of cytoskeleton parallel with the direction of oriented fibers (Supplementary Fig. 5a). In addition, the expression of integrinβ1 indicated by green fluorescence showed that the fiber membranes could promote the expression of integrinβ1 subunit, which indirectly indicated that the fiber membrane could provide favorable adhesion ability for cells. The fluorescence semi-quantitative analysis (Supplementary Fig. 5b) showed that there was no significant difference between the fiber membranes, but it was lower than the control group. This could be explained that the petri dishes have higher extracellular matrix hardness as mentioned above to provide better biomechanical stress for cells. The morphology of cell adhesion observed by SEM was similar to that of integrin staining (Supplementary Fig. 5c–f). The cell adhesion morphology was also consistent with the direction of fiber membrane and grew well.

**Characterization of immunomodulatory properties**. The expression of pro-inflammatory and anti-inflammatory genes in BMMs cultured in different pH was detected after co-culture for 1, 3, 5, and 7 days. In the culture medium with pH of 5.8, the expression of pro-inflammatory genes IL-1β and TNF-α in the MSaP-aL/p group decreased gradually as time went on, reaching the lowest expression of 9.63 ± 0.80- and 0.30 ± 0.03-fold (Fig. 6a, d) on the seventh day, respectively. At the same time, the expression of anti-inflammatory gene IL-10 and transforming growth factor (TGF)-β were 3.42 ± 0.07-, 11.59 ± 0.09-, 27.02 ± 0.45-, and 29.30 ± 0.51-fold and 2.85 ± 0.06-, 7.99 ± 0.07-, 22.52 ± 0.37-, and 27.47 ± 0.59-fold on the first, third, fifth, and seventh day, respectively. Expression of all anti-inflammatory genes showed an upward trend (Fig. 6g, j), which was significantly different from that of the groups ($p < 0.01$). Under the pH of 6.6, the MSaP-aL/p group also showed response to acidic environment. However, due to the decreased acidity, the response of Schiff base to environment decreased, followed by relatively less significant changes on pro-inflammatory and anti-inflammatory gene expression, compared with those under pH of 5.8 (Fig. 6b, e, h, k). Under the pH of 7.4, the differences on pro-inflammatory and anti-inflammatory gene expression was smaller and showed insignificant difference between different groups at each time point. Surprisingly, pro-inflammatory genes, IL-1β and TNF-α, showed a downward trend while anti-inflammatory genes, IL-10 and TGF-β, showed an upward trend (Fig. 6c, f, i, l). This could be explained that the lower chamber BMSCs had a certain degree of immunoregulation function, could shift M1 macrophages into M2 macrophages, and inhibit the expression of inflammatory genes[35].

Similarly, the secretion of pro-inflammatory and anti-inflammatory cytokines detected by enzyme-linked immunosorbent assay (ELISA) showed a downward trend of pro-inflammatory factors IL-1β and TNF-α under pH of 5.8. However, the secretion of anti-inflammatory factors IL-10 and TGF-β increased gradually. Under the pH of 6.6 and 5.8, the results were consistent with those of quantitative real-time polymerase chain reaction (qRT-PCR). It could be concluded that the constructed nerve fiber membrane was highly responsive to the microenvironment of SCI, and the rapid release of loaded plasmid liposomes played its role in transfection, with the resulting IL-4 inducing the polarization of microglia and macrophages to M2 type in time (Supplementary Fig. 6).

Immunofluorescence staining of BMM subtypes under the pH of 5.8 on the seventh day showed that BMM marker F4/80 (red fluorescence) was highly expressed in every group, indicating the high purity of BMMs. The measurement of M1 subtype marker, inducible nitric oxide synthase, showed the fluorescence amount of 2.82 ± 1.25 in the MSaP-aL/p group, which was significantly lower than other groups (Supplementary Fig. 7a, b). The expression of M2 marker, CD206, exhibited higher signal in the MSaP-aL/p group, with the semi-quantitative showing value of

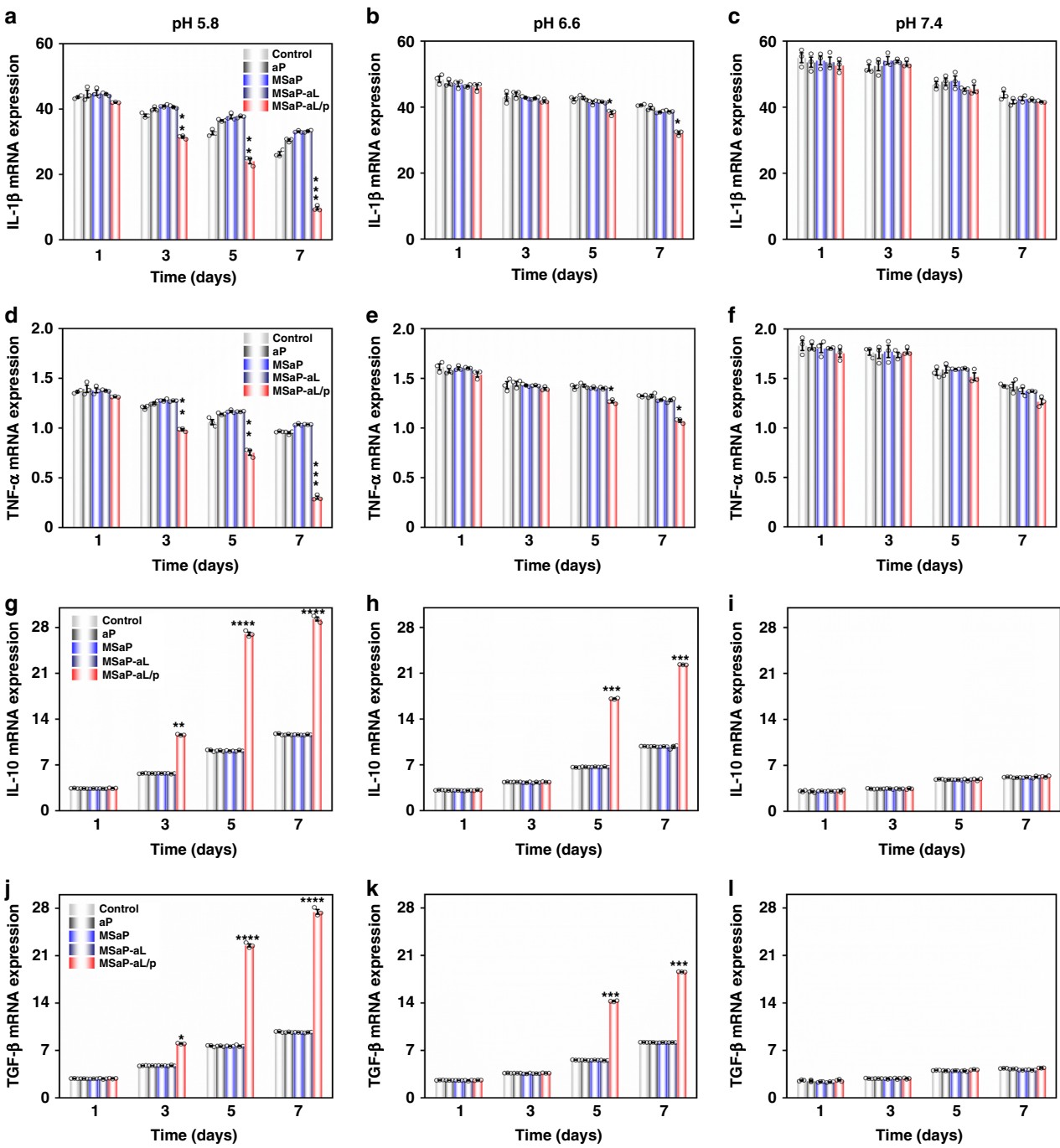

**Fig. 6 Gene expression of BMMs in different pH culture environments.** The expression of **a–c** IL-1β mRNA, **d–f** TNF-α mRNA, **g–i** IL-10 mRNA, and **j–l** TGF-β mRNA at different pH of BMMs ($n = 3$, all values were mean ± std. dev., $*p < 0.05$, $**p < 0.01$, $***p < 0.001$, and $****p < 0.0001$ when comparing MSaP-aL/p and other groups via two-way analysis of variance (ANOVA) with Tukey's post hoc test).

$11.06 ± 0.80$, which was significantly higher than the other groups (Supplementary Fig. 7c, d). We could confirm that, in acidic culture environment, with the help of MSaP-aL/p fiber membrane, BMMs was induced to M2 subtype. Combined with the aforementioned results of qRT-PCR and ELISA, the responsive fiber membrane could significantly inhibit BMM pro-inflammatory factor and promote the secretion of anti-inflammatory factor, which could create a favorable immune microenvironment for subsequent nerve regeneration.

**Characterization of neurogenic activity.** Endogenously secreted NGF was responsible for the maintenance of multiple

neurophysiological functions[36]. BMSC was introduced to mimic endogenous stem cell and investigate the biological effect of NGF-loaded fibers. After culture on different fiber membranes for 10 days, immunofluorescence staining and qRT-PCR technique were employed to detect the expression of neuron-specific markers in neuron-like cells, including neurofilament protein, neuron-specific enolase, nerve cytoskeleton tubulin (Tau protein), and neuron cell-specific differentiation tubulin (Tuj-1), in order to evaluate the neurogenic activity of fiber membranes. As indicated in the images of immunofluorescence staining (Fig. 7a), BMSCs seeded on fiber membranes with morphology arranged directionally along the fiber distribution direction exhibited

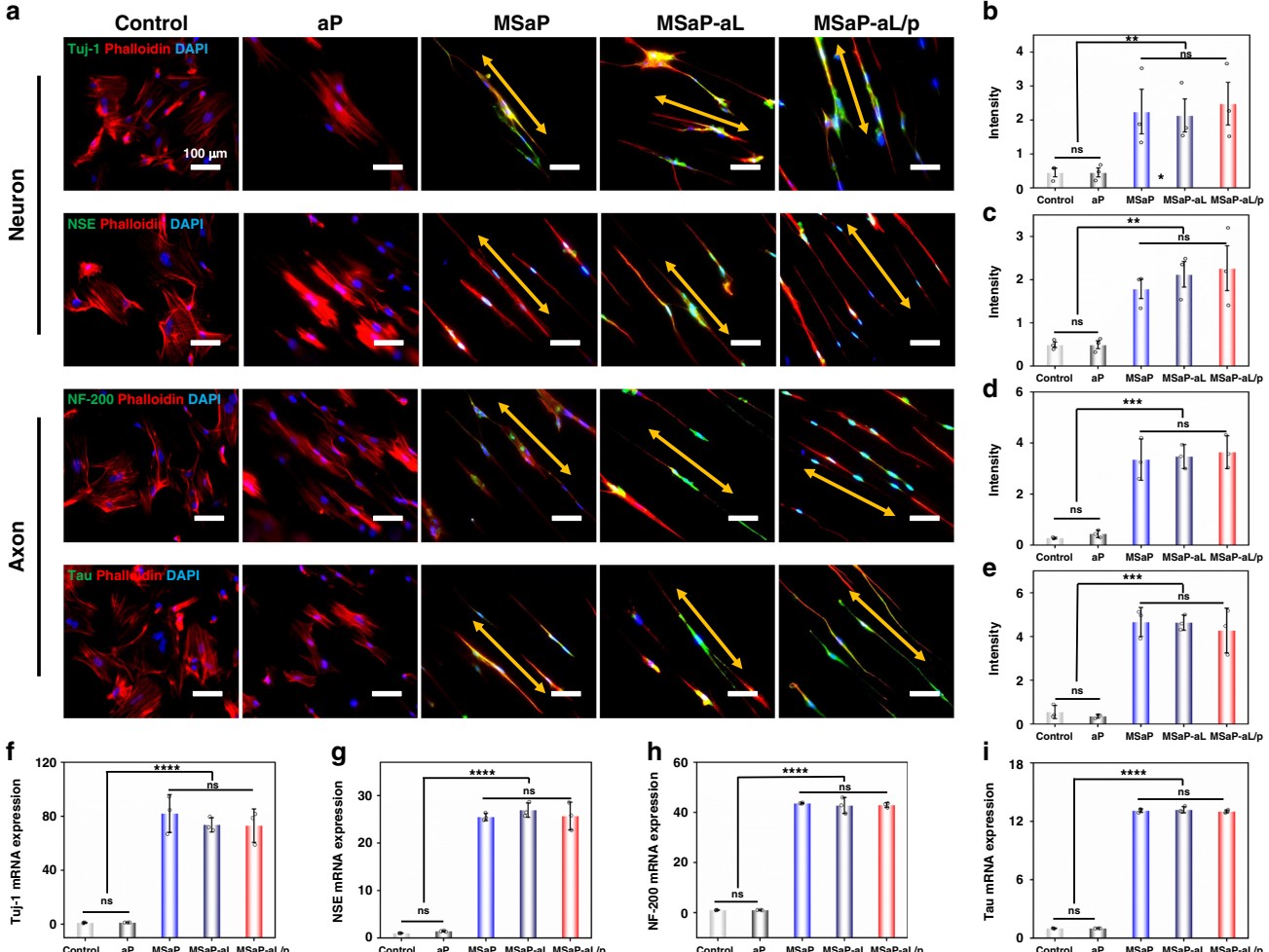

**Fig. 7 Neuron-specific marker expression. a** Immunofluorescence staining of Tuj-1, NSE, NF-200, and Tau staining images on different fiber scaffolds (scale bars were 100 μm, 4 independent experiments) with **b–e** corresponding fluorescence semi-quantitative analysis. **f–i** mRNA expression analysis of Tuj-1, NSE, NF-200, and Tau. The yellow bi-directional arrows represented the contact guiding effect of oriented nanofibers on stem cells ($n = 3$, all values were mean ± std. dev., **$p < 0.01$, ***$p < 0.001$, and ****$p < 0.0001$ when comparing microsol electrospun groups (MSaP, MSaP-aL, MSaP-aL/p) and control and aP groups, respectively, via one-way analysis of variance (ANOVA) with Tukey's post hoc test, ns not significant).

different expression levels of neurogenic marker. All of the three NGF-loaded fiber membranes including the MSaP-aL/p, MSaP, and MSaP-aL groups showed typical neurogenic effects with the seeded cell exhibiting neuron-like changes as indicated by axon-like and dendritic processes around cell bodies. Intense green fluorescent signal could be captured within cells seeded on MSaP-aL/p, MSaP, and MSaP-aL fiber membranes compared with weak signal on non-loaded fiber (aP) and control groups. Semi-quantitative analysis of fluorescent signal (Fig. 7b–e) also revealed significantly higher expression levels of neurogenic markers in the NGF-loaded groups. It is suggested that the microsol electrospun fiber membrane has the function of controlled release of NGF and can continuously promote the neural differentiation of endogenous stem cells. Meanwhile, similar tendency could also be found through the introduction of qRT-PCR (Fig. 7f–i), which revealed the significantly higher expressed neuron-specific marker genes in the NGF-loading groups, further proving the neurogenic effect of NGF loading in microsol fibers.

During the neural regeneration process, it is necessary to provide long-term biological signal supply chains for neural progenitor cells. Therefore, it is very important to prove that the sustained release of NGF from biomimetic electrospun fibers has biological activity. The neuron-specific marker Tuj-1 was still able to label neuron-like cells differentiated from BMSCs induced by

release solution (Supplementary Fig. 8a). At the same time, without the contact guidance of oriented electrospinning fibers, the cell morphology changed from spindle to long, but no neurite-like formation was observed. The PC (positive control group) is higher in the semi-quantitative fluorescent results than the r-NGF group, but there was no statistical difference between the two groups (Supplementary Fig. 8b). However, the negative control group was significantly different from the r-NGF and the PC groups, respectively. This result combined with in vitro simulated release experiments proved that the biomimetic fibers can sustain the released NGF and that it can be protected by the core–shell structure to maintain its biological activity.

**Foreign body reaction induced by nerve bundles.** Foreign body reaction is an immune defense response mediated by macrophages, foreign body giant cells and granulation, and finally the formation of fibrous tissue around the biomaterial[37]. In order to look into the potential response of in vivo immune system, different membranes were implanted under the epidermis on the back of SD rats (Supplementary Fig. 9c). At 2 weeks after operation, hematoxylin and eosin (H&E) staining showed that the distribution of inflammatory response band of MSaP-aL/p was narrower than that of other groups, and the edge was smooth

(Supplementary Fig. 9a). The area of inflammatory response area was semi-quantitatively analyzed by the ImageJ software, which showed significantly smaller inflammatory area of MSaP-aL/p (3.28 ± 0.13%) compared with other groups (Supplementary Fig. 9b). These differences could be explained that, after the implantation of the composite fiber membranes, ischemia, infiltration of immune cells and inflammatory cytokines reduces the local pH, the subcutaneous inflammatory acidic microenvironment promotes the release of IL-4 plasmid-loaded liposomes into the surrounding tissue cells to secrete IL-4, and finally transformed the M1 to the M2 phenotype, as described in the literature[38]. It could be confirmed that the responsive biomimetic microfibers–nanofibers had favorable biocompatibility and low immunogenicity, which laid the foundation for further evaluation of their biological characteristics.

**Animal motor function**. The in vivo performance of fiber bundle (Fig. 8a (1), (2), (3), (4)) was evaluated by SD rat T9 spinal cord hemi-section model. The recovery of hindlimb motor dysfunction after SCI was evaluated by Basso, Beattie, Bresnahan (BBB) and inclined plane test (IPT) scores every week after operation. The results showed that the hindlimb motor function of rats in each group recovered in varied degrees. Superior recovery of motor function could be identified in the MSaP-aL/p group compared with other groups, as indicated by significantly higher BBB score and IPT score at each time points from fourth week postinjury. At the eighth week, the BBB score of 13.71 ± 1.11 and the IPT score of 55.86 ± 4.67° could be found in the MSaP-aL/p group, both of which were significantly higher than those of the other groups (Fig. 8b, c). In addition, the minimum time required to

achieve similar motor functions during recovery is more convincing in characterizing the functionality of neural repair materials. Therefore, in the comparison of 10 points (the highest average score of the negative control group in the BBB score) and 47° (the highest average angle of the negative control group in the IPT test), the two scores of the MSaP-aL/p group exceeded the control values at the fourth week after operation, reaching 11.14 ± 2.04 (BBB) and 52.43 ± 5.56° (IPT), respectively. The results of motor function recovery suggested that immunoregulatory fiber bundle (MSaP-aL/p) implantation could reduce the risk of further damage to surviving motor neurons by effectively inhibiting the acute inflammatory response of SCI and promoted nerve repair in the subsequent stage of secondary SCI.

**Immune regulation in SCI in vivo**. Macrophages and microglia reached the recruitment peak within 7 days after SCI, so immunomodulatory bundles should play the role of biological response regulation during this period. On the seventh day after the implantation of biomimetic immunomodulatory fiber bundles, we first tested whether they could guide changes in local immune cell subtypes (Fig. 9a). There was no significant difference in the number of CD11b/CD86-positive macrophages between the non-immunomodulatory fiber bundle groups (ap, MSaP, MSaP-aL) and the blank control group (Fig. 9b). However, due to the large porosity of oriented electrospinning fiber bundles, it could lead to the increase of CD11b/CD206-positive macrophages (Fig. 9c)[39]. In addition, the in vivo fiber bundle regulation of macrophage polarization was surprisingly consistent with the conclusion drawn in the in vitro study that the MSaP-aL/p group not only significantly decreased the proportion of CD11b/CD86-

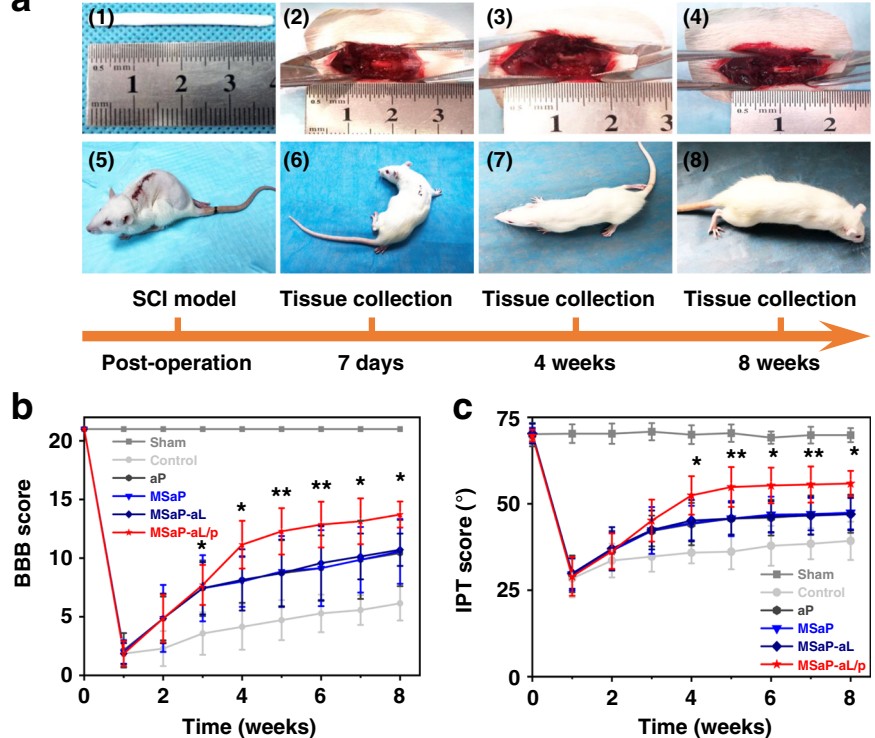

**Fig. 8 Timeline of animal assay and motor function score. a** Pictures of animal experiments and tissue collection timeline. Image of (1) the responsive fiber bundle, (2) the fully exposed T9 spinal cord, (3) 3 mm hemi-section made in the right site of the T9 spinal cord (the yellow arrow indicated the injured site), (4) implanted fiber bundles (the yellow arrow indicated the fiber bundle implantation position) along with (5), (6), (7), (8) post-operation condition of rat at 7 days, 4 weeks, and 8 weeks (3 independent experiments). **b** Evaluation of motor function recovery of lower limb by BBB score in rats assisted by **c** rats' motor function IPT score (n = 7, all values were mean ± std. dev., *p < 0.05, **p < 0.01 when comparing MSaP-aL/p and other spinal cord injury groups at the same time point via two-way analysis of variance (ANOVA) with Tukey's post hoc test).

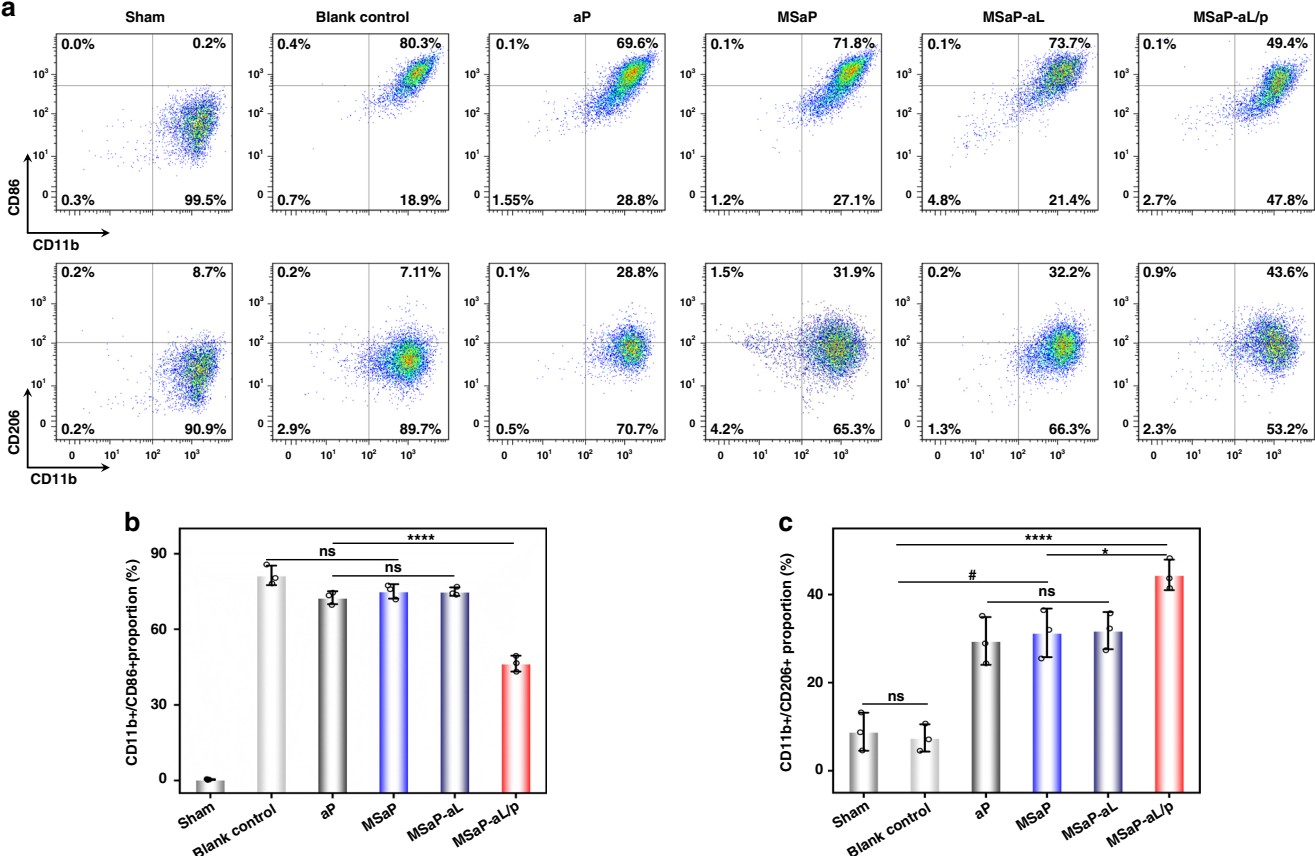

**Fig. 9 Phenotypic identification of immune cells in vivo. a** Flow cytometric analysis (3 independent experiments) of CD11b/CD86-positive cells with **b** its quantification analyzed of the ratio and CD11b/CD206-positive cells with **c** its quantification analyzed of the ratio ($n = 3$, all values were mean ± std. dev., #$p < 0.001$ when comparing non-immunoregulatory fiber groups and blank control group and *$p < 0.05$, ****$p < 0.0001$ when comparing MSaP-aL/p and other control groups via one-way analysis of variance (ANOVA) with Tukey's post hoc test, ns not significant).

positive macrophages but also significantly increased CD11b/CD206-positive macrophages. The results of flow cytometric analysis showed that the implantation of immunomodulatory fiber bundles could promote the polarization of local microglia/macrophages to M2 phenotype in vivo.

The immunoregulatory function of biomimetic composite fiber bundle during acute SCI is based on promoting the local secretion of IL-4 cytokines. Anti-IL-4 was used to label IL-4 cytokines secreted by biomimetic bundles in response to acidic microenvironment of SCI. According to the results of immunofluorescence semi-quantitative analysis, more fluorescent-stained IL-4 cytokines were found in the MSaP-aL/p group than in the other bundle transplantation groups (Fig. 10a, b). The transformation of microglia/macrophage subtypes at the injured site was bound to cause changes in inflammatory factors. The marked cytokines secreted by M1 and M2 immune cells were marked by anti-TNF-α and anti-IL-10, respectively, which further reflected the immunomodulatory effect of biomimetic bundle. The secretion of TNF-α, a landmark cytokine of M1 immune cells, in the MSaP-aL/p group was significantly lower than that in the blank control group and other bundle transplantation groups (Fig. 10c). However, the secretion of fluorescent-labeled IL-10 in the MSaP-aL/p group was significantly higher than that in the other control groups. Meanwhile, according to the results the expression of IL-4 gene in the MSaP-aL/p group was significantly higher than that of the other groups (Fig. 10d). Combined with the above fluorescence-labeled IL-4 results, it was further proved that it had played the function of transfection in response to the acidic environment of SCI. The expression levels of pro-inflammatory

genes (TNF-α and IL-1β) in MSaP-aL/p were significantly decreased (Fig. 10e, f). On the other hand, the expression levels of anti-inflammatory genes IL-10 and TGF-β were significantly higher than those in the other control groups, respectively (Fig. 10g, h). The results showed that the immune bundles had played an immunomodulatory role during acute SCI. In addition, strong green fluorescence of eGFP could be observed in spinal cord tissue of the aPLA microsol fibers carrying aL loading eGFP plasmid (MSaP-aL/g) and MSaP-aL/p groups, showing no significant difference in semi-quantitative between these two groups (Supplementary Fig. 10a). On the other hand, no green fluorescence could be detected in the MSaP-aL group compared with the other two groups (Supplementary Fig. 10b). Meanwhile, M2 macrophages labeled against CD206 with red fluorescence were found in all three groups to distribute around the cells. Corresponding semi-quantitative analysis showed significantly lower CD206 levels in the MSaP-aL and MSaP-aL/g groups without IL-4 plasmid when compared with the plasmid-containing MSaP-aL/p group (Supplementary Fig. 10c). Taking the results of IL-4 immunofluorescent staining into consideration, which showed significantly higher IL-4 expression in the MSaP-aL/p group (Fig. 10a, b), it could be confirmed that the cationic liposomes loading with IL-4 plasmid have been responsively transfected into the surrounding tissue to regulate the polarization of macrophages.

The systemic inflammatory response of animals was evaluated by ELISA detection of serum to comprehensively assess the immunoregulatory function of fiber bundles. Owing to the large amount of IL-4 secreted into the systemic circulation at the injury

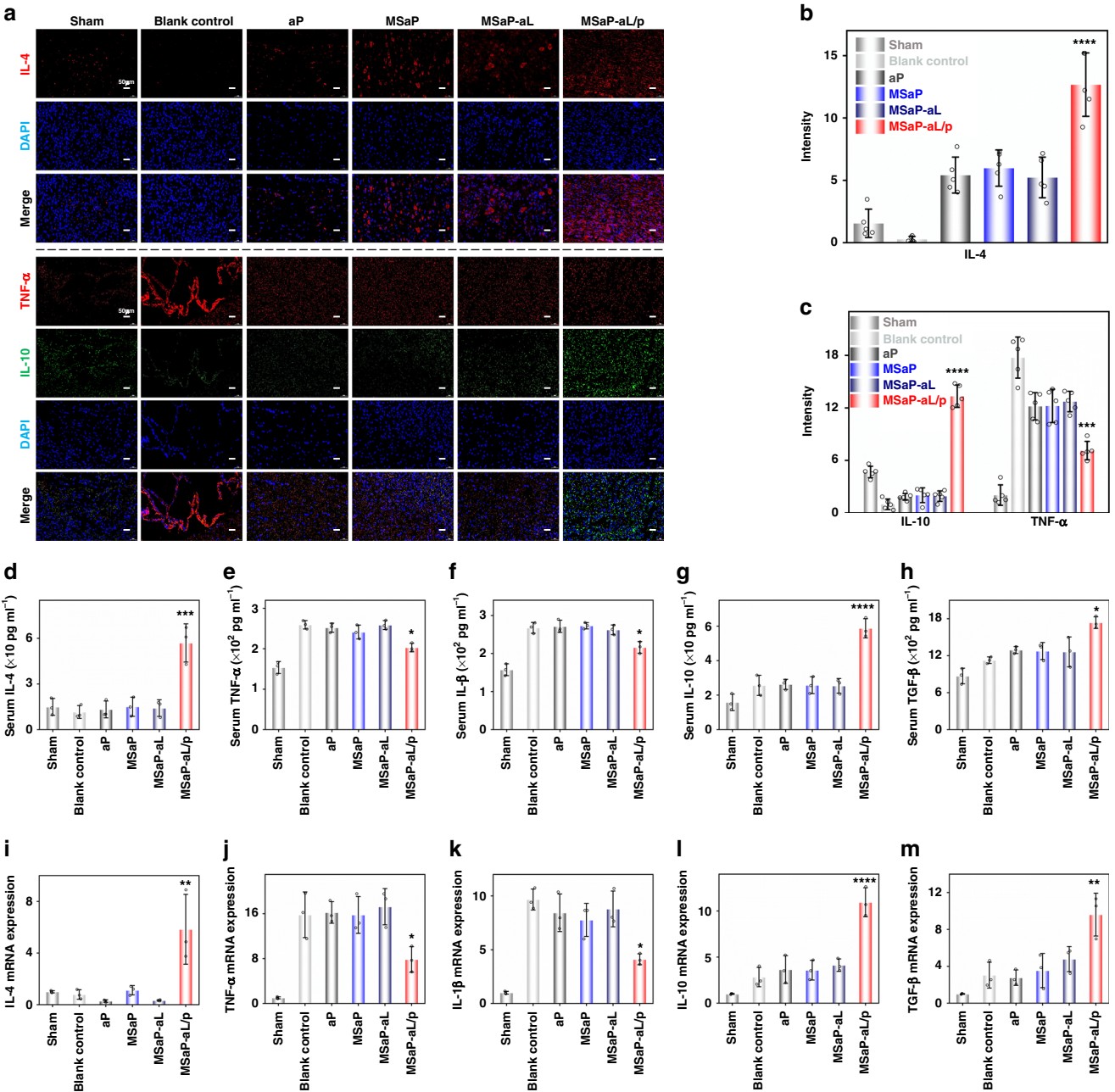

**Fig. 10 Evaluation of immune regulation 7 days after operation. a** IL-4 immunofluorescence single labeling and TNF-α, IL-10 immunofluorescence double labeling (scale bars were 50 μm, 4 independent experiments). Quantitatively analysis of **b** IL-4 immunofluorescence, **c** IL-10 and TNF-α immunofluorescence ($n = 5$, all values were mean ± std. dev., ***$p < 0.001$, ****$p < 0.0001$ when comparing MSaP-aL/p and other groups via two-way analysis of variance (ANOVA) with Tukey's post hoc test). Spinal cord specimens of IL-4 (**d**), TNF-α (**e**), IL-1β (**f**), IL-10 (**g**), and TGF-β (**h**) gene expression levels were assessed by qRT-PCR. Serum levels of IL-4 (**i**), TNF-α (**j**), IL-1β (**k**), IL-10 (**l**), and TGF-β (**m**) were assessed by ELISA ($n = 3$, qRT-PCR and ELISA values were mean ± std. dev., *$p < 0.05$, **$p < 0.01$, ***$p < 0.001$, ****$p < 0.0001$ when comparing MSaP-aL/p and other groups via one-way analysis of variance (ANOVA) with Tukey's post hoc test).

site, the serum IL-4 content was significantly increased in the MSaP-aL/p group compared with the other groups (Fig. 10i). The results of pro-inflammatory and anti-inflammatory cytokines showed that, on the one hand, the serum inflammatory factors TNF-α and IL-1β in the blank control, ap, MSaP, and MSaP-aL groups were significantly higher than those in the MSaP-aL/p group ($p < 0.05$; Fig. 10j, k). On the other hand, the MSaP-aL/p group had higher levels of IL-10 and TGF-β ($p < 0.05$; Fig. 10l, m), which highly suggested that the implantation of immune-regulating fiber bundles can not only improve the local immune environment but also reduce systemic inflammation.

**Histology and immunohistochemical of spinal cord repair.** Despite of the encouraging progress made in in vitro study, in vivo performance of biomaterials for SCI was still compromised by multiple cascade response factors. The content of NGF in spinal cord tissue of each group was detected at the pre-determined time point after operation (Supplementary Fig. 11). NGF in the SCI group (blank control, aP, MSaP, MSaP-aL, MSaP-aL/p) increased in varying degrees, indicating certain degrees of self-repair activities after SCI. Meanwhile, the concentration of NGF in the MSaP, MSaP-aL, and MSaP-aL/p groups were significantly higher than that in the aP and blank control

groups on the first day after operation indicating that loaded cytokines have been partially released from the microsol fiber groups. At each time point after day 6, the content of NGF in each group generally showed a downward trend, but the concentration of NGF in the MSaP-aL/p group was significantly higher than that in the other groups, and on the 21st day, MSaP-aL/p group exhibited a concentration twice as much as that in the other microsol fiber groups, indicating the process of endogenous NGF consumption in local tissue to maintain the homeostasis of surviving nerve tissue after SCI. Such difference could further indicate that the regulation of inflammatory response in acute SCI is beneficial to inhibit local tissue necroptosis[40] and reduce the leakage of cell content enzymes to interfere and decompose endogenous and exogenous NGF[41].

Inflammation-triggered secondary injury followed by chronic wrapping of glial tissue makes it difficult for regenerative neurons and axons to penetrate and reconstruct the damaged tissue, leaving a dilemma to the commonly reported neurogenic materials[1]. In the current scenario, immunoregulation and neurogenic factors were introduced simultaneously for the suppression of glial tissue formation and promotion of neuroregeneration. As indicated in H&E staining images (Supplementary Fig. 12a), after the implantation of fiber bundles, all the material groups exhibited significantly smaller spinal cord cavity area when compared with the control group (Supplementary Fig. 12b, c). However, dense glial tissue could be found around fiber bundles from the aP, MSaP, and MSaP-aL groups as indicated by a large number of parallel red-stained collagen fibers and the slender fibrous cell nucleus. On the other hand, due to the responsive release of IL-4 plasmid and the resulting immunoregulation effect, formation of glial tissues around MSaP-aL/p fiber bundle was significantly suppressed at both 4 and 8 weeks after implantation.

The secretion of inflammatory factors such as IL-1α and C1q by activated M1 microglia after SCI can induce astrocytes to enter the mature A1 state, produce toxicity, and participate in scar tissue formation[42]. In addition, acute inflammatory storm causes glial cells to deposit chondroitin sulfate proteoglycan[43], resulting in the formation of glial scar tissue in the process of secondary SCI to hinder nerve repair. In order to further study the effect of biomimetic immunoregulatory bundles on anti-scar formation, we employed antibody against GFAP and NG2 to label the activated astrocytes and glial scar, respectively (Supplementary Fig. 13a, c). Both the activated astrocytes and glial scar tissues of aP, MSaP, MSaP-aL, and MSaP-aL/p bundles at 4 and 8 weeks were significantly less than those in the blank control group; it was inferred that the oriented electrospun fibers structure simulated the morphological distribution of nerve tissue, which could reduce the formation of central traumatic neuroma after nerve injury. In addition, the fluorescence of semi-quantitative analysis showed (Supplementary Fig. 13b, d) that the MSaP-aL/p bundle produced significantly lower signal intensity than blank control group and three other negative control groups. It was confirmed that the biomimetic bundle has a superior ability to regulate the local immune microenvironment during acute SCI and prolong the effect on reducing the formation of scar tissue in the stage of secondary SCI, paving the way for endogenous neural progenitor cells to migrate to the injured site and differentiate into nerve tissue.

After SCI, endogenous stem cells have the potential to migrate to the injured site and differentiate into nerves. However, due to the occurrence of secondary SCI, the formation of scar tissue hinders cell migration. The distribution of nestin-labeled stem cells at the injured site (Supplementary Fig. 14a) was observed in each bundle group at 4 and 8 weeks after injury. The neural progenitor cells in the sham group was lower than the other

groups, which may be attributed to the low proliferation activity of spinal ependymal cells and the lack of immune response to nestin protein[44]. The corresponding fluorescence quantitative study displayed that, at the same time point (Supplementary Fig. 14b), the MSaP-aL/p group could bring significantly more nestin-labeled stem cells than the other control groups. The results indicated that biomimetic bundle could significantly inhibit the blocking effect of surrounding scar tissue on endogenous stem cells through immunomodulatory effect in acute phase.

Tuj-1 and GAP-43 were used to label neural progenitor cells, neuron cells, and axonal sprouting, respectively, to evaluate the ability of biomimetic bundles to promote nerve regeneration in the middle and late stage of SCI. At 4- and 8-week time points, endogenous stem cells were continuously promoted by NGF released from microsol electrospun fibers (MSaP, MSaP-aL, and MSaP-aL/p), and a large number of Tuj-1-labeled neurons appeared (Fig. 11a). It was significantly better than that in the blank control and aP groups (Fig. 11b). Meanwhile, the fluorescence intensity of each microsol electrospinning group at 8 weeks was significantly higher than that at 4 weeks, respectively. The result further proved that the biomimetic fiber bundle has the function of continuously promoting nerve regeneration. In the MSaP-aL/p group, due to the effect of responsive immunomodulation, the toxicity of astrocytes and the formation of scar tissues were significantly reduced, so at two time points, the amounts of neurons were significantly larger than that in the other control groups. The above results confirmed that the biomimetic bundle not only had the ability of immunomodulation in the area of SCI but also consistent with that reported in the literature that the core–shell structure of microsol electrospun fibers has the ability to protect NGF from the adverse biological environment, which could continuously provide differentiation information for nerve regeneration[13]. Under normal physiological conditions, low expression of GAP-43 only maintains normal function of the spinal cord, but GAP-43 is highly expressed in the process of nerve development and regeneration, which can regulate axonal extension, enhance nerve plasticity, and release neurotransmitters (Fig. 11c). Therefore, the fluorescence density of the sham group was lower than that of the other groups at the same time (Fig. 11d). Meanwhile, the fluorescence density of GAP-43 in the MSaP-aL/p group was significantly higher than that in the other control groups, which proved that immunoregulation of local inflammatory response during acute SCI could promote nerve regeneration for a long time. In addition, the microsol groups (MSaP, MSaP-aL, MSaP-aL/p) were significantly better than other groups at the same time point, suggesting that continuous supply of NGF could significantly promote nerve repair. Surprisingly, with the development of time, the expression of GAP-43 in all the experimental groups except the sham group at 4 weeks were significantly higher than those at 8 weeks. The results suggested that there is a limited time window for the repair of SCI. Immune regulation to reduce inflammatory injury in the stage of acute SCI can better promote nerve regeneration.

The ability of biological bundles to promote angiogenesis is an important guarantee for the long-term survival and repair of injured tissues. Previous studies have shown that the formation of neovascularization in the area of SCI can accelerate nerve regeneration[45]. In the quantitative results of fluorescence staining at 4 and 8 weeks after injury, there were a large number of CD31-labeled vascular endogenous cells in the MSaP-aL/p group, which were more than those in the other control groups (Supplementary Fig. 15a, b). In addition, the number of vascular endogenous cells labeled by CD31 at 8 weeks in the MSaP-aL/p group was more than that at 4 weeks. Coincidentally, the quantitative analysis of neovascularization labeled by von Willebrand factor (VWF)

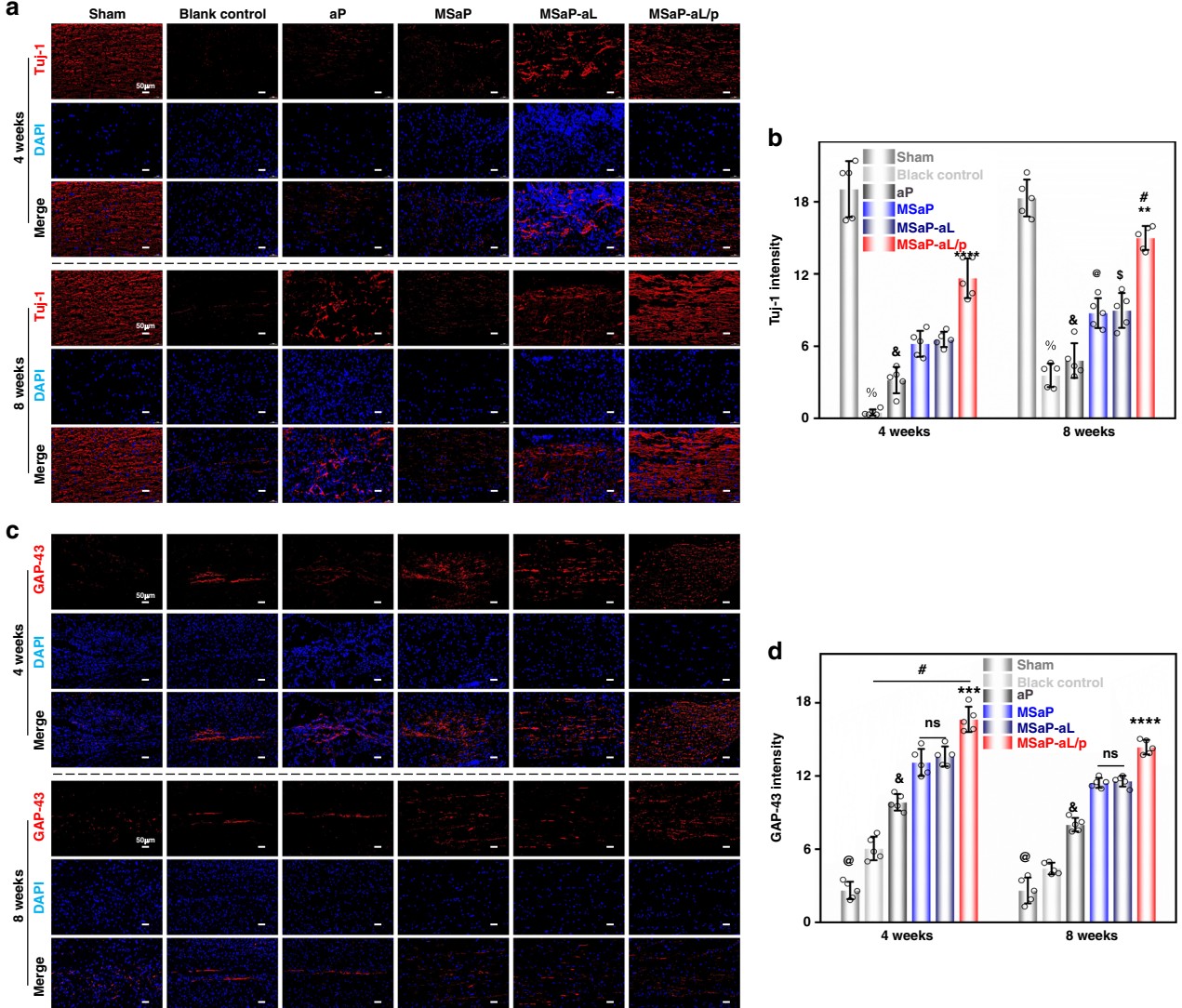

**Fig. 11 Immunofluorescence staining of neuron cells and axons. a** Immunofluorescence staining of neuron cells (scale bars are 50 μm, 3 independent experiments) and **b** quantification analyzed with optical density ($n = 5$, Tuj-1 intensity values were mean ± std. dev., $^{\%}p < 0.0001$ when comparing blank control and microsol electrospun groups (MSaP, MSaP-aL, and MSaP-aL/p); $^{\&}p < 0.05$ when comparing aP and microsol electrospun groups (MSaP, MSaP-aL, and MSaP-aL/p); $^{@}p < 0.01$ when comparing 4w and 8w in MSaP; $^{\$}p < 0.01$ when comparing 4w and 8w in MSaP-aL and $^{\#}p < 0.01$ when comparing 4w and 8w in MSaP-aL/p; $^{**}p < 0.01$, $^{****}p < 0.0001$ when comparing MSaP-aL/p and other control groups at the same time point via two-way analysis of variance (ANOVA) with Tukey's post hoc test). **c** Immunofluorescence staining of axon (scale bars were 50 μm, 3 independent experiments) with **d** its quantification analyzed with optical density ($n = 5$, GAP-43 intensity values were mean ± std. dev., $^{\#}p < 0.05$ when comparing 4w and 8w in blank control, aP, MSaP, MSaP-aL, and MSaP-aL/p, respectively; $^{@}p < 0.01$ when comparing sham group and other groups, $^{\&}p < 0.0001$ when comparing aP and microsol electrospun groups (MSaP, MSaP-aL, and MSaP-aL/p), and $^{***}p < 0.001$, $^{****}p < 0.0001$ when comparing MSaP-aL/p and other control groups at the same time point via two-way analysis of variance (ANOVA) with Tukey's post hoc test).

fluorescence showed that the intensity in the MSaP-aL/p group was significantly higher than that in the other control groups at the same time (Supplementary Fig. 15c, d). Similarly, the number of neovascularization labeled by VWF at 4 weeks was significantly less than that at 8 weeks, which indicated that biomimetic bundle could not only reduce the invasion of local inflammatory reaction but also induce macrophages and microglia cells to be polarized to M2 subtype, which could secrete vascular endothelial growth factor and promote blood vessel formation, providing a superior platform for tissue regeneration for a long time[46].

In this study, a microenvironment-responded fiber scaffold was constructed through grafting the immune regulatory carrier, aL, onto NGF-loaded microsol fibers through acid-sensitive chemical bond Schiff base. The responsive fiber scaffold was endowed with the capability of local immune regulation, which

could respond to the acid microenvironment after the acute SCI, as well as the sustained promoting effect on neural tissue regeneration. As indicated by in vitro studies, MSaP-aL/p had played the pivotal role in regulating the M2-type polarization of BMMs and promoting the neural differentiation of mesenchymal stem cells. Correspondingly, significantly reduced inflammatory response, downregulated glial fiber acidic protein secretion, reduced scar tissue formation, promoted angiogenesis, improved neurogenic bioability, and better motor function recovery could be verified through in vivo studies with the help of MSaP-aL/p scaffold. In conclusion, the microenvironment-responsive fiber scaffold constructed in this study had provided a strategy for the repair of acute SCI, which relied on the coordination between immune-microenvironment regulation and neurogenic biomaterials.

## Methods

**Preparation of aLs**. aLs were prepared by reverse evaporation. Specifically, 160 mg Lecithin (Yuanye, Shanghai, China), 40 mg Cholesterol (Arcos, Belgium), and 4 mg DSPE-PEG-CHO (Aladdin, Shanghai, China) were dissolved in 5 ml trichloromethane (Lingfeng, Shanghai, China) and mixed with 1 ml trichloroethane dissolved with 5 mg Octadecyl amine (Aladdin, Shanghai, China). The resulting solution was further mixed by ultrasound to obtain uniform emulsion, after which the organic solvent was removed in rotary evaporator to obtain colloidal product. Final hydration treatment was carried out to prepare liposome emulsion. The aL was obtained by ultrasound and filtration using 450 and 220 nm polycarbonate membrane (Millex-GP, Ireland).

**Liposome loading IL-4 plasmid**. In all, 100 μl Opti-MEM (Thermo Fisher Scientific, USA) was respectively added to five 1-ml centrifuge tubes marked as A, B, C, D, and E. The IL-4 plasmid DNA (pDNA) (Taitool, Shanghai, China; Supplementary Fig. 1a) with eGFP and aL were added into the tubes at the doses indicated in Supplementary Table 2. After slight shaking, kept stationary at room temperature for 30 min, mixed solution with pDNA/aLiposome ratios of 1:1, 1:1.5, 1:2, 1:2.5, and 1:3 could be obtained in these five tubes.

**Characterization of aLs**. The surface morphology of aLs was observed by TEM (HT7700, Hitachi, Japan) at the voltage of 120 kV. Before detection, 2 μl of aLs were dripped on copper net and dried at room temperature for 4 h. The stability of liposomes was detected by the Phospholipid Detection Kit (Sigma-Aldrich, MAK122, USA) after being preserved for 1, 4, 9, 14, 19, and 24 h, respectively. The size, PDI, and potentiodynamic potential of the mixture were measured by DLS particle size analyzer (Zeta sizer, Malvern, Nano-ZS90, UK). In order to evaluate the encapsulation efficiency (EE%) of different aL/p proportion, 1-ml samples of each group were added to the ultrafiltration centrifuge tube (Mw = 500 kDa, Millipore, USA) and centrifuged for 15 min at the speed of $2795 \times g$. Same procedure was repeated for three times after adding 500 μl deionized water in the tube each time. The pDNA content was measured in the collected liquid using the dsDNA Kit (Invitrogen, USA) to quantify the encapsulation efficiency, which could be calculated according to the following formula (Eq. 1):

$$EE\% = \frac{\text{Initial content of pDNA} - \text{unloaded pDNA}}{\text{Initial content of pDNA}}. \tag{1}$$

**Screening of pDNA/aLiposome transfection**. SD rat BMSCs were implanted in 24-well plates. After reaching the confluent rate of around 70%, cells were washed with PBS for three times, followed by the addition of 100 μl pDNA/aLiposome mixture and 400 μl per well Opti-MEM in each well. After culturing for 6 h, the transfection medium was replaced with 500 μl per well normal serum containing culture medium (10% fetal bovine serum). After culturing for 72 h, the cells were washed for 3 times with PBS, and the living nucleus was stained with 200 μl Hoechst33342 (Beyotime, Shanghai, China) staining solution at 37 °C for 30 min. After washing, the fluorescent images of eGFP were observed under fluorescence microscope (Zeiss Axiovert 200). The fluorescence co-localization repetition rate was analyzed by the ImageJ 1.8.0 (USA) software, and the transfection efficiency was evaluated. At the same time, the culture medium was collected on the first, third, fifth, and seventh day of BMSC transfection and centrifuged at 4 °C for 15 min. The supernatant was collected and stored at −80 °C. The secretion of IL-4 after BMSC transfection was evaluated by the IL-4 ELISA Kit (Abcam, ab100770). The pDNA/aLiposome with the highest transfection efficiency was chosen for further study.

**Preparation of oriented electrospun fiber scaffolds**. Aminated PLA electrospinning solution was prepared by dissolving 0.5 g aPLA (Mw = 100 kDa, Mw/Mn = 2.06, Ruixi, Xi'an, China) in 4 g DCM (Sinopharm, Beijing, China), and 2 g N, N-dimethylformamide (DMF, Lingfeng, Shanghai, China) to obtain uniform and stable solution under magnetic stirring at room temperature. The preparation method of PLA (Mw = 100 kDa, Mw/Mn = 1.61, Daigang, Jinan, China) electrospinning solution is the same as above. The preparation of microsol electrospinning started with the preparation of sodium hyaluronate hydrosol (1 wt%) (HA, Mw = 0.5 MDa, Yuancheng, Wuhan, China), which was obtained by dissolving 0.1 g HA in 9.9 g deionized water and rotating to complete dissolution at room temperature[47]. Rat β-NGF (R&D System, USA) was resuspended in 0.1 wt% bovine serum albumin (BSA; Solarbio, Beijing, China) solution to achieve the final concentration of 100 μg ml⁻¹. The uniform HA-β-NGF hydrosol was obtained by mixing 10 μl resuspended β-NGF with 50 μl HA solution, after which 0.01 g Span-80 (Sigma, USA) and 4 g DCM were added into the mixture and stirred at high speed for 30 min at room temperature to obtain a homogeneous and stable water-in-oil emulsion containing β-NGF microsol particles. The microsol (MS) spinning solution was obtained eventually by adding 0.5 g aPLA and 2 g DMF into the emulsion.

**Preparation of different oriented electrospinning fibers**. As for the preparation of different oriented fiber scaffolds, the aforementioned electrospinning solutions were loaded in a 10-ml syringe with a length of 10 cm and an inner needle diameter

of 0.9 mm. The electrospinning process was conducted with a propulsion pump speed of 70 μl min⁻¹, at a voltage of 15–18 kV, and a distance between needle tip and the parallel electrode receiver of 15 cm. The parallel oriented fiber scaffolds can be collected between the electrode rods. In order to remove the residual solvent from the obtained fibers, all prepared fiber scaffolds were dried under vacuum overnight. Traditional electrospun directional fiber (PLA, aP) and microsol electrospun oriented fibers (MSaP) were also obtained. In the following studies, the oriented fiber membrane was used in in vitro study, while the fiber bundle was used in in vivo research.

**Preparation of responsive nerve fiber scaffold**. The responsive nerve fiber scaffold was prepared by grafting of pDNA-loaded liposomes on the surface of the electrospun scaffold[48]. In short, the microsol electrospun fiber scaffold (MSaP) was immersed in the 5 ml screening liposome mixed solution (pDNA/aLiposome = 1:2.5) and placed in an oven at 37 °C for at least 24 h. MSaP-aL/p was prepared by washing the obtained scaffolds with deionized water for three times.

**Sequential release study**. In order to investigate the release of liposome under different pH, 71.6 g disodium hydrogen phosphate (Na₂HPO₄) and 31.2 g sodium dihydrogen phosphate (NaH₂PO₄) were dissolved in 1000 ml deionized water, respectively, and configured to obtain 0.2 M Na₂HPO₄ and NaH₂PO₄ mother liquor, and PBS solutions could be obtained via mixing the mother liquor according to Supplementary Table 3.

MSaP-aL/p was immersed in 50 ml centrifuge tubes containing 10 ml pH 7.4, pH 6.6, and pH 5.8 PBS, respectively. All centrifugal tubes were placed at 37 °C, on the 120 cycles min⁻¹ constant temperature vibrator (Thermo, USA). After 3, 6, 9, 12, 15, 18, 21, and 24 h of vibration, the resulting solution was collected and stored at −20 °C with 10 ml fresh PBS re-added into the tubes. The effects of different environment pH on Schiff base breakage and liposome release was studied using the Phospholipid Assay Kit (Sigma-Aldrich, MAK122, USA). Similar to the above methods, the effects of different acidic environments on the release of NGF from microsol electrospun fiber scaffolds were also analyzed using the NGF ELISA Kit (R&D Systems, USA). The cumulative release curves were drawn, respectively.

**Preparation for cell research**. Before characterization of cellular performance, the fiber membranes collected on the 14 mm diameter and 100-μm-thick glass slides were gently placed at the bottom of the Transwell plate and sterilized by irradiation, and then the aLs loading pDNA were grafted according to the aforementioned method and incubated at 37 °C. BMSCs (Institute of Orthopaedics of the first affiliated Hospital of Soochow University) from SD rats with a density of $1 \times 10^4$ per well were implanted on fiber membranes, and BMMs (Procell, China) from SD rats with a density of $5 \times 10^3$ per well were seeded in the upper chamber. The Transwell plate was incubated at 37 °C with 95% relative humidity and 5% CO₂ partial pressure. The culture medium was changed every 2 days. Samples without membrane were used as blank control; the traditional fiber membranes (aP), the microsol fiber membranes (MSaP), and the blank liposome grafted microsol fiber membranes (MSaP-aL) were used as negative controls.

**Characterization of immune regulation**. The concentrated hydrochloric acid (HCL) and sodium hydroxide (NaOH) were added in α-MEM medium, with pH accurately adjusted to 5.8, 6.6, and 7.4 by pH tester (EZDO, Taiwan, China) to simulate the inflammatory acidic microenvironment of SCI. In order to evaluate the immunomodulatory performance of responsive fiber membranes, the expression of BMM gene was analyzed by qRT-PCR on the first, third, fifth, and seventh day of co-culture. According to the literature, the expression levels of pro-inflammatory factors IL-1β and TNF-α mRNA and anti-inflammatory factors IL-10 and TGF-β mRNA were evaluated after polarization of BMM in different acidic environments[49]. PCR amplification primers were designed by Genewiz and are shown in Supplementary Table 4. Glyceraldehyde-3-phosphate dehydrogenase was used as an internal reference gene. The PCR experimental data in the study were collected by QuantStudio 7 Flex Real-Time PCR system.

**Transplantation of fiber bundles**. A total of 330 female SD rats (around 200 g) were purchased from JOINN Laboratories (Soochow, China) and were randomly divided into six groups. Animal management and surgery were carried out according to the plan approved by the Ethics Committee of the First Affiliated Hospital of Soochow University.

The spinal cord hemi-section model was established[50]. Briefly, the rats were anesthetized by intraperitoneal injection of 2% Pentobarbital (50 mg kg⁻¹). After anesthesia, a 3-cm longitudinal incision centered on T9 was made on the back of the rat, and the paraspinal muscles were separated. After removal, the T9 lamina was fully exposed to the spinal canal; under the accurate measurement of the scale, the right half of the spinal cord was cut off to create a 3-mm spinal cord hemi-section defect. Followed by saline irrigation, the fiber bundles (2 mm × 2 mm × 3 mm) were placed inside the defect (Fig. 8a (2), (3), (4)). After sewing the fascia and skin layer by layer with sutures, each rat was intramuscular injected with $2 \times 10^5$ units of antibiotics every day for 5 days and manually emptied the bladder every 12 h. The bare laminectomy was conducted in the sham operation group, and spinal cord hemi-section without the implantation of bundle was introduced as the

blank control group. Models implanted with aP, MSaP, and MSaP-aL were used as the negative control group.

**Local immune inflammatory response to SCI**. In order to evaluate the local immunomodulatory effect of nerve bundles in response to SCI microenvironment, 15 rats were randomly selected from each group at 1 week after operation (Fig. 8a (6)).

**Identification of microglia/macrophage phenotype**. After euthanasia, the thoracic cavity was cut open with the left ventricle and inferior vena cava fully exposed. In all, 100 ml 0.9% saline was perfused into the heart. Spinal cord samples were used for flow cytometric detection[51]. A 5-mm section of the injured spinal cord tissue centered at the epicenter of the injury site was harvested and was immediately cut into small fragments and mechanically separated by a 100-μm cell strainer. The cell suspensions of different groups were centrifuged at $300 \times g$ for 10 min at 4 °C and then the antibodies were added and incubated at 4 °C for 30 min and fixed in 1% paraformaldehyde. The following rat-conjugated antibodies were used: mouse anti-CD11b-APC (BD Pharmingen™, 562102, 1:20), mouse anti-CD86-FITC (BD Pharmingen™, 561961, 1:20), and mouse anti-CD206-PE (Santa Cruz, sc-58986, 1:50). Cells were analyzed on a flow cytometer (Merck Millipore, USA), and the results were analyzed by the FlowJo 7.6 software. For analysis, cells were first gated for CD11b to ensure that only myeloid cells were selected, then the following combination of specific markers were used to identify M1 (CD86) and M2 (CD206). In the assay, three samples were randomly selected from each group for detection ($n = 3$).

**Evaluation of animal motor function**. The recovery of neuromotor function of hindlimb in the six groups of rats was evaluated by open field test and scored by BBB and IPT scoring system. The score range of BBB scale was 0 to 21, with 0 indicating no hindlimb movement and 21 indicating that motor function was normal. During each test, two unwitting examiners evaluated separately each week after the operation. As a supplementary study of BBB scoring test, IPT can improve the effectiveness and sensitivity of scoring. Using Rivlin method, the rats were placed on the inclined plate with rubber pad, the longitudinal axis of the rat was kept parallel to the longitudinal axis of the inclined plate, the head was raised to one side, and raised 5° from 0° within 5 s. The highest angle of stay of the rats was recorded 5 times and the larger the angle indicated the stronger the load-bearing capacity of the lower extremities. The above two scoring methods were performed by two blinded observers and seven rats were randomly selected from each group at a fixed time point every week for evaluation ($n = 7$).

**Histological analysis of spinal cord**. Thirty rats in each group were randomly divided into 4- and 8-week groups and were euthanized at the pre-designed time. According to the aforementioned method, the 5-μm slice along the long axis of spinal cord was stained by the H&E Staining Kit, observed, and photographed under a bright-field microscope. The spinal cord cavity area of each group was measured by the ImageJ software, and three fields were randomly selected for statistical analysis ($n = 3$). As for the immunohistochemical evaluation, the antigen was repaired with 0.3% hydrogen peroxide and nonspecific antigen was blocked by 5% BSA. The primary antibody was added and incubated at 4 °C overnight, and the secondary antibody was incubated at room temperature for 1 h after PBS washing. The samples were observed and photographed under fluorescence microscope, with the fluorescence intensity analyzed with the ImageJ software. The values are represented in the form of average optical intensity, which had been corrected for optical density, de-background, and normalized by the number of cell nuclei. A total of five images were randomly selected from each group for statistical analysis ($n = 5$). The primary antibody included rabbit anti-Tuj-1(Abcam, ab18207, 1:1000) for neuron staining, rabbit anti-GAP-43 (Abcam, ab75810, 1:800) for axon staining, goat anti-nestin (R&D Systems, AF2736, 1:1000) for neural progenitor cell staining, mouse anti-GFAP (Service bio, GB12096, 1:800) for staining of astrocytes, mouse anti-NG2 (Abcam, ab50009, 1:250) for staining of glial scars, mouse anti-CD31 (Service bio, GB12063, 1:300) for staining of vascular endothelial cells, and rabbit anti-VWF (Abcam, ab6994, 1:600) for staining of neovascularization. Second antibody includes goat anti-rabbit (Service bio, GB21303, 1:300), goat anti-mouse (Service bio, GB25301, 1:400), and goat anti-rabbit (Abcam, ab150079, 1:500).

**Statistical analysis**. The data were presented in the form of means ± standard deviations. The statistical analysis (Origin 9.1 or GraphPad Prism 7.0 software) was calculated by one- or two-way analysis of variance by Tukey's multiple comparison test to evaluate the differences among the groups unless otherwise stated. The probability value ($p$) <0.05 was considered to be statistically significant.

**Reporting summary**. Further information on research design is available in the Nature Research Reporting Summary linked to this article.

## Data availability

All the data supporting the findings are available within the manuscript and supplementary information. The sequence of PCR primers is located within the paper and supplementary information. A reporting summary is available as Supplementary Information. Other figures or data supporting the results of this study are available from the corresponding author upon any reasonable request. Source data are provided with this paper.

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

## Acknowledgements

This work was supported by the National Natural Science Foundation of China (81772312, 81972059, 81601891, 81702190, and 51873107), Research and Development of Biomedical Materials and Substitution of Tissue and Organ Repair under the National Key R&D Program (2016YFC1101505), Shanghai Municipal Education Commission−Gaofeng Clinical Medicine Grant Support (20171906), Natural Science Foundation of the Jiangsu Higher Education Institutions of China (15KJB320012), Shanghai Jiao Tong University "Medical and Research" Program (ZH2018ZDA04), Shanghai talent development fund (2018099), Key Talented Man Project of Jiangsu Province (RC2011102), Standardized Diagnosis and Treatment Project of Key Diseases in Jiangsu Province (BE2015641), the Natural Science Foundation of Jiangsu Province (BK20170370), Jiangsu Provincial Special Program of Medical Science (BL2012004), Jiangsu Provincial Clinical Orthopedic Center, and the Priority Academic Program Development of Jiangsu Higher Education Institutions (PAPD).

## Author contributions

K.X. contributed to the conception and design of the work; the acquisition, analysis, and interpretation of data; and drafted and revised the manuscript. Y.G. and J.T. contributed to design of the work; the acquisition, analysis, interpretation of functional data; and revised the manuscript. H.C., Y.X., L.W., and F.C. contributed to the acquisition of partial data. L.D. and H.Y. contributed to revision of the manuscript. Q.S., W.C., and L.C. contributed to the conception and design of the work and drafted and revised the manuscript.

## Competing interests

The authors declare no competing interests.
