## [Peer Review File · Nature Communications]

Reviewers' Comments:

Reviewer #1:

Remarks to the Author:

The manuscript describes the preparation and characterization of aldehyde-modified cationic liposomes loaded with pDNA encoding for interleukin-4 (IL-4) grafted through Schiff base bond at the surface of amino-modified oriented electrospun fibrous scaffolds. These fibrous scaffolds were produced by a so-called microsolv electrospinning, a process previously proposed by the authors, where the core is loaded with nerve growth factor (NGF).

The pH-responsive immunoregulation-assisted neural regeneration strategy seems to be able to release the pDNA under acidic environment, inducing some cell polarization and anti-inflammatory factors (IL-10 and TGF- β) secretion from macrophages and microglia. Indeed, the amino PLA microsolv electrospinning carrying aldehyde cationic liposomes loading IL-4 plasmid scaffold seems to reduce scar tissue formation, resulting in marginally enhanced motor function recovery when implanted at the spinal cord hemisection site. However, the histology results refer only to one animal analysed, are the histological results relative to the best animal? To have a clear perspective of the results, it is important to show results of the best, worst and more frequent result obtained. Without clarifying those results, it is impossible to derive useful conclusions.

The developed fibrous scaffold with dual release capacity represents an attractive concept for the application envisaged. The novelty of the present approach is moderate and appears to be in the combination of the local immune regulation and sustained stimulation of neural tissue regeneration. The experimental design has important shortcomings to conclude about the results achieved, particularly the statistics and the cohort of animals considered in the reported results.

Questions regarding the concept of the implant:

- There is no evidence provided in the manuscript regarding the release of plasmid and of NGF over the time-span of the experiment (4-8 weeks). Thus, it is not clear if the implant is really operating in accordance with the projected performance.
- To be effective, the plasmid needs to be internalised by the target cells. No evidence is provided in the histology of the explants that the cells were effectively transfected by the plasmid. Indeed, the plasmid should have a reporter gene that should facilitate the evaluation of the transfection at the defect site.
- The NGF is present in the nano fibres during the whole procedure for the grafting of the liposomes at the surface. How much of the NGF is preserved and to what extent it is bioactive after being exposed to this procedure?
- The size of the liposomes is quite large to enable an easy internalization. The authors should discuss this important aspect in the manuscript.
- No details are provided regarding the method used to obtain aligned nano fibrous constructs and its details (dimensions, porosity, degree of alignment, ...). Also the dose of plasmid and NGF effectively implanted and released should be provided.

Questions and comments raised by the animal study and its outcomes:

- The spinal cord implant is insufficiently detailed in the manuscript, being very difficult to understand its size, the shape, how it is fixed in the defect site,, how it is ensured a good and reproducible coverage of the hemisection defect created, the alignment of the fibres in the implant, the confinement at the defect site.
- How many researchers were involved in creating the defects? How was the training of those researchers performed? How consistent is the reproducibility of the defects created in the animal model?
- The results reported do not identify the number of animals that were considered for the data reported in the results (both the histology and the functional evaluation of the recovery). Indeed, given the large number of animals tested, if the analysis does not include either all the animals tested or comparable subsets of randomly selected animals, it is very difficult to have a clear

perception of the efficacy of this implant.

- Much more details are needed to ensure that the procedures used to obtain the safety and efficacy profile of the implant are robust and representative of the spectra of outcomes obtained for each condition.

Other issues of the manuscript:

- The authors described similar information on the Introduction and on the Results and Discussion sections. Please avoid duplication of information along the manuscript, by providing relevant background information in the Introduction section.
- The achieved results are poorly compared to the achievements of others already described in the literature.
- Approximately 90% of IL-4 plasmid release is achieved in 24h at pH 5.8 (Figure 4 E). How relevant the action of IL-4 when kept under culture or implanted for longer time periods?
- Why was the effect on the pro-inflammatory and anti-inflammatory expression only observed on the 3rd day of BBM culture (Figure 6 G and J)?
- Since approximately 50% of IL-4 plasmid is released in 24h at pH 6.6 (Figure 4 E), how you explain the few differences regarding IL-10 and TGF- β expression at pH 5.8 and pH 6.6 (Figure 6 G and J vs H and K)?
- The authors claim that the microsol electrospinning allows for a prolonged release of protein molecules (6 weeks) (based on a previous authors' work on chloroquine-based DDS). However, approximately 80% of NGF is released in 25 days (Figure 4 F). Please revise the claims regarding the microsol electrospinning advantages having in consideration the present results. The results should be also presented in terms of the effective amount of protein released over time and also its bioactivity.
- How did the authors acquire the BMSCs? Is the procedure previously approved by the Ethical Committee of the certified institution for animal experimentation? Are those isolated from a single donor or pooled? Did the authors assess BMSCs putative stem cell potential?
- The neurogenic activity of the testing condition was assessed by culturing the BMSCs onto the NGF-loaded fibrous scaffolds for 10 days. The authors used qRT-PCR technique to assess the expression of earlier neuron-specific markers in neuron-like cells. To increase the significance of these results, it is suggested to assess the expression of lately expressing neuronal markers.
- During neuronal differentiation, neurite extension and outgrowth occurs. Therefore, it is important to assess the neurite outgrowth by counting the Neurite-bearing cells.
- From the results of Figure 8 B and C, it is important to denote the minimum time needed to achieve similar motor function. Please discuss it accordingly.
- In Figure 8B, please clarify the significant differences between the conditions MSaP-aL/P and MSaP-aL. They seem very similar, namely when considering the SD at each time point.
- In Figure 8C, no significant differences were clearly observed. Please revise the statistical analysis, namely testing the normality and variance homogeneity.
- In Figure 9 please provide the results for the same time points (i.e. 4 and 8 weeks), as for other in vivo results (i.e. Figures 10 and 11).
- The authors assess the pro-inflammatory (IL-1 β , TNF- α and IL-10) and anti-inflammatory (TGF- β) cytokines expression of BMM in in vitro assays. However, the authors only assessed the pro-inflammatory cytokines TNF- α and IL-10 in vivo. Please be consistent on the presentation of results along the manuscript.
- The Figure 9 quantitative results are based on immunofluorescence images. Following the same presentation approach depicted in Figure 7, please provide mRNA expression results for the in vivo assay.
- For all the immunofluorescence images quantification, it is suggested to normalize the staining intensity by the number of cells nuclei.
- The authors should refer previous works in which liposomes carrying plasmids were immobilised at the surface of electrospun meshes.

Reviewer #2:

Remarks to the Author:

The authors provide a refined strategy to target both immune reaction and neural regeneration in spinal cord injuries. The approach is interesting and rather solid.

I recommend to address the following points to facilitate the comprehension of the paper as well as to strengthen the obtained results.

A lot of the data are described in the manuscript, but I am afraid that 12 figures and 28 pages of manuscript are way too many for a communication. The authors should condense data presentation and panels: some of less significant data could be moved to the supplementary material section.

For a better understanding of the impact of their work the authors should state clearly that the present work deals with acute spinal cord injuries.

Line 568: there is actually no trustable specific marker for NSC in vivo, Nestin can provide hints of PROGENITOR cells, but not neural stem cells. Tuj-1 marks immature neurons. Therefore I would remove any statement regarding endogenous stem cell mobilization and I will focus on nervous regeneration with newly formed nervous fibers.

Line 595: I would recommend to add stainings against GAP-43 and SMI31 to better strengthen the findings about the neurogenic potential of the scaffold.

Line 601: I would add stainings against VWF to strengthen this result about neo-angiogenesis too.

Figure 8: Behavioral recovery data are significant but of limited magnitude if compared to control groups.

Figure 11: I recommend to substitute "NSC" with neural progenitor cells

Reviewer #3:

Remarks to the Author:

The manuscript by Xi et al. entitled " Microenvironment-Responsive Immunoregulatory Electrospun Fibers for Promoting Nerve Regeneration " presents results suggesting that treatment with a pH-responsive strategy based on the grafting of interleukin-4 (IL-4) plasmid-loaded and aldehydes-modified cationic liposome onto the nerve growth factor (NGF)-loaded electrospun fibrous scaffolds leads to immunomodulation and improves functional recovery and tissue repair after spinal cord injury.

Points

1. The authors do not provide any clear evidence demonstrating that this strategy modulates inflammation. This is of key importance since their rationality to boost IL-4 levels after spinal cord injury was to modulate the inflammatory response. Were macrophage and microglia numbers changed by the treatment? Did the treatment shift microglia and macrophage polarization towards an M2-like phenotype in vivo?. These are key questions that the authors should have addressed. In additions, this must be assessed by using flow cytometry. This is the only methodology that allows to distinguish microglia from macrophages in vivo, since some new specific markers for microglia, such as TMEM119 or P2Y12, only discriminate microglia from macrophages in physiological but not in pathological conditions. The authors just showed that the treatment modulated cytokine levels in the spinal cord by using immunofluorescence. However, this is not a reliable technique to assess cytokine levels since these antibodies bind to various unspecific proteins. For these reason, cytokine levels have to be assessed by ELISA (not WB) or Luminex assays. The authors need also to demonstrate that IL-4 protein levels is really increased after the treatment, since this therapeutic approach is based, in part, in increasing the levels of this immunomodulatory cytokine. Again, this needs to be done by ELISA or Luminex assays.

2. The title of the manuscript includes the words "Promoting Nerve Regeneration". This is quite surprising since the author do not demonstrate that their experimental approach leads to axon

regeneration. This is very important and needs to be assessed by the authors by means of tracer, viral plasmids or transgenic animals.

3. The authors also applied NGF to stimulate mobilization and differentiation of stem cells into neurons. According to their data, the authors stated that this novel approach increased the stem cell counts and differentiation into neurons. However, this is based on immunohistochemistry against Nestin and Tuj1. These experiments are not appropriated to demonstrate or even to suggest their statements.

4. The authors reveal that this novel approach improved functional outcomes based on the BBB and IPT. It is quite surprising to observe statistical difference between groups with such small difference. It is also surprising to observe this tiny standard deviation after hemisection injury. Highlight that the statistics used are not appropriate since there are two factors (experimental group and time). Two-way RM ANOVA needs to be used.

5. English should be revised throughout the text.

6. The n should be included in all the experiments

Reviewers' comments:

The black text was copied from the editors' letter, and the blue indentation is our response.

Reviewer #1 (Remarks to the Author):

The manuscript describes the preparation and characterization of aldehyde-modified cationic liposomes loaded with pDNA encoding for interleukin-4 (IL-4) grafted through Schiff base bond at the surface of amino-modified oriented electrospun fibrous scaffolds. These fibrous scaffolds were produced by a so-called microsol electrospinning, a process previously proposed by the authors, where the core is loaded with nerve growth factor (NGF).

Thank you for your patient and thoughtful reading as well as the constructive comments and advices about our manuscript. We have completely revised the original manuscript based on your comments and suggestions. Detailed point-by-point responses are provided below.

The pH-responsive immunoregulation-assisted neural regeneration strategy seems to be able to release the pDNA under acidic environment, inducing some cell polarization and anti-inflammatory factors (IL-10 and TGF- β) secretion from macrophages and microglia. Indeed, the amino PLA microsol electrospinning carrying aldehyde cationic liposomes loading IL-4 plasmid scaffold seems to reduce scar tissue formation, resulting in marginally enhanced motor function recovery when implanted at the spinal cord hemisection site. However, the histology results refer only to one animal analysed, are the histological results relative to the best animal? To have a clear perspective of the results, it is important to show results of the best, worst and more frequent result obtained. Without clarifying those results, it is impossible to derive useful conclusions.

Re : Thank you for the comment and suggestion.

(1). We consider three aspects when choosing animal: (I). Similar biological characteristics to humans; (II). Animal ethical requirements and prices. According to previous research results , Animal ethical requirements and prices. According to previous research results, rats and humans share high similarity on biological characteristics, immune response, behavioral manifestations, and electrophysiological parameters, as well as the process of axon regeneration and structural remodeling in the central nervous system ¹. In addition, SD rats are relatively inexpensive and we hold a license for experimental animal ethics research. (III). Meanwhile, we reviewed the literature of central nervous system regeneration in nerve tissue engineering which showed that the actual operation process of

rats is effective and the experimental results are feasible and reproducible ². After modeling *in vivo*, the loss of ipsilateral lower extremity motor function can be observed after surgery which confirmed the feasibility of SD rats for constructing spinal cord injury model (Figure 7A(e)).

(2). We selected representative images in the manuscript to show the results of each animal assay, and obtained the statistical analysis results by random sampling with several samples. Moreover, in order to show the results of the experiments more clearly, we have added the sample size (n) for each experiment to all the results descriptions and improved the description of the highlighted results in the manuscript. In addition, scatter points were added to all column charts to represent the distribution of all data results.

The developed fibrous scaffold with dual release capacity represents an attractive concept for the application envisaged. The novelty of the present approach is moderate and appears to be in the combination of the local immune regulation and sustained stimulation of neural tissue regeneration. The experimental design has important shortcomings to conclude about the results achieved, particularly the statistics and the cohort of animals considered in the reported results.

Re : Thank you for the comment. Following revision was conducted according to your advice.

(1). We have reconducted the statistical analysis of the data in the manuscript through GraphPad Prism 7.0 software. The specific measures are as follows: one-way ANOVA followed by Tukey's multiple comparison test for single variable data analysis. Two-way ANOVA followed by Tukey's multiple comparison test when analyzing two variable data. In addition, the sample size (n) is added to each result description as mentioned in the above question. We have corrected the statistical analysis part of the article as follows:

The data was presented in the form of means ± standard deviations. The statistical analysis (Origin 9.1or GraphPad Prism 7.0 software) was calculated by one-way or two-way analysis of variance by Tukey's multiple comparison test to evaluate the differences among the groups unless otherwise stated. The probability value (p) less than 0.05 was considered to be statistically significant (Supplementary information, page 17, line 446).

(2). The original manuscript in the animal experiment section on the cohort grouping was not appropriate, the correction was restated as follows:

a. *A total of 270 female Sprague-Dawley (SD) rats (around 200g) were purchased from JOINN Laboratories (Soochow, China). The animals were randomly divided into six groups with 45 rats in each group (Supplementary information, page 13, line 325).*

b. **6.3 Evaluation of the degree of local immune inflammatory response to spinal cord injury.** *In order to evaluate the local immunomodulatory effect of nerve bundles in response to spinal cord injury microenvironment, 15 rats were randomly selected from*

each group at 1 week after operation (Fig7. A(f)) (Supplementary information, page 14, line 346).

6.3.1 Identification of microglia/macrophages phenotype. In the assay, three samples were randomly selected from each group for detection (n=3) (Supplementary information, page 14, line 364).

6.3.2 Fluorescently labeled immune factors. Photographing by fluorescence microscope, and semi-quantitative analysis of five randomly selected images which had been corrected for optical density, de-background, and finally normalized the staining intensity by the number of cell nuclei using ImageJ software (Supplementary information, page 15, line 377).

6.3.3 Assessment of inflammatory genes in injured local tissues. To evaluate the transfection performance and immunoregulatory function after fiber bundles implantation, three randomly selected spinal cord specimens were dehydrated in ethanol followed by xylene (n=3) (Supplementary information, page 15, line 382).

6.3.4 Quantitative serum immune factors. Briefly, three animals were randomly selected from each group, and the blood was collected with a centrifuge tube containing coagulant during cardiac perfusion (n=3) (Supplementary information, page 16, line 398).

c. **Evaluation of animal motor function.** Seven rats were randomly selected from each group at a fixed time point every week for BBB and IPT evaluation (n=7) (Supplementary information, page 16, line 417).

d. **Histological analysis of spinal cord.** Thirty rats in each group were randomly divided into 4w group and 8w group, and were euthanized at pre-designed time (Supplementary information, page 17, line 421).

H&E staining evaluation: Different fiber bundles were cut to the size of 10mm×10mm×0.1mm and implanted under the epidermis on the back of SD rats to evaluate the foreign body reaction in vivo. The specimens were harvested 2 weeks after operation. After fixed in 10% formalin solution, the specimens were sliced and stained with H&E to observe the inflammatory response zone, and randomly selected three fields of view for semi-quantitative and statistical analysis of inflammation area using ImageJ software (n=3) (Supplementary information, page 13, line 318). The spinal cord cavity area of each group was measured by ImageJ software, and three fields were randomly selected for statistical analysis (n=3) (Supplementary information, page 17, line 425).

Immunohistochemical evaluation: The primary antibody was added and incubated at 4 °C overnight, and the second antibody was incubated at room temperature for 1 hour after PBS washing. The samples were observed and photographed under fluorescence microscope, with the fluorescence intensity analyzed with ImageJ software. The values were represented in the form of average optical intensity, which had been corrected for optical density, de-background, and finally normalized by the number of cell nuclei. A total of five images were randomly selected from each group for statistical analysis (n=5) (Supplementary information, page 17, line 428).

Questions regarding the concept of the implant:

- There is no evidence provided in the manuscript regarding the release of plasmid and of NGF over the time-span of the experiment (4-8 weeks). Thus, it is not clear if the implant is really operating in accordance with the projected performance.

Re : Thank you for the comment. This study aimed to construct a responsive biomimetic immunoregulatory electrospinning fiber scaffold that can respond and then immune-regulate the local microenvironment during the acute spinal cord injury and provide a differentiation platform for subsequent nerve regeneration. Therefore, demonstration of early rapid released IL-4 plasmid and sustained released NGF *in vivo* was of great importance in our study. Thus, we've further evaluated and discussed whether biomimetic composites can achieve expected performance in terms of biological effects in *in vivo* research. Additionally, we had referred to associated references to support our discussion³.

(1). We've established the model for the simulation of the *in vivo* pH-responsive release process of liposome according to the procedure previously reported⁴. Figure 4E showed that the biomimetic composite scaffold responded to the acidic environment and released cationic liposomes loading with IL-4 plasmid (approximately 90%) within 24 hours, which was consistent with the time window of blood-derived macrophages appearing at the spinal cord injury site⁵. In addition, the release kinetics of NGF under simulated pH environment was also measured in long time-span. As we could see in release kinetic of NGF measured using ELISA kits in Figure 4F, the acidic pH did have some effect on the release of NGF, reflected by the burst release of 40.06±2.29% (320.48±18.33ng) under pH5.8, 36.59±3.19% (292.69±25.52ng) under pH6.6 and 20.86±5.92% (166.88±47.35ng) under pH7.4 in the first two days. However, owing to the effect of microsol spinning, which provided a relatively stable environment for the core part loaded with NGF, a generally controlled release of the factor could be identified, indicated by the slowly and stably release in the last 6 weeks under all three pH, verifying the effectiveness of our system. Specifically, NGF was allowed for a stable and sustained release for 40 days in an acidic environment of pH 5.8 (87.99±1.90%, 703.89±15.24ng). Our group have several years' experience on microsol spinning technique, which in current study, showed acceptable performance on protecting the factor loaded by core against harsh microenvironment (Figure 4F).

Specifically, the releasing rates of liposome was tested through immersing the liposome-grafted fiber scaffold in PBS solutions with different pH (Fig. 4E). In the acidic environments with pH of 5.8 and 6.6, the release of liposomes increased gradually with time. The release rate of liposomes under the pH of 5.8 reached 61.66±1.38% within 10 hours, which was significantly higher than that of other pHs (p<0.05). Therefore, quick respond to acidic environment of the responsive fiber scaffold could be verified in vitro which provided immunoregulatory signals for gathered immune cells during acute spinal cord injury in time (page 12, line 315).

Hence, the release characteristics of NGF from microsolv electrospun fibers under different pH were also evaluated (Fig. 4F). The results showed that the release rate under the pH of 5.8 was slightly increased in acidic environment, reaching $71.64 \pm 1.82\%$ ($544.21 \pm 21.41 \text{ ng}$) on the 10th day, while $68.14 \pm 4.59\%$ ($545.09 \pm 36.72 \text{ ng}$) and $59.54 \pm 1.93\%$ ($476.29 \pm 15.41 \text{ ng}$) under pH of 6.6 and 7.4. All scaffolds under different pH exhibited sustained releases of NGF which lasted for more than 40 days (page 12, line 326).

Figure 4E, F. (E) Cumulative release curve of aliposomes in different pH environments. ($n=3$, aliposome release values were mean \pm std. dev., $p < 0.05$ when comparing pH 5.8 and the other two pHs via two-way analysis of variance (ANOVA) with a Tukey's post-hoc test) (F) Cumulative release curve of NGF in microsolv electrospinning in different pH environments. ($n=3$, NGF release values were mean \pm std. dev., statistical analysis evaluated by two-way analysis of variance (ANOVA) with a Tukey's post-hoc test)

(2). Apart from the release kinetics, *in vivo* biological effects of immune regulation in acute spinal cord injury was also characterized in order to verify the immunological effect of liposome. One week after surgery, immunofluorescence staining was employed to evaluate the localized levels of IL-4, TNF- α (M1), and IL-10 (M2) in rat (Figure 9A), which indicated significantly higher IL-4 and IL-10 fluorescence intensity in the MSaP-aL/p group ($p < 0.05$) compared with control groups (blank control, aP, MSaP, MSaP-aL). However, significantly lower TNF- α intensity was captured in MSaP-aL/p group ($p < 0.05$) (Figure 9B, C). In order to more accurately evaluate the transfection performance and the levels of pro-inflammatory and anti-inflammatory response, we collected spinal cord tissue for IL-4, TNF- α , IL-1 β , IL-10 and TGF- β gene expression detection using qRT-PCR assay one week after surgery (Figure 9D-H). The qRT-PCR test results were consistent with the fluorescence staining results, which also showed that biomimetic electrospun fibers have exerted immunoregulatory effects at the injured site as expected during acute spinal cord injury. The specific experimental method and results are shown below and highlighted in the revised manuscript.

Method: To evaluate the transfection performance and immunoregulatory function after fiber bundles implantation, three randomly selected spinal cord specimens were dehydrated in ethanol followed by xylene ($n=3$). Then total RNA of spinal cord was isolated by using the RNeasy Pure FFPE kit (TIANGEN BIOTECH, China) according to the manufacturer's instructions. cDNA was synthesized with 1 μg of RNA using a Hiscript II 1st Strand cDNA Synthesis kit (Vazyme, Nanjing, China). Quantitative RT-PCR was

carried out in a 20 μ L total volume ChamQ Universal SYBR qPCR Master Mix (Vazyme, Nanjing, China). The gene expression levels were normalized to GAPDH. Primers were shown in Table 5 (Supplementary information, page 15, line 382).

Table 5. The primer sequence of immune factor for qRT-PCR in vivo.

Gene	Primer	Sequence	Tm (°C)
IL-4	Forward	GCAACAAGGAACACCACGG	60
	Reverse	AAGCACGGAGGTACATCACGT	57.8
TNF- α	Forward	TACTGAACTTCGGGGTGATTGGTCC	62.22
	Reverse	CAGCCTTGTCCTTGAAGAGAACC	62.11
IL-1 β	Forward	GAAAGACGGCACACCCACC	62
	Reverse	AAACCGCTTTTCCATCTTCTTCT	54.86
IL-10	Forward	CCCTCTGGATACAGCTGCG	62
	Reverse	GCTCCACTGCCTTGCTTTTATT	56.26
TGF- β	Forward	CTTCAGCTCCACAGAGAAGAACTGC	62.22
	Reverse	CACGATCATGTTGGACAACTGCTCC	62.22
GAPDH	Forward	AACTCCCATTCTTCCACCT	56
	Reverse	TTGTCATACCAGGAAATGAGC	53.90

Results and discussion. Meanwhile, according to the results of the expression of IL-4 gene in MSaP-aL/p group was significantly higher than that of other groups ($p < 0.05$) (Fig. 9D). Combined with the above fluorescence labeled IL-4 results, it was further proved that it had played the function of transfection in response to the acidic environment of spinal cord injury. The expressions of pro-inflammatory genes TNF- α and IL-1 β in MSaP-aL/p, shown as Fig. 9E, F, were significantly decreased ($p < 0.05$). On the other hand, Fig. 9G, H showed the expressions of anti-inflammatory genes IL-10 and TGF- β were significantly higher than those in other control groups, respectively. The results showed that the immune bundles had played an immunomodulatory role during the acute spinal cord injury (page 19, line 533).

Figure 9D-H. Spinal cord specimens of IL-4 (D), TNF- α (E), IL-1 β (F), IL-10 (G) and TGF- β (H) gene expressions were assessed by qRT-PCR. ($n=3$, qRT-PCR values were mean \pm std. dev., * $p < 0.05$ when comparing MSaP-aL/p and other groups via one-way analysis of variance (ANOVA) with a Tukey's post-hoc test)

(3). The long-term biological effects of immune regulation on spinal cord injury was also

characterized using immunofluorescence to label activated astrocytes and scar tissues at 4w and 8w after operation (Figure S11). We found significantly lower intensity from labeled activated astrocytes and scar tissues in the MSaP-aL/p group compared with non-immunized control group (blank control, aP, MSaP, MSaP-aL) ($p < 0.05$). This result confirmed that immune intervention during acute spinal cord injury can significantly reduce astrocyte activation and scar tissues formation during chronic stage of spinal cord injury, which could pave the way for neural regeneration.

Figure S11. Immunofluorescence staining of activated astrocytes and glial scar. (A), (C) Immunofluorescence staining of astrocytes and quantitatively analyzed with optical density along with (B), (D) their quantification with optical density. ($n=5$, all values were mean \pm std. dev., $*p < 0.05$ when comparing MSaP-aL/p and other control groups via two-way analysis of variance (ANOVA) with a Tukey's post-hoc test, $^{\#}p < 0.05$ when comparing material transplantation groups and blank control group via two-way analysis of variance (ANOVA) with a Tukey's post-hoc test, scale bars were $50\mu\text{m}$)

(4). Neural regeneration was also fully characterized. As you stated in the question, it is important to explore whether biomimetic composite fibers can control the release of NGF and play a role in promoting neural differentiation *in vivo* as expected. Therefore, the assessment of biological effects on neural regeneration was conducted according to the previously reported study with the time point set for 4 and 8 weeks after surgery ⁶. As shown in Figure 10A, B, the sustained release of NGF could be verified through significantly promoted neural differentiation of endogenous progenitor cells. In order to ensure the accuracy of the description of the experimental results, we have partially modified the original description as shown below and highlighted it in the revised manuscript.

At 4w and 8w time points, neural progenitor cells were continuously promoted by NGF

released from microsol electrospun fibers (MSaP, MSaP-aL and MSaP-aL/p), and a large number of Tuj-1-labeled neurons appeared as shown in Fig. 10A. It was significantly better than that in blank control and aP groups (Fig. 10B) ($p < 0.05$). Meanwhile, the fluorescence intensity of each microsol electrospinning group at 8w was significantly higher than that at 4w respectively ($p < 0.05$). The result further proved that the biomimetic fiber bundle has the function of continuously promoting nerve regeneration. In the MSaP-aL/p group, due to the effect of responsive immunomodulation, the toxicity of astrocytes and the formation of scar tissues were significantly reduced, so at two time points, the amounts of neurons were significantly larger than that in other control groups ($p < 0.05$). The above results confirmed that the biomimetic bundle not only had the ability of immunomodulation in the area of spinal cord injury, but also consistent with the reported in the literature that the core-shell structure of microsol electrospun fibers has the ability to protect NGF from the adverse biological environment which could continuously provide differentiation information for nerve regeneration (page 21, line 598).

Figure 10A, B. (A), Immunofluorescence staining of neuron cells and (B) quantification analyzed with optical density. ($n=5$, Tuj-1 intensity values were mean \pm std. dev., $^{\%}p < 0.05$ when comparing blank control and microsol electrospun groups (MSaP, MSaP-aL and MSaP-aL/p); $^{\&}p < 0.05$ when comparing aP and microsol electrospun groups (MSaP, MSaP-aL and MSaP-aL/p); $^{\textcircled{a}}p < 0.05$ when comparing 4w and 8w in MSaP; $^{\textcircled{s}}p < 0.05$ when comparing 4w and 8w in MSaP-aL; and $^{\textcircled{\#}}p < 0.05$ when comparing 4w and 8w in MSaP-aL/p; $^{\ast}p < 0.05$ when comparing MSaP-aL/p and other control groups via two-way analysis of variance (ANOVA) with a Tukey's post-hoc test, scale bars were $50\mu\text{m}$)

- To be effective, the plasmid needs to be internalised by the target cells. No evidence is provided in the histology of the explants that the cells were effectively transfected by the plasmid. Indeed, the plasmid should have a reporter gene that should facilitate the evaluation of the transfection at the defect site.

Re : Thank you for the thoughtful suggestion. The material characterization experiments have initially demonstrated that the Schiff base bond responds to the local acidic

microenvironment, releasing nearly 90% of the cationic liposomes loading with the IL-4 plasmid within 24h (Figure 4E). One week after surgery, we would like to prove the transfection effect of the IL-4 plasmid by paraffin section, but we could not observe the tissue fluorescence of eGFP expression due to the samples soaked with formalin or hydrogen peroxide interference. Hence, in order to fully demonstrate the transfection and immunoregulatory effect of the cationic liposome loading with IL-4 plasmid *in vivo*, we added qRT-PCR technique to access IL-4, TNF- α , IL-1 β , IL-10, TGF- β gene expression in local tissue.

We have supplied the IL-4 plasmid profile to Fig. S1A, and the qRT-PCR experimental method and results have described as above (Figure 9D-H) and highlighted in the revised manuscript.

Figure S1A. IL-4-eGFP plasmid profile. GFP was designed for the report protein of IL-4 expression.

- The NGF is present in the nano fibres during the whole procedure for the grafting of the liposomes at the surface. How much of the NGF is preserved and to what extent it is bioactive after being exposed to this procedure?

Re : Thank you for the significantly comment.

(1). Drug loading efficiency (LE) is an important parameter in electrospinning as a cytokine sustained release system. To clarify, we have added the experimental content of microsol electrospinning nanofiber related drug loading efficiency as shown below and highlighted in the original article.

Method: *To characterize the cytokine content in the fibers, three prepared MSaP-aL/p scaffolds were placed in three 50ml centrifuge tubes containing 2g DCM, centrifuged at 15000rpm for 5 minutes, and the aPLA-containing supernatant was removed. This was repeated three times to ensure that all NGF in the organic phase precipitation. After adding 2ml PBS for resuspension, the ELISA kit was used to detect the NGF content, and the drug loading efficiency of electrospun fibers was calculated as follows:*

(Supplementary information, page 7, line 171)

$$LE (\%) = \frac{M (\text{total drug in fibers})}{M (\text{total drug before spinning})} \times 100\%$$

Results and discussion: *As measured, the drug loading efficiency (LE) of all NGF-loaded microsolv electrospun scaffolds were about 79.53±1.44% in this study (page 12, line 322).*

(2). In order to ensure the biological activities of the IL-4 plasmid and NGF in the biomimetic immunoregulatory fibrous scaffold, we carried out relevant characterization immediately after preparation or stored at -20°C for subsequent experiments. In the study of biomimetic fiber membrane release, we also kept it under sterile conditions at -20°C after each sampling to maximally maintain the biological factor activity. We've used the previously collected release solutions which have been stored at -20°C for the 30th to 40th days to culture BMSCs *in vitro* to verify the long-term bioactivity released NGF. The specific methods and results are shown below and highlighted in the revised manuscript.

Method: *Rat bone marrow mesenchymal stem cells (BMSCs) were also seeded into 24-well plates at a density of 1×10⁴/well (n=3). When the cells were about 70% -80% fused, added MSaP-aL/p release solution (day 30 to 40, pH5.8). Five days later, the neuronal marker Tuj-1 was used to label differentiated neuron-like cells and semi-quantitative fluorescence analysis was performed by imageJ software to evaluate the long-term biological activity of NGF released from the fibrous membrane (r-NGF group). Normally cultured bone marrow mesenchymal stem cells were used as the negative control group (NC), and rat β-NGF cytokines (20ng/ul) were added as the positive control group (PC). Three separate dissociations were carried out to achieve biological triplicate (n=3) (Supplementary information, page 12, line 296).*

Results and discussion :*During the neural regeneration process, it is necessary to provide long-term biological signal supply chains for neural progenitor cells. Therefore, it is very important to prove that the biological activity of NGF that is sustained released by biomimetic electrospun fibers. The neuron-specific marker Tuj-1 was still able to label neuron-like cells differentiated from bone marrow mesenchymal stem cells induced by release solution (Fig. S8A). At the same time, without the contact guidance of oriented electrospinning fibers, the cell morphology changed from spindle to long, but no neurite-like formation was observed. As shown in Fig. S8B, the PC (positive control group) is higher in the semi-quantitative fluorescent results than the r-NGF group, but there was no statistical difference between the two groups (p>0.05). However, the negative control group was significantly different from the r-NGF group and the PC group, respectively. This result combined with in vitro simulated release experiments proved that the biomimetic fibers can sustained release NGF, and that it can be protected by the core-shell structure to maintain its biological activity (page 16, line 446).*

Figure S8. Assessing long-term biological activity of NGF in biomimetic fibers. (A) Immunofluorescence staining of Tuj-1 staining images on cell culture plate (NC), release solution group (r-NGF) and cytokine NGF group (PC) with (B), Corresponding fluorescence semi-quantitative analysis. ($n=3$, all values were mean \pm std. dev., $*p < 0.05$ when comparing r-NGF, PC and NC group via one-way analysis of variance (ANOVA) with a Tukey's post-hoc test., ns, not significant, scale bars were 200µm)

- The size of the liposomes is quite large to enable an easy internalization. The authors should discuss this important aspect in the manuscript.

Re : Thank you for the comment. As the reviewer said, the size of liposomes is indeed crucial for cell transfection, so we supplemented the discussion concerning the size effect of liposomes in the revised manuscript.

In addition, according to reports, liposomes with a diameter of 200nm-1µm can enter the cell slowly through endocytosis, so that DNA can be released from the liposome before reaching the lysosome, which can prevent its degradation in the lysosome⁷. After screening the pDNA/aLiposome complex at various ratios in this study, it was found that the transfection efficiency was the highest at 1: 2.5, and the average size was 224.1±4.4nm, so it could be used in subsequent studies (page 8, line 191).

- No details are provided regarding the method used to obtain aligned nano fibrous constructs and its details (dimensions, porosity, degree of alignment, ...). Also the dose of plasmid and NGF effectively implanted and released should be provided.

Re : Thank you for the comment.

(1). We have perfected the description of each group of nano-electrospinning solution and the oriented electrospinning process. The details are as follows and highlighted in the original manuscript:

a. Conventional electrospinning solution : *Aminated polylactic acid electrospinning solution was prepared by dissolving 0.5g amino polylactic acid (aPLA, Mw=100kDa, Mw / Mn=2.06, Ruixi, Xi'an, China) in 4g dichloromethane (DCM, Sinopharm, Beijing, China) and 2g N, N-dimethylformamide (DMF, Lingfeng, Shanghai, China) to obtain uniform and stable solution under magnetic stirring at room temperature. The preparation method of polylactic acid (PLA, Mw=100kDa, Mw/Mn=1.61, Daigang, Jinan, China) electrospinning solution is the same as above (Supplementary information, page 4, line 91).*

b. Microsol electrospinning solution : *The preparation of microsol electrospinning solution as described in the literature⁸. Briefly, started with the preparation of sodium hyaluronate hydrosol (1wt%) (HA, Mw=0.5MDa, Yuancheng, Wuhan, China), which was obtained by dissolving 0.1g HA in 9.9g deionized water and rotating to complete dissolution at room temperature. Rat β -NGF (R&D system, USA) was resuspended in 0.1wt% bovine serum albumin (BSA, Solarbio, Beijing, China) solution to achieve the final concentration of 100 μ g/ml. The uniform HA- β -NGF hydrosol was obtained by mixing 10 μ l resuspended β -NGF with 50 μ l hyaluronic acid solution, after which 0.01g Span-80 (Sigma, USA) and 4g DCM were added into the mixture and stirred at high speed for 30 minutes at room temperature to obtain a homogeneous and stable water-in-oil (W/O) emulsion containing β -NGF microsol particles. The microsol (MS) spinning solution was obtained eventually by adding 0.5g aPLA and 2g DMF into the emulsion. Preparation of different oriented electrospinning fibers (Supplementary information, page 4, line 97).*

c. Oriented electrospinning fibers: *As for the preparation of different oriented fiber scaffolds, the aforementioned electrospinning solutions were loaded in a 10ml syringe with a length of 10cm and an inner needle diameter of 0.9mm. The electrospinning process was conducted with a propulsion pump speed of 70 μ l/min, at a voltage of 15 to 18kV, and a distance between needle tip and the parallel electrode receiver of 15cm. The parallel oriented fiber scaffolds can be collected between the electrode rods. In order to remove the residual solvent from the obtained fibers, all prepared fiber scaffolds were dried under vacuum overnight. Traditional electrospun directional fiber (PLA, aP) and microsol electrospun oriented fibers (MSaP) were also obtained. In the following studies, the oriented fiber membrane was used in vitro study, while the fiber bundle was used in vivo research (Supplementary information, page 5, line 110).*

(2). We have performed scanning electron microscopy (SEM), transmission electron

microscopy (TEM), X-ray photoelectron spectroscopy (XPS), atomic mechanical microscopy (AFM) and diameter analysis on traditional oriented electrostatic spinning (aP), oriented microsolv electrospinning (MSaP) and oriented responsive nerve fibers (MSaP-aL/p) for material morphological characterization, and representative images were shown in Figure 3 to show the differences among groups. In addition, we supplemented the characterization of the fibers orientation and porosity with corresponding experimental methods and results were added in the article.

Figure 3. The morphology of different fiber scaffolds. (A) Schematic diagram of amino poly(lactic acid) microsolv electrospun fiber and aldehyde cationic liposomes connected with Schiff base. (B), (C), (D) SEM images of different fiber scaffolds. (E), (F), (G) Histogram of frequency distribution of different fiber scaffolds diameters. ($n=100$ fibers from each sample) (H), (I), (J) TEM images of different fiber scaffolds. (K), (L), (M) XPS indicating chemical elements on different fiber scaffold surfaces. (N), (O), (P) AFM images of scaffolds. The yellow arrows in the images indicate the location of liposome. ($n=3$, all SEM scale bars were $5\mu\text{m}$, all TEM scale bars were $0.2\mu\text{m}$)

Orientation method: To characterize the orientation of the fiber scaffold, we randomly selected three SEM images from each group, and the overall direction of the fiber membrane was set to 0° . Orientation analysis through the orientation plugin of ImageJ

software (Supplementary information, page 5, line 137).

Results and discussion: Nano-topology of neural tissue engineering biomaterials is one of the important factors affecting cell behavior. Biomimetic oriented electrospinning fiber scaffold can provide contact guidance in vivo or in vitro for the growth and extension of axons of in situ neurons or induced differentiated neuron-like cells. As shown in Fig. S2C, the orientations of the three groups of scaffold fibers at 0° were 10799.59±1152.65, 30435.12±3440.99, 49274.01±2292.76, which indicates that the fiber orientation of each group is highly consistent (page 9, line 221).

Figure S2C. Analysis of orientation of different fiber scaffolds. ($n=3$, orientation values were mean \pm std. dev)

Porosity method: The degree of porosity of the each group scaffolds was determined by the pycnometer method following the procedure described elsewhere⁹. Briefly, the weight of the pycnometer filled with ethanol was measured and labeled as W_1 ; the sample with a weight of W_s , and was immersed in ethanol. Subsequently, the sample was saturated by ethanol; additional ethanol was added to complete the volume of the pycnometer. Then, the pycnometer was weighted and labeled as W_2 ; the sample filled with ethanol was taken out of the pycnometer and the residual weight of the ethanol and the pycnometer was labeled as W_3 . The porosity of the membrane was calculated according to:

$$\mathcal{E} = \frac{W_2 - W_3 - W_s}{W_1 - W_3} \times 100\%$$

The porosity of each scaffold was obtained as the mean value of the porosity determined in three samples (Supplementary information, page 8, line 197).

Results and discussion: One of the advantages of electrospinning fiber scaffolds becoming a hot spot in tissue engineering research is that it has large porosity, which can provide more growth space for cells. As mentioned above, there is no statistically significant difference in the diameter of the three groups of fiber scaffolds ($p>0.05$), so the factors that change the porosity due to different diameters can be excluded. The experimental group MSaP-aL/p compared with aP and MSaP two groups, the porosity increased significantly ($p<0.05$) (Fig. S3C). This phenomenon can also be explained by the lipophilicity of aminated PLA and phospholipid, during the grafting process, some adjacent fibers fused, and the internal bulk density decreased, resulting in increased porosity (page 11, line 278).

Figure S3C. Porosity of different fiber scaffolds. ($n=3$, porosity values were mean \pm std. dev and p values were determined by one-way analysis of variance (ANOVA) with a Tukey's post-hoc test)

(3). In this study, we hoped that the IL-4 plasmid loaded with aldehyde cationic liposomes will play a transfection role during the acute early stage of spinal cord injury to secrete IL-4, and immunoregulatory the local microenvironment of spinal cord injury to pave the way for long-term neural regeneration. In this article, we discussed the encapsulation efficiency, release characteristics, and biological activity of the IL-4 plasmid from the following three aspects :

a. In the process of characterizing the biomimetic fiber scaffold, we used a dsDNA quantification kit to assess the concentration change of the IL-4 plasmid before and after grafting. The initial concentration of the IL-4 plasmid was $1.16 \pm 0.06 \mu\text{g}/\mu\text{l}$. The concentration was $0.83 \pm 0.01 \mu\text{g}/\mu\text{l}$ (approximately 71.49%) after forming the pDNA/aLiposome complex, with $0.23 \pm 0.01 \mu\text{g}/\mu\text{l}$ (approximately 19.38%) grafted on the fiber membrane by Schiff base bond (Figure 4D). Subsequently, the response characteristics of the biomimetic composite fiber membrane showed that $0.19 \pm 0.02 \mu\text{g}/\mu\text{l}$ (approximately 85.72%) was released within 24 hours in an acidic environment at pH 5.8 (Figure 4E). This result indicates that the composite fiber scaffold rapidly responds to release a large number of liposomes for immunoregulation in an acidic environment, and as mentioned above, it coincides with the time window of the occurrence of blood-derived macrophages in the spinal cord injury.

Figure 4D, E. (D) Concentration of pDNA in the aliposome solutions before and after grafting at the surface of scaffold. (E) Cumulative release curve of aliposomes in different pH environments. ($n=3$, aliposome release values were mean \pm std. dev., $p < 0.05$ when comparing pH5.8 and the other two pHs via two-way analysis of variance (ANOVA) with a Tukey's post-hoc test)

b. In the *in vitro* cell assay, macrophages were stimulated by IL-4 cytokines after transfection of the IL-4 plasmid in the co-culture environment at pH 5.8. Compared with other groups, MSaP-aL/p significantly decreased the pro-inflammatory genes expression while increased the anti-inflammatory genes expression at day 3 ($p < 0.05$) (Figure 5A, D and Figure 5G, J), and similar trend could be found on the secretion of pro-inflammatory and anti-inflammatory factors (Figure S6A, D and Figure S6G, J).

Figure 5A, D, G, J. Evaluation of pro-inflammatory and anti-inflammatory factor gene expression of BMMs in different pH culture environments to simulate the microenvironment of spinal cord injury. The expression of IL-1 β mRNA (A), TNF- α mRNA (D), IL-10 mRNA (G), TGF- β mRNA (J) under pH5.8. ($n=3$, all values were mean \pm std. dev., * $p < 0.05$ and ** $p < 0.01$ when comparing MSaP-aL/p and other groups via two-way analysis of variance (ANOVA) with a Tukey's post-hoc test)

Figure S6. Evaluation of pro-inflammatory and anti-inflammatory factors secreted by BMM at different pHs by ELISA. IL-1 β (A), TNF- α (D), IL-10 (G), TGF- β (J) under pH5.8. ($n=3$, all values were mean \pm std. dev., Comparison between groups via two-way analysis of variance (ANOVA) with a Tukey's post-hoc test)

c. *In vivo*, we collected 1w postoperative spinal cord samples for immunofluorescence labeling of IL-4, TNF- α (M1 macrophages/microglia-specific cytokines), and IL-10 (M2-type macrophages/microglia-specific cytokines) (Figure 9A, B, C) and flow cytometry (FCM) to evaluate macrophages/microglia cell subtype changes in damaged tissues (Figure 8). Also, qRT-PCR was employed to test the expression of immune inflammation-related genes in damaged tissues (Figure 9D-H), followed by ELISA quantification of IL-4, pro-inflammatory and anti-inflammatory factors production in animal serum (Figure 9I-M). The above experiments have proved that the IL-4 plasmid plays an immunoregulatory role after implantation and improves the local environment of spinal cord injury.

Figure 8. Phenotypic identification of immune cells in vivo. (A), Flow cytometry analysis of CD11b/CD86-positive cells with (B) its quantification analyzed of the ratio and CD11b/CD206-positive cells with (C) its quantification analyzed of the ratio. (n=3, all values were mean \pm std. dev., #p<0.05 when comparing non-immunoregulatory fiber groups and blank control group and *p<0.05 when comparing MSaP-aL/p and other control groups via one-way analysis of variance (ANOVA) with a Tukey's post-hoc test., ns, not significant)

Figure 9. Evaluation of immune regulation 7 days after operation. (A) IL-4 immunofluorescence single labeling and TNF- α , IL-10 immunofluorescence double labeling. Quantitatively analysis of (B) IL-4 immunofluorescence, (C) IL-10 immunofluorescence, (D) IL-4 mRNA Expression, (E) TNF- α mRNA Expression, (F) IL-10 mRNA Expression, (G) IL-10 mRNA Expression, (H) TGF- β mRNA Expression, (I) Serum IL-4, (J) Serum TNF- α , (K) Serum IL-10, (L) Serum IL-4, (M) Serum TNF- α , (N) Serum IL-10.

immunofluorescence and TNF- α immunofluorescence. ($n=5$, all values were mean \pm std. dev., * $p<0.05$ when comparing MSaP-aL/p and other groups via two-way analysis of variance (ANOVA) with a Tukey's post-hoc test, scale bars are 50 μ m) Spinal cord specimens of IL-4 (D), TNF- α (E), IL-1 β (F), IL-10 (G) and TGF- β (H) gene expressions were assessed by qRT-PCR. Serum levels of IL-4 (I), TNF- α (J), IL-1 β (K), IL-10 (L), and TGF- β (M) were assessed by ELISA. ($n=3$, qRT-PCR and ELISA values were mean \pm std. dev., * $p<0.05$ when comparing MSaP-aL/p and other groups via one-way analysis of variance (ANOVA) with a Tukey's post-hoc test)

Based on the above three points, we could conclude the IL-4 plasmid released by the biomimetic responsive immunoregulatory fiber scaffolds during acute spinal cord injury participates in the immune response process, timely regulates blood-derived macrophages and in situ microglia subtypes, and improves the inflammatory environment to provide favorable conditions for subsequent long-term neural regeneration.

(4). As mentioned in the manuscript, the microsol electrospinning technology is based on the fact that the organic solvent evaporates faster than water during the spinning process. HA hydrosol moves and stretches inside the PLA to form a core-shell structure which could be able to protect the internal drug or cytokine activity from interference and destruction from external environment¹⁰.

a. In this study, the NGF loading efficiency was 79.53 \pm 1.44%, and we simulated the release process of *in vivo* by characterizing the sustained release properties of fiber scaffolds in different pH environments *in vitro* (Figure 4F). The cumulative release results were 320.48 \pm 18.33ng (40.06 \pm 2.29%, pH5.8), 292.69 \pm 25.52ng (36.59 \pm 3.19%, pH6.6) and 166.88 \pm 47.35ng (20.86 \pm 5.92%, pH7.4), which showed that in different acidic environments, MSaP-aL/p had a sudden release in the first two days. The NGF content in the release solution from day 38 to day 40 of each group can still be detected by the ELISA kit, which were 7.89 \pm 2.01ng (Rc (cumulative release)=703.89 \pm 15.24ng, pH5.8), 7.11 \pm 1.39ng (Rc=678.3 \pm 35.33ng) , PH6.6) and 10.24 \pm 1.63ng (Rc=669.87 \pm 38.14ng, PH7.4) respectively.

Figure 4F. Cumulative release curve of NGF in microsol electrospinning in different pH environments. ($n=3$, NGF release values were mean \pm std. dev., statistical analysis evaluated by two-way analysis of variance (ANOVA) with a Tukey's post-hoc test)

b. In order to further characterize the biological activity of NGF released by MSaP-aL/p, as

mentioned above, BMSCs were cultured with the release solution from day 30 to day 40, and the results showed that differentiated neuron-like cells could be labeled by neuronal specific marker tuj-1 after 5 days (Figure S8) indicating the bioactivity of controlled released NGF. The above results proved that the biomimetic microsolv electrospun fibers could sustained release NGF and provide neural differentiation biological information for stem cells.

Figure S8. Assessing long-term biological activity of NGF in biomimetic fibers. (A) Immunofluorescence staining of Tuj-1 staining images on cell culture plate (NC), release solution group (r-NGF) and cytokine NGF group (PC) with (B), Corresponding fluorescence semi-quantitative analysis. (n=3, all values were mean ± std. dev., *p<0.05 when comparing r-NGF, PC and NC group via one-way analysis of variance (ANOVA) with a Tukey's post-hoc test., ns, not significant, scale bars were 200µm)

Questions and comments raised by the animal study and its outcomes:

- The spinal cord implant is insufficiently detailed in the manuscript, being very difficult to understand its size, the shape, how it is fixed in the defect site,, how it is ensured a good and reproducible coverage of the hemisection defect created, the alignment of the fibres in the implant, the confinement at the defect site.

Re :Thank you for the comment. We modified and supplemented the surgical procedure for

rat spinal cord injury model as follows :

The spinal cord hemi-section model was established following the procedure described previous article ¹¹. Briefly, the rats were anesthetized by intraperitoneal injection of 2%

Pentobarbital (50mg/kg). After anesthesia, a 3cm longitudinal incision centered on T9 was made on the back of the rat, and the paraspinal muscles were separated. After removal the T9 lamina was fully exposed to the spinal canal, under the accurate measurement of the scale, the right half of the spinal cord was cut off to create a 3mm spinal cord hemi-section defect. Followed by saline irrigation, the fiber bundles (2mm×2mm×3mm) placed inside the defect (Fig7.A(b, c, d)). After sewing the fascia and skin layer by layer with sutures each rat was intramuscular injected with 2×10^5 units of antibiotics every day for 5 days and manually emptied the bladder every 12 hours. The bare laminectomy was conducted in sham operation group, and spinal cord hemi-section without the implantation of bundle was introduced as the blank control group. Models implanted with aP, MSaP and MSaP-aL were used as the negative control group (Supplementary information, page 13, line 330).

- How many researchers were involved in creating the defects? How was the training of those researchers performed? How consistent is the reproducibility of the defects created in the animal model?

Re : Thank you for the comment. The technique of rat hemisected spinal cord injury model is well-established in our group¹¹. All animal models were constructed by a spine surgeon from the First Affiliated Hospital of Soochow University with extensive clinical experience and solid surgical skills. During the modeling process, the operator used a scale to accurately measure the defect of each rat, and strived to make the spinal cord resection range of each rat the same size in order to minimize the operation error.

- The results reported do not identify the number of animals that were considered for the data reported in the results (both the histology and the functional evaluation of the recovery). Indeed, given the large number of animals tested, if the analysis does not include either all the animals tested or comparable subsets of randomly selected animals, it is very difficult to have a clear perception of the efficacy of this implant.

Re : Thank you for the comment. The histology and functional recovery assessment in the original manuscript listed representative images. At present, we have added the number of samples (n) for testing in each data description, and also corrected the methodological description as shown above.

- Much more details are needed to ensure that the procedures used to obtain the safety and efficacy profile of the implant are robust and representative of the spectra of outcomes obtained for each condition.

Re : Thank you for the comment.

(1). Although liposomes benefited from their low toxicity, low immunogenicity and genomic safety during transfection, their low encapsulation efficiency and transfection efficiency limit their development. To overcome this bottleneck limitation, we added stearylamine to endow liposomes a positive charge and the stability of liposomes were evaluated using a phospholipid detection kit at different time points within 24 hours after preparation (Figure S1B), which is crucial during the transfection. We then optimized the ratio of pDNA / aLiposome as 1:2.5 for subsequent experiments due to the best transfection efficiency according to the results of immunofluorescence labeling and ELISA (Figure 2).

Figure S1B. Detection of stability of aldehyde cationic liposomes. ($n=3$, all values were mean \pm std. dev)

Figure 2. Evaluation of transfection efficiency of aldehyde cationic liposomes loading different ratio of pDNA. (A) GFP expression indicated by green fluorescence in BMSCs

after transfecting 72h. (B) Analysis of nuclear and eGFP fluorescence overlap coefficient rate. (C) Quantification of the IL-4 secretion after 1, 3, 5, and 7 days of transfection. The red marks represented the highest group of transfection efficiency. (n=3, all values were mean \pm std. dev., * p <0.05 when comparing 1:2.5 and other ratios of pDNA/aLiposome via one-way analysis of variance (ANOVA) with a Tukey's post-hoc test, scale bars were 1000 μ m)

(2). We kept all grafted biomimetic composite fiber membranes at -20°C to maintain the bioactivity of cytokines and plasmids. The results of simulated release studies in different pH environments *in vitro* showed that the composite fiber scaffold had a sensitive acid response (rapid release of cationic liposomes loaded with IL-4 plasmid and sustained release of NGF), which can provide experimental support for subsequent cell and animal research (Figure 4E, F). *In vitro* experiments, we evaluated the proliferation and adhesion properties of fiber scaffolds using live/dead staining, CCK8 and Integrin β 1 immunofluorescence staining labeling methods, respectively (Figure S4 and Figure S5). Both results showed that fiber scaffolds have favorable biocompatibility. In addition, we implanted each group of fiber membranes into the subcutaneous tissue of the rat's back to further evaluate the foreign body reaction in the animal (Figure S9), and proved the safety of the implantation of fibrous scaffolds.

Figure S4. Cell survival and proliferation assays of membranes. (A) Live/dead (green/red) fluorescence staining images. (all scale bars are 1000 μ m) (B) Fluorescence semi-quantitative analysis of living cells. (n=3, fluorescence density values were mean \pm std. dev., * p <0.05 when comparing control group and four other groups via one-way analysis of variance (ANOVA) with a Tukey's post-hoc test., ns, not significant) (C) Detection of cell proliferation by CCK8 kit at 1, 3 and 5 days. (n=6, absorbance values were mean \pm std. dev., * p <0.05 when comparing control group and four other groups via two-way analysis of variance (ANOVA) with a Tukey's post-hoc test., ns, not significant)

Figure S5. Cell adhesion assays of scaffolds. (A) Integrin $\beta 1$ immunofluorescence staining images of BMSCs inoculated with fiber scaffolds for 1 day. (B) Fluorescence semi-quantitative analysis of Integrin $\beta 1$. (C), (D), (E), (F) SEM images of BMSCs cultured on different fiber scaffolds for three days. ($n=3$, integrin $\beta 1$ intensity values were mean \pm std. dev., $*p < 0.05$ when comparing control group and four other groups via one-way analysis of variance (ANOVA) with a Tukey's post-hoc test., ns, not significant, fluorescent images scale bar were $100\mu\text{m}$ and $50\mu\text{m}$ in SEM images)

Figure S9. Characterization of foreign body reaction of fiber scaffolds. (A) General observation and corresponding H&E staining after subcutaneous implantation on the back of rats for 2 weeks. The yellow arrow refers to the inflammatory response area around the fiber scaffolds. (B) Analysis of inflammatory band area of HE staining by ImageJ software. (C) Schematic diagram of fibrous scaffolds implanted subcutaneously on the back of rats. ($n=3$, all values were mean \pm std. dev., $*p < 0.05$ when comparing

MSaP-aL/p and other groups via one-way analysis of variance (ANOVA) with a Tukey's post-hoc test., ns, not significant, scale bars were 50 μ m)

Other issues of the manuscript:

- The authors described similar information on the Introduction and on the Results and Discussion sections. Please avoid duplication of information along the manuscript, by providing relevant background information in the Introduction section.

Re : Thank you for the comment. We have made the following modifications to the duplicates and highlighted in the revised manuscript.

(1). **Results and discussion opening remarks.** *The bioinspired microenvironment-responsive fiber has been constructed step-by-step as shown in scheme 1. In order to verify the effect of materials, physiochemical property, in vitro and in vivo biological characteristics of different combination were characterized comprehensively in this study with the group setting depicted in Table 1 (page 6, line 138).*

Table 1. Material grouping.

Group	Denoted	Function
PLA electrospinning	PLA	Control
Amino PLA electrospinning	aP	Control
Amino PLA microsol electrospinning	MSaP	Control
Amino PLA microsol electrospinning carrying blank cationic liposome	MSaP-L	Control
Amino PLA microsol electrospinning carrying blank aldehyde cationic liposomes	MSaP-aL	Control
Amino PLA microsol electrospinning carrying aldehyde cationic liposomes loading IL-4 plasmid	MSaP-aL/p	Test

(2). **Biomimetic composite fiber simulates the release of liposomes in vitro.** *Severe injury of local blood vessels caused by external mechanical damage of spinal cord will lead to a sharp decrease of pH, resulting in an acidic microenvironment in the injured area. Based on this pathological characteristic, the focal pH was simulated in vitro for the characterization of microenvironment-responsive releasing functionality. Specifically, the releasing rates of liposome was tested through immersing the liposome-grafted fiber scaffold in PBS solutions with different pH (Fig. 4E). In the acidic environments with pH of 5.8 and 6.6, the release of liposomes increased gradually with time. The release rate of liposomes under the pH of 5.8 reached 61.66 \pm 1.38% within 10 hours, which was significantly higher than that of other pHs ($p < 0.05$). Therefore, quick respond to acidic environment of the responsive fiber scaffold could be verified in vitro which provided basis*

for further study (page 12, line 311).

Figure 4E. Cumulative release curve of aliposomes in different pH environments. ($n=3$, aliposome release values were mean \pm std. dev., $p < 0.05$ when comparing pH 5.8 and the other two pHs via two-way analysis of variance (ANOVA) with a Tukey's post-hoc test)

(3). **Gene expression profiles of BMMs in vitro.** The expression of pro-inflammatory and anti-inflammatory genes in BMMs cultured in different pH was detected after co-culture for 1, 3, 5 and 7 days. In the culture medium with pH of 5.8, the expression of pro-inflammatory genes IL-1 β and TNF- α in MSaP-aL/p group decreased gradually as time went on, reaching the lowest expression of 9.63 ± 0.80 and 0.30 ± 0.03 fold (Fig. 5A, D) on the 7th day respectively. At the same time, the expression of anti-inflammatory gene IL-10 and TGF- β were 3.42 ± 0.07 , 11.59 ± 0.09 , 27.02 ± 0.45 , 29.30 ± 0.51 fold and 2.85 ± 0.06 , 7.99 ± 0.07 , 22.52 ± 0.37 , 27.47 ± 0.59 fold on the 1st, 3rd, 5th and 7th day, respectively. Expression of all anti-inflammatory gene showed an upward trend (Fig. 5G, J), which was significantly different from that of groups ($p < 0.01$). Under the pH of 6.6, MSaP-aL/p group also showed response to acidic environment. However, due to the decreased acidity, the response of Schiff base to environment decreased, followed by relatively less significant changes on pro-inflammatory and anti-inflammatory gene expression, compared with those under pH of 5.8 (Fig. 5B, E, H, K). Under the pH of 7.4, the differences on pro-inflammatory and anti-inflammatory genes expression was smaller and showed insignificant difference between different groups at each time point ($p > 0.05$). Surprisingly, pro-inflammatory genes, IL-1 β and TNF- α , showed a downward trend while anti-inflammatory genes, IL-10 and TGF- β , showed an upward trend (Fig. 5C, F, I, L). This could be explained that the lower chamber BMSCs had a certain degree of immunoregulation function, and could shift M1 macrophages into M2 macrophages, and inhibit the expression of inflammatory genes (page 14, line 377).

Figure 5. Evaluation of pro-inflammatory and anti-inflammatory factor gene expression of BMMs in different pH culture environments to simulate the microenvironment of spinal cord injury. The expression of (A), (B), (C) IL-1 β mRNA, (D), (E), (F) TNF- α mRNA, (G), (H), (I) IL-10 mRNA and (J), (K), (L) TGF- β mRNA at different pHs of BMMs. (n=3, all values were mean \pm std. dev., * $p < 0.05$ and ** $p < 0.01$ when comparing MSaP-aL/p and other groups via two-way analysis of variance (ANOVA) with a Tukey's post-hoc test)

(4). **BMSCs differentiate into neuron-like cells in vitro.** Endogenously secreted NGF was responsible for the maintenance of multiple neurophysiological functions. BMSCs was introduced to mimic endogenous stem cell and investigate the biological effect of NGF-loaded fibers. After cultured on different fiber membranes for 10 days, immunofluorescence staining and qRT-PCR technique were employed to detect the expression of neuron-specific markers in neuron-like cells, including neurofilament protein (NF-200), neuron specific enolase (NSE), nerve cytoskeleton tubulin (Tau protein) and neuron cell specific differentiation tubulin (Tuj-1), in order to evaluate the neurogenic activity of fiber membranes (Figure 6) (page 16, line 422).

Figure 6. Neuron-specific marker immunofluorescence staining and gene detection assays of scaffolds. (A) Immunofluorescence staining of Tuj-1, NSE, NF-200, Tau, staining images on different fiber scaffolds with (B), (C), (D), (E) Corresponding fluorescence semi-quantitative analysis. (F), (G), (H), (I) mRNA expression analysis of Tuj-1, NSE, NF-200, Tau. ($n=3$, all values were mean \pm std. dev., * $p < 0.05$ when comparing microsol electrospun groups (MSaP, MSaP-aL, MSaP-aL/p) and control, aP group respectively via one-way analysis of variance (ANOVA) with a Tukey's post-hoc test., ns, not significant, scale bars were 100 μ m)

- The achieved results are poorly compared to the achievements of others already described in the literature.

Re : Thank you for the comment. Studies have shown that *in vitro* biomaterials carrying neural-promoting cytokines can induce neural differentiation of stem cells. However, when the cytokine-loaded biomaterials were implanted into the spinal cord injury site of animals, it was found that the immune inflammatory response in situ and the foreign body reaction brought by the implant significantly compromised the regeneration of neural tissue, and even the loaded cytokines would lose bioactivity in the malignant pathological environment of spinal cord injury¹². Therefore, researchers had focused on reducing the local immune response to spinal cord injury for better prognosis¹³. However, the single use of immune regulation measures to suppress the inflammatory response can only prevent the further development of spinal cord injury. It cannot provide biological signals that promote neuronal regeneration during the later-stage repair and regeneration process of central nervous tissue, bringing the bottleneck in the research of neural tissue

engineering treating spinal cord injury. In our previous research, it was found that oriented electrospinning scaffolds can contact guidance nerve cells well ¹⁴, and microsol electrospinning technology had excellent sustained-release properties and the function of core-shell structure could protect the biological activity of cytokines ¹⁵. Hence, in order to verify the immunoregulating effect of responsive transfection system, we grafted the cationic liposome onto this oriented microsol fibers which had been thoroughly studied and well recognized by our group. Since the pH-responsive immunoregulating function played the key role in this study, the oriented microsol fiber basically acted as a platform instead of the key point of the study. Despite that our study may not be comparable to those based on other platforms and materials, we've made our point on confirming the satisfying effect of pH-responsive immunoregulating system in this study. In order to achieve better outcome in the treatment of spinal cord injury, platforms made of better biomaterials and endowed with different attributes would be explored in our future experiment.

- Approximately 90% of IL-4 plasmid release is achieved in 24h at pH 5.8 (Figure 4 E). How relevant the action of IL-4 when kept under culture or implanted for longer time periods?

Re : Thank you for the comment. In situ microglia activation and macrophage infiltration appeared within 24 hours after acute spinal cord injury, reached the peak within 3 days and 7 days respectively ⁵, followed by differentiated into type M1 under the regulation of local environment, which can release inflammatory factors to aggravate local inflammatory response, promote apoptosis of surviving tissue cells, induce astrocyte activation and expand the scope of lesions, resulting in secondary spinal cord injury. However, the infiltrating macrophages could be induced to differentiate into M2, secreting biological factors such as anti-inflammation and promoting tissue regeneration ¹⁶. Based on the above pathological basis of spinal cord injury, this study hoped that cationic liposomes loaded with IL-4 plasmid could play an immunomodulatory role in a short time, promote local tissue to secrete IL-4, so that local and infiltrating related immune cells can shorten M1 action phase and differentiate into M2 at earlier stage, so as to reduce acute inflammatory reaction and reduce the release of toxic substances and retain a glimmer of vitality for later nerve regeneration. Hence, according to the physiological features, the effect of plasmid release at later stage after implantation was not our interest when characterizing immunoregulating capability.

In the release assay, the acid-sensitive Schiff base bond rapidly responded to the pH5.8 environment and released the 90% IL-4 plasmid within 24 hours(Figure 4E), and the results of *in vitro* BMMs polarization study (Figure 5, Figure S6, Figure S7) together with the results of *in vivo* immunoregulation effect characterization 1 week post-operation (Figure 8 and Figure 9) all supported the conclusion that the biomimetic composite fiber had an immunomodulatory function during acute spinal cord injury.

- Why was the effect on the pro-inflammatory and anti-inflammatory expression only observed on the 3rd day of BBM culture (Figure 6 G and J)

Re : Thank you for the comment. As mentioned above, the expectation of this study is that immunoregulation during acute spinal cord injury provides a favorable microenvironment for subsequent neural regeneration. *In vitro* cell assay, biomimetic responsive immunoregulatory fiber membranes in response to pH5.8 culture environment released IL-4 plasmid loaded cationic liposomes transfected with mesenchymal stem cells in the lower chamber of Transwell plate, and secreted IL-4 to exert immune regulation on bone marrow macrophages (BMMs) in the upper compartment. Figure 5G, J showed that there was no significant difference in the expression of anti-inflammatory gene (IL-10, TGF- β) among the groups on the first day ($p > 0.05$). We speculate that this stage is in the response phase and cell transfection stage of the material. However, on the 3rd, 5th and 7th day, MSaP-aL/p anti-inflammatory genes were highly expressed, and there was significant difference compared with other groups ($p < 0.05$). The above *in vitro* experimental results supported that the biomimetic composites we constructed could respond positively to the acidic environment and induce BMMs to differentiate into M2.

Figure 5G, J. Evaluation of pro-inflammatory and anti-inflammatory factor gene expression of BMMs in different pH culture environments to simulate the microenvironment of spinal cord injury. The expression of IL-10 mRNA (G), TGF- β mRNA (J) under pH5.8. ($n=3$, all values were mean \pm std. dev., * $p < 0.05$ and ** $p < 0.01$ when comparing MSaP-aL/p and other groups via two-way analysis of variance (ANOVA) with a Tukey's post-hoc test)

- Since approximately 50% of IL-4 plasmid is released in 24h at pH 6.6 (Figure 4 E), how you explain the few differences regarding IL-10 and TGF-b expression at pH 5.8 and pH 6.6 (Figure 6 G and J vs H and K)?

Re : Thank you for the comment. Figure 6 in the original manuscript is now placed in Figure 5 as shown below. The simulated release showed that about 90% loaded IL-4 plasmid cationic liposomes were released under pH 5.8 environment within 24 hours, while the cumulative release under pH6.6 was about 50% (Figure 4E), which had a significant difference in response and release between the two. Therefore, under the condition of co-culture of pH5.8 and pH6.6, there was a significant difference in the concentration of cationic liposomes for transfection of stem cells. According to the experimental results, Figure 5G showed that the expression of IL-10 gene in MSaP-aL/p group increased significantly on the 3rd day (11.59±0.09 fold), 27.02±0.45 fold on the 5th day and 29.30±0.51 fold on the 7th day. In the pH6.6 culture condition, Figure 5H showed that the expression of IL-10 gene was 4.37±0.04, 17.09±0.10 and 22.29±0.04 fold on the 1st, 3rd and 5th day respectively. However, under pH 5.8 culture conditions (Figure 5J), the gene expression of TGF-β was 2.85±0.06, 7.99±0.07, 22.52±0.37, 27.47±0.59 fold on the 1st, 3rd, 5th, and 7th days, and the pH was 6.6, the expression of TGF-β on the 1st, 3rd, 5th, and 7th days (2.60±0.05, 3.64±0.03, 14.24±0.08, 18.57±0.03 fold) had significant differences(Figure 5K). As shown in the figure below, at the same time in the MSaP-aL/p group, the expression levels of the anti-inflammatory genes IL-10 and TGF-β at pH5.8 were significantly higher than the expression levels at pH6.6 ($p<0.05$).

Measurement of IL-10 and TGF-β gene expression in BMM of MSaP-aL/p group in different pH environments

Figure 5. Evaluation of pro-inflammatory and anti-inflammatory factor gene expression of BMMs in different pH culture environments to simulate the microenvironment of spinal cord injury. The expression of (A), (B), (C) IL-1 β mRNA, (D), (E), (F) TNF- α mRNA, (G), (H), (I) IL-10 mRNA and (J), (K), (L) TGF- β mRNA at different pHs of BMMs. (n=3, all values were mean \pm std. dev., * $p < 0.05$ and ** $p < 0.01$ when comparing MSaP-aL/p and other groups via two-way analysis of variance (ANOVA) with a Tukey's post-hoc test)

- The authors claim that the microsol electrospinning allows for a prolonged release of protein molecules (6 weeks) (based on a previous authors' work on chloroquine-based DDS). However, approximately 80% of NGF is released in 25 days (Figure 4 F). Please revise the claims regarding the microsol electrospinning advantages having in consideration the present results. The results should be also presented in terms of the effective amount of protein released over time and also its bioactivity.

Re : Thank you for the comment. We've revised the claim according to your advice. Also, the release of NGF from both quantitative and qualitative aspects was thoroughly characterized.

(1). As described above, the NGF *in vitro* simulated release solution was quantified by ELISA technology at each predetermined time point (Figure 4F).

(2). Currently, we had added the characterization of NGF concerning its biological activity after the release. As shown above, we used fiber membranes release solution from the 30th to the 40th day (the longest period of time) to induce the differentiation of bone marrow

mesenchymal stem cells for five days. As a result, NGF-induced neuron-like could be identified through fluorescently labeled by Tuj-1 (Figure S8), and the results of semi-quantitative analysis also showed significant statistical difference from the negative control group.

- How did the authors acquire the BMSCs? Is the procedure previously approved by the Ethical Committee of the certified institution for animal experimentation? Are those isolated from a single donor or pooled? Did the authors assess BMSCs putative stem cell potential?

Re : Thank you for the comment and reminding. Rat bone marrow mesenchymal stem cells (BMSCs) were provided by the Institute of Orthopaedics of the First Affiliated Hospital of Soochow University, with corresponding extracting procedure certified and approved by the Ethics Committee.

The BMSCs used in the study were isolated from pooled donors. Before employed in *in vitro* assessment, BMSCs were subjected for flow cytometry to identify the surface specific markers and confirm their purity. Flow cytometry method and results are described below.

Method : 3rd generation bone marrow mesenchymal stem cells were collected, after washed with PBS, cells were treated with American Hamster anti-rat-PE CD29 (Invitrogen, 12-0291-81), mouse Brilliant Violet 650TM anti-rat CD90 (Biolegend, 202533), mouse anti-rat-FITC CD34 (Novus, NBP2-47911F) and Alexa Fluor[®] mouse anti-rat CD45 (Biolegend, 202212) flow cytometry antibodies (1: 100). One negative control group and three separate experimental groups were set for each antibody. Putative identification by flow cytometry after incubation at 4 ° C for 30 minutes.

Results: In order to confirm the purity of BMSCs employed for *in vitro* study, the molecular markers on the cell surface were detected by flow cytometry which showed high expression of CD29 and CD90, and low expression of CD34 and CD45, indicating the high purity of bone marrow mesenchymal stem cells as well as the feasibility of employing these cells in following studies¹⁷.

Identification of BMSCs purity by flow cytometry

- The neurogenic activity of the testing condition was assessed by culturing the BMSCs onto the NGF-loaded fibrous scaffolds for 10 days. The authors used qRT-PCR technique to assess the expression of earlier neuron-specific markers in neuron-like cells. To increase the significance of these results, it is suggested to assess the expression of lately expressing neuronal markers.

Re : Thank you for the suggestion. Spinal cord injury repair occurs in a complex pathological environment, the study of biomimetic fiber membranes *in vitro* can only preliminarily simulate the early processing of nerve regeneration *in vivo*. In order to more accurately evaluate the function of biomimetic fibers in promoting nerve regeneration, we have added the *in vivo* immunofluorescence staining of growth associated protein-43 (GAP-43), a lately expressing neuronal marker, 4w and 8w after surgery, which could reflect the neurogenic activity at later stage. The supplemented content in Method and Result & Discussion section were provided as follow ¹⁸:

Method: *As for the immunohistochemical evaluation, the antigen was repaired with 0.3% hydrogen peroxide and nonspecific antigen was blocked by 5% BSA. The primary antibody (rabbit anti GAP-43 (Abcam, ab75810) for axon staining) was added and incubated at 4 °C overnight, and the second antibody (goat anti-rabbit IgG (Alexa Fluor 647, Abcam, ab150079) was incubated at room temperature for 1 hour after PBS washing. The samples were observed and photographed under fluorescence microscope, with the fluorescence intensity analyzed with ImageJ software. The values were represented in the form of average optical intensity, which had been corrected for optical density, de-background, and finally normalized the staining intensity by the number of cell nuclei. A total of five images were randomly selected from each group for statistical analysis (n=5) (Supplementary information, page 17, line 426).*

Results and discussion: *Under normal physiological conditions, low expression of*

GAP-43 only maintains normal function of spinal cord, but GAP-43 is highly expressed in the process of nerve development and regeneration, which can regulate axonal extension, enhance nerve plasticity and release of neurotransmitters (Fig. 10C). Therefore, the fluorescence density of sham group was lower than that of other groups at the same time ($p < 0.05$) (Fig. 10D). Meanwhile, the fluorescence density of GAP-43 in MSaP-aL/p group was significantly higher than that in other control groups, which proved that immunoregulation of local inflammatory response during acute spinal cord injury could promote nerve regeneration for a long time ($p < 0.05$). In addition, aP group was lower than that in the microsol electrospun groups (MSaP, MSaP-aL, MSaP-aL/p) at the same time ($p < 0.05$), suggesting that continuous supply of NGF could significantly promote nerve repair. Surprisingly, with the development of time, the expression of GAP-43 in all experimental groups except sham group at 4 weeks were significantly higher than those at 8 weeks ($p < 0.05$). The results suggested that there is a limited time window for the repair of spinal cord injury. Immune regulation to reduce inflammatory injury in the stage of acute spinal cord injury can better promote nerve regeneration (page 22, line 613).

Figure 10C, D. Immunofluorescence staining of neuron cells and axons. (C), Immunofluorescence staining of axon with (D) its quantification analyzed with optical density. ($n=5$, GAP-43 intensity values were mean \pm std. dev., # $p < 0.05$ when comparing 4w and 8w in blank control groups, aP, MSaP, MSaP-aL and MSaP-aL/p; @ $p < 0.05$ when comparing sham group and other groups; & $p < 0.05$ when comparing aP and microsol electrospun groups (MSaP, MSaP-aL and MSaP-aL/p), * $p < 0.05$ when comparing MSaP-aL/p and other control groups via two-way analysis of variance (ANOVA) with a Tukey's post-hoc test, scale bars were $50\mu\text{m}$)

- During neuronal differentiation, neurite extension and outgrowth occurs. Therefore, it is important to assess the neurite outgrowth by counting the Neurite-bearing cells.

Re : Thank you for the comment. In neurorepair studies, it is necessary to count the number of neurites during neuron regeneration. The literature reports that after induction of spinal dorsal root ganglion (DRG) and PC12, the neurite extension and growth can be clearly

observed in the differentiated neuronal cell morphology^{3, 19}. However, in this study, in order to explore the function of biomimetic fibers in promoting nerve regeneration, bone marrow mesenchymal stem cells were selected as seed cells to differentiate into neuron-like cells induced by NGF. Due to the limitation of cellular characteristics, neuron-like cells differentiated from mesenchymal stem cells are similar to neuronal cells in protein molecule and gene expression, but the morphological changes are not completely consistent. To the end, for better assessment, after the neural differentiation of bone marrow mesenchymal stem cells were induced by microsol electrospinning groups *in vitro*, higher expression of neuron specific marker gene and corresponding protein fluorescence semi-quantitative were obviously observed (Figure 6). *In vivo*, Tuj-1-labeled neurons and GAP-43-labeled axon sprouting were observed at 4 and 8 weeks (Figure 10). Based on the current results of *in vitro* and *in vivo*, it has been preliminarily confirmed that biomimetic composite fibers possess the capacity of continuously promoting nerve regeneration. We will use nervous system-specific cell lines to assess the neurite outgrowth in the further study of the mechanism of biomimetic fibrous membrane promoting nerve regeneration.

- From the results of Figure 8 B and C, it is important to denote the minimum time needed to achieve similar motor function. Please discuss it accordingly.

Re : Thank you for the thoughtful comment. As the reviewer mentioned earlier, to have a clear result, it is necessary to show the best, worst and more frequent experimental results. When the original manuscript was first submitted, we realized that the sample size in the animal motor function scoring part was inappropriate (n=3), so we repeated this part of assay and increased the sample size (n=7) so as to highlight the significance of our study. We are sorry for this mistake. Figure 8 in the original manuscript is placed in Figure 7 and the revised version in different sections and figure caption were provided as follows:

Method: *The recovery of neuromotor function of hindlimb in six groups of rats was evaluated by BBB (Basso, Beattie, Bresnahan) and IPT (inclined plane test) scoring system. The score range of BBB scale was 0 to 21 with 0 indicating no hindlimb movement, and 21 indicating that motor function was normal. During each test, two unwitting examiners evaluated separately each week after the operation. As a supplementary study of BBB scoring test, IPT can improve the effectiveness and sensitivity of scoring. Using Rivlin method, the rats were placed on the inclined plate with rubber pad, the longitudinal axis of the rat was kept parallel to the longitudinal axis of the inclined plate, the head was raised to one side, and raised 5 °from 0 ° within 5 seconds. The highest angle of stay of the rats was recorded and the larger the angle indicated the stronger the load-bearing capacity of the lower extremities. The above two scoring methods are performed by two blinded observers and seven rats were randomly selected from each group at a fixed time point every week for evaluation (n=7) (Supplementary information, page 16, line 405).*

Results and discussion: The *in vivo* performance of fiber bundle (Fig.7A (a, b, c, d)) was evaluated by SD rat T9 spinal cord hemisection model. The recovery of hindlimb motor dysfunction after spinal cord injury was evaluated by BBB and IPT scores every week after operation. The results showed that the hindlimb motor function of rats in each group recovered in varied degrees. Superior recovery of motor function could be identified in MSaP-aL/p group compared with other groups, as indicated by significantly higher BBB score and IPT score at each time points from 4th week post-injury ($p < 0.05$). At the 8th week, the BBB score of 13.71 ± 1.11 , and the IPT score of $55.86 \pm 4.67^\circ$ could be found in MSaP-aL/p group, both of which were significantly higher than those of the other groups ($p < 0.05$) (Fig. 7B, C). In addition, the minimum time required to achieve similar motor functions during recovery is more convincing in characterizing the functionality of neural repair materials. Therefore, in the comparison of 10 points (the highest average score of the negative control group in the BBB score) and 47° (the highest average angle of the negative control group in the IPT test), the two scores of the MSaP-aL/p group exceeded the control values at the 4th week after operation, reaching 11.14 ± 2.04 (BBB) and $52.43 \pm 5.56^\circ$ (IPT) respectively. The results of motor function recovery suggested that immunoregulatory fiber bundle (MSaP-aL/p) implantation could reduce the risk of further damage to surviving motor neurons by effectively inhibiting the acute inflammatory response of spinal cord injury, and promote nerve repair subsequently (page 18, line 482).

Figure 7. (B) Evaluation of motor function recovery of lower limb by BBB score in rats assisted by (C) rats motor function IPT score. ($n=7$, all values were mean \pm std. dev., $*p < 0.05$ when comparing MSaP-aL/p and other groups via two-way analysis of variance (ANOVA) with a Tukey's post-hoc test)

- In Figure 8B, please clarify the significant differences between the conditions MSaP-aL/P and

MSaP-aL. They seem very similar, namely when considering the SD at each time point.

Re : Thank you for the comment. In order to clearly observe the recovery of animal motor function, we've increased the sample size for analysis (n=7), and highlighted the corrected content in the manuscript as described above.

In terms of composition, MSaP-aL/p and MSaP-aL represent loaded or unloaded IL-4 plasmid (Table 1). Functionally, MSaP-aL/p can protect part of surviving motor neurons and inhibit the formation of glial scar tissue by regulating the polarization of infiltrating macrophages and in situ microglia during the acute stage (Figure 7-10), and improve neural progenitor cell effect (Figure S12).

- In Figure 8C, no significant differences were clearly observed. Please revise the statistical analysis, namely testing the normality and variance homogeneity.

Re : Thank you for the comment. Figure 8 in the original manuscript is now placed in Figure 7. In order to observe the statistical differences among groups more clearly, we had reorganized and corrected this part of content as follows.

Figure 7C. rats motor function IPT score. (n=7, all values were mean \pm std. dev., * $p < 0.05$ when comparing MSaP-aL/p and other groups via two-way analysis of variance (ANOVA) with a Tukey's post-hoc test)

- In Figure 9 please provide the results for the same time points (i.e. 4 and 8 weeks), as for other in vivo results (i.e. Figures 10 and 11).

Re : Thank you for the comment. As we've mentioned in the manuscript, the immunofluorescence labeling of IL-4, TNF- α and IL-10 were conducted to evaluate the immunoregulating effect of biomimetic fibers on macrophage polarization at early stage which usually means within 1w after surgery. Similar set of time point on this topic could be found in previous literature concerning immunoregulation^{16, 20}. While the later time points (4w and 8w) for the labeling of activated astrocyte (GFAP) and glial scar tissue (NG2) were employed to observe the long-term effect after regeneration.

(1). In the original manuscript, Figure 9 was the result of evaluating the immunomodulatory function of biomimetic fiber bundles during acute spinal cord injury, so immunofluorescence labeled IL-4, TNF- α (M1 specific marker) and IL-10 (M2 specific marker) in spinal cord specimens 1 week after operation.

(2). Biomimetic fiber bundles will produce long-term biological effects after reducing inflammatory response in the period of acute spinal cord injury. In the previous manuscript, Figure 10A and Figure 10B were activated astrocyte (GFAP) and glial scar tissue (NG2) labeled by immunofluorescence at the same time point (4 weeks and 8 weeks), respectively, and were separated by black dotted lines at different time points. In addition, in order to clearly show the fluorescence quantitative difference among groups at the same time point, we draw the results of each experimental group separately according to the time point, such as the abscissa display in Figure 10C and Figure 10D ("4w" and "8w"). In order to show the results at the same point more clearly, we have made improvements as follows and the original Figure 10 was placed in Figure S11 in the revised manuscript:

Figure S11. Immunofluorescence staining of activated astrocytes and glial scar. (A), (B) Immunofluorescence staining of astrocytes and quantitatively analyzed with optical density

along with (C), (D) their quantification with optical density. ($n=5$, all values were mean \pm std. dev., $*p<0.05$ when comparing MSaP-aL/p and other control groups via two-way analysis of variance (ANOVA) with a Tukey's post-hoc test, $^{\#}p<0.05$ when comparing material transplantation groups and blank control group via two-way analysis of variance (ANOVA) with a Tukey's post-hoc test, scale bars were $50\mu\text{m}$)

(3). Activated astrocytes and scar tissue after spinal cord injury will directly affect nerve regeneration. Following the long-term biological effects of the immune regulation of the above-mentioned fiber bundles, we labeled the migrated neural progenitor cells and differentiated neuron-like neurons at the injured site at the same time point, as shown in Figure S12 and Figure 10A, B. In order to observe the results at the same time more clearly, we have also improved it.

Figure S12. Immunofluorescence staining of neural progenitor cells. (A), Immunofluorescence staining of neural progenitor cells with (B) its quantification analyzed with optical density. ($n=5$, intensity values were mean \pm std. dev., $^{\#}p<0.05$ when comparing sham group and other groups and $*p<0.05$ when comparing MSaP-aL/p and other control groups via two-way analysis of variance (ANOVA) with a Tukey's post-hoc test, scale bars were $50\mu\text{m}$)

Figure 10A, B. (A), Immunofluorescence staining of neuron cells and (B) quantification analyzed with optical density. ($n=5$, Tuj-1 intensity values were mean \pm std. dev., % $p < 0.05$ when comparing blank control and microsolv electrospun groups (MSaP, MSaP-aL and MSaP-aL/p); & $p < 0.05$ when comparing aP and microsolv electrospun groups (MSaP, MSaP-aL and MSaP-aL/p); @ $p < 0.05$ when comparing 4w and 8w in MSaP; \$ $p < 0.05$ when comparing 4w and 8w in MSaP-aL; and # $p < 0.05$ when comparing 4w and 8w in MSaP-aL/p; * $p < 0.05$ when comparing MSaP-aL/p and other control groups via two-way analysis of variance (ANOVA) with a Tukey's post-hoc test, scale bars were 50 μ m)

- The authors assess the pro-inflammatory (IL-1 β , TNF- α and IL-10) and anti-inflammatory (TGF- β) cytokines expression of BMM in *in vitro* assays. However, the authors only assessed the pro-inflammatory cytokines TNF- α and IL-10 *in vivo*. Please be consistent on the presentation of results along the manuscript.

Re : Thank you for raising the key point. The characterization of pro-inflammatory and anti-inflammatory biological effects *in vitro* and *in vivo* are very important to evaluate the immunomodulatory function of biomimetic fiber membrane. TNF- α and IL-10 are specific secretions of M1 and M2 microglia/macrophages respectively ⁵. As usual, IL-10 is regarded as the anti-inflammatory cytokine. Therefore, in the previous manuscript, Figure 9A listed the representative images of TNF- α and IL-10 in spinal cord specimens of rats in each group with immunofluorescence double labeling one week after operation, as well as the results of semi-quantitative analysis (Figure 9C). The results showed that the fluorescence density of IL-10 in MSaP-aL/p group was significantly higher than that in other groups ($p < 0.05$), while the fluorescence density of TNF- α was significantly lower ($p < 0.05$). Combined with the experimental results *in vitro* and *in vivo*, it was confirmed that the biomimetic composite fiber membrane plays an immunoregulatory function in response to the local acidic microenvironment.

Figure 9A-C. Evaluation of immune regulation 7 days after operation. (A) IL-4 immunofluorescence single labeling and TNF- α , IL-10 immunofluorescence double labeling. Quantitatively analysis of (B) IL-4 immunofluorescence, (C) IL-10 immunofluorescence and TNF- α immunofluorescence. ($n=5$, all values were mean \pm std. dev., $*p<0.05$ when comparing MSaP-aL/p and other groups via two-way analysis of variance (ANOVA) with a Tukey's post-hoc test, scale bars were $50\mu\text{m}$)

- The Figure 9 quantitative results are based on immunofluorescence images. Following the same presentation approach depicted in Figure 7, please provide mRNA expression results for the in vivo assay.

Re : Thank you for the comment. In order to ensure the consistency of the experimental results *in vivo* and *in vitro*, we added qRT-PCR to evaluate the expression of inflammatory genes in spinal cord specimens of animals one week after operation. The methods and results of gene detection in animal samples are as described above. The results showed that the implantation of biomimetic immunomodulatory fiber bundle could significantly inhibit the expression of pro-inflammatory gene and promote the expression of anti-inflammatory gene.

Figure 9D-H. Spinal cord specimens of IL-4 (D), TNF- α (E), IL-1 β (F), IL-10 (G) and TGF- β (H) gene expressions were assessed by qRT-PCR. ($n=3$, qRT-PCR values were

*mean ± std. dev., * $p < 0.05$ when comparing MSaP-aL/p and other groups via one-way analysis of variance (ANOVA) with a Tukey's post-hoc test)*

-For all the immunofluorescence images quantification, it is suggested to normalize the staining intensity by the number of cells nuclei.

Re : Thank you for the comment. As recommended, we have recalibrated all semi-quantitative analyses of immunofluorescence images and corrected them in the manuscript.

- The authors should refer previous works in which liposomes carrying plasmids were immobilised at the surface of electrospun meshes.

Re : Thank you for the comment.

We searched the literatures based on the keywords "electrospun meshes", "liposomes" and "plasmids" in the comment. Three representative literatures were selected for comparison and discussion.

(1). Monteiro, N. et al., Antibacterial activity of chitosan nanofiber meshes with liposomes immobilized releasing gentamicin. *Acta Biomater.* 18. 196-205 (2015).

In the first article, gentamicin-loaded liposomes were grafted on the surface of chitosan fiber membrane with Maleimide, which proved to have significant antibacterial activity against *Escherichia coli*, *Pseudomonas aeruginosa* and *Staphylococcus aureus*. It was pointed out that it was feasible and advantageous for liposomes to be immobilized on the surface of electrospun meshes to construct nanostructured composite delivery system.

(2). Monteiro, N. et al., Instructive nanofibrous scaffold comprising runt-related transcription factor 2 gene delivery for bone tissue engineering. *ACS Nano.* 8, 8082-8094 (2014).

In the second article, the liposome loaded with RUNX2 plasmid was immobilized on the surface of polycaprolactone fiber membrane to promote bone gene expression for a long time, which was helpful to enhance the tissue regeneration induction of biomaterials.

(3). He, S. H. et al., Multiple release of polyplexes of plasmids VEGF and bFGF from electrospun fibrous scaffolds towards regeneration of mature blood vessels. *Acta Biomater.* 8. 2659-2669 (2012).

In the third article, basic fibroblast growth factor plasmid (pbFGF) and vascular endothelial growth factor plasmid (pVEGF) were loaded into the core-shell structure of

emulsion electrospun fiber membrane at the same time. After 4 weeks of sustained release, it was proved that the multiple delivery strategy of biomaterials could significantly promote tissue regeneration.

Corresponding discussion regarding these representative literatures were supplemented in the manuscript and provided as follow:

Studies have shown that combining electrospinning with cationic liposomes would be more effective in antibacterial, inducing bone tissue regeneration and promoting vascular repair^{21, 22, 23}. However, few studies were reported to employ similar strategy to simultaneously control the inflammatory response of acute spinal cord injury and promote nerve regeneration continuously (page 4, line 91).

Reviewer #2 (Remarks to the Author):

The authors provide a refined strategy to target both immune reaction and neural regeneration in spinal cord injuries. The approach is interesting and rather solid.

I recommend to address the following points to facilitate the comprehension of the paper as well as to strengthen the obtained results.

Thank you for your thoughtful reading of our manuscript as well as your constructive comments. We have carefully revised the manuscript, with some new experiments supplemented to ensure the integrity of all the conclusions of this study. Detailed point-by-point responses are provided below.

A lot of the data are described in the manuscript, but I am afraid that 12 figures and 28 pages of manuscript are way too many for a communication. The authors should condense data presentation and panels: some of less significant data could be moved to the supplementary material section.

Re : Thank you for the comment. We considered to place some figures into the supplementary information. There were some changes in these figures as follows:

(1). Figure 5 in the original manuscript is now placed in Figure S4.

Figure S4. Cell survival and proliferation assays of membranes. (A) Live/dead (green/red) fluorescence staining images. (all scale bars were 1000 μ m) (B) Fluorescence semi-quantitative analysis of living cells. (n=3, fluorescence density values were mean \pm std. dev., * p <0.05 when comparing control group and four other groups via one-way analysis of variance (ANOVA) with a Tukey's post-hoc test., ns, not significant) (C) Detection of cell proliferation by CCK8 kit at 1, 3 and 5 days. (n=6, absorbance values were mean \pm std. dev., * p <0.05 when comparing control group and four other groups via two-way analysis of variance (ANOVA) with a Tukey's post-hoc test., ns, not significant)

(2). Figure 10 in the original manuscript is now placed in Figure S11

Figure S11. Immunofluorescence staining of activated astrocytes and glial scar. (A), (B) Immunofluorescence staining of astrocytes and quantitatively analyzed with optical density along with (C), (D) their quantification with optical density. ($n=5$, all values were mean \pm std. dev., $*p<0.05$ when comparing MSaP-aL/p and other control groups via two-way analysis of variance (ANOVA) with a Tukey's post-hoc test, $^{\#}p<0.05$ when comparing material transplantation groups and blank control group via two-way analysis of variance (ANOVA) with a Tukey's post-hoc test, scale bars were $50\mu\text{m}$)

(3). The Figure 11A of the original manuscript is placed independently in Figure S12.

Figure S12. Immunofluorescence staining of neural stem cells and neuron cells. (A), Immunofluorescence staining of neural stem cells with (B) its quantification analyzed with optical density. ($n=5$, Nestin intensity values were mean \pm std. dev., $^{\#}p<0.05$ when comparing sham group and other groups and $*p<0.05$ when comparing MSaP-aL/p and other control groups via two-way analysis of variance (ANOVA) with a Tukey's post-hoc test, scale bars were $50\mu\text{m}$)

(4). Figure 12 in the original manuscript is now in Figure S13, and VWF fluorescently labeled neovascularization was added to this section.

Figure S13. (A), Immunofluorescence staining of vascular endothelial cells and (B) their quantitative analysis with optical density. (C), Immunofluorescence staining of neovascularization and (D) their quantitative analysis with optical density. (n=5, all values were mean \pm std. dev., * $p < 0.05$ when comparing MSaP-aL/p and other control groups at the same time and # $p < 0.05$ when comparing 4w and 8w in MSaP-aL/p via two-way analysis of variance (ANOVA) with a Tukey's post-hoc test, scale bars were 50 μ m)

For a better understanding of the impact of their work the authors should state clearly that the present work deals with acute spinal cord injuries.

Re : Thank you for the thoughtful suggestion. We have made corresponding changes in the article to highlight the main purpose of this study is acute spinal cord injury. The specific modifications are as follows.

(1). **Abstract:** *The strategies concerning modification of complex immune pathological inflammatory environment during acute spinal cord injury remain oversimplified and superficial. Inspired by the acidic microenvironment at acute injury site, a functional pH-responsive immunoregulation-assisted neural regeneration strategy was constructed. With the capability of directly responding to acidic microenvironment at focal area*

followed by triggered release of the IL-4 plasmid-loaded liposomes within few hours to suppress the release of inflammatory cytokines and promote neural differentiation of mesenchymal stem cells in vitro, the microenvironment-responsive immunoregulatory electrospun fibers were implanted into acute spinal cord injury rats. Together with slow and sustained release of NGF achieved by microsol core-shell structure, the immunological fiber scaffolds were revealed to bring significantly shifted immune cells subtype to down-regulate acute inflammation response, reduce scar tissue formation, promote angiogenesis as well as neural differentiation at injury site, and enhance functional recovery in vivo. Overall, this strategy provided a novel delivery system through microenvironment-responsive immunological regulation effect so as to break through the current dilemma from the contradiction between immune response and nerve regeneration, which providing an alternative for the treatment of acute spinal cord injury. (page 2, line 24).

(2). Introduction: In the stage of acute spinal cord injury, the balance between M1 and M2 subtypes could be one of the key factors in determining the prognosis of spinal cord injury (page 3, line 56).

In the study of conventional tissue engineering using biomaterials to transmit biological information to promote neural differentiation of stem cells, nerve repair is at risk of failure under the influence of severe immune inflammation exerted by acute spinal cord injury. Therefore, the construction of the bioinspired material that can rapidly respond to local microenvironment during the acute stage and accurately regulate macrophages and microglia cells polarization to reduce inflammatory response, and provide sustained neurogenic platform for stem cells in the later stage may break through the bottleneck of the current treatment of acute spinal cord injury (page 3, line 60).

In this work, inspired by the inflammation and acid-enriched feature of the microenvironment after acute spinal cord injury, a biomimetic fiber scaffold with both immunoregulation function and neurogenic potential was designed to satisfy the specific demand during the acute stage suppression of inflammation and later-stage neural regeneration in scenario of spinal cord injury. (page 5, line 117).

(3). Conclusion: The responsive fiber scaffold was endowed with capability of local immune regulation which could respond to the acid microenvironment after the acute spinal cord injury, as well as the sustained promoting effect on neural tissue regeneration (page 23, line 651).

Line 568: there is actually no trustable specific marker for NSC in vivo, Nestin can provide hints of PROGENITOR cells, but not neural stem cells. Tuj-1 marks immature neurons. Therefore I would remove any statement regarding endogenous stem cell mobilization and I will focus on nervous regeneration with newly formed nervous fibers.

Re : Thank you for the thoughtful suggestion. We have removed the content of endogenous

stem cell mobilization and made the following changes in the revised manuscript.

(1). **2.4 In vitro biological characteristics of nerve fiber membranes.** *In this study, BMSCs and bone marrow macrophages (BMMs) were co-cultured through Transwell to simulate the effects of fiber membranes on cell-to-cell interaction in the microenvironment of acute spinal cord injury, and to evaluate the biological characteristics of nerve fiber membranes (page 13, line 337).*

(2). **2.5.4 Histology and immunohistochemical evaluation of spinal cord repair in vivo.** *Nestin, Tuj-1 and GAP-43 were used to label neural progenitor cells, neuron cells and axonal sprouting respectively to evaluate the ability of biomimetic bundles to promote nerve regeneration in the middle and late stage of spinal cord injury. The distribution of nestin-labeled neural progenitor cells at the injured site (Fig. S12A) was observed in each bundle group at 4 and 8 weeks after injury. The neural progenitor cells in the sham group was lower than the other groups, which may be due to the low proliferation activity of spinal ependymal cells and the lack of immune response to nestin protein ($p < 0.05$), and the fluorescence quantitative display (Fig. S12B) at the same time point in the MSaP-aL/p group was significantly higher than that in the other control groups ($p < 0.05$). The results showed that biomimetic bundle could significantly inhibit the blocking effect of surrounding scar tissue on neural progenitor cells through immunomodulatory effect. At 4w and 8w time points, neural progenitor cells were continuously promoted by NGF released from microsol electrospun fibers (MSaP, MSaP-aL and MSaP-aL/p), and a large number of Tuj-1-labeled neurons appeared as shown in Fig. 10A (page 21, line 587).*

Line 595: I would recommend to add stainings against GAP-43 and SMI31 to better strengthen the findings about the neurogenic potential of the scaffold.

Re : Thank you for the suggestion.

(1). GAP-43 and SMI31 play an indispensable role in nervous system research. Growth associated protein-43 (GAP-43) is a major component of the tips of elongating axon, as well as a suitable marker for labeling axonal sprouting, playing an important role in nerve regeneration²⁴. At present, we supplied GAP-43 fluorescence labeling to visualize regenerated nerve axon sprouting *in vivo*. Corresponding supplemented content in **Method** and **Results and discussion** section was provided as follows:

Method: *As for the immunohistochemical evaluation, the antigen was repaired with 0.3% hydrogen peroxide and nonspecific antigen was blocked by 5% BSA. The primary antibody (rabbit anti GAP-43 (Abcam, ab75810) for axon staining) was added and incubated at 4 °C overnight, and the second antibody (goat anti-rabbit IgG (Alexa Fluor 647, Abcam, ab150079) was incubated at room temperature for 1 hour after PBS washing. The samples were observed and photographed under fluorescence microscope, with the fluorescence*

intensity analyzed with ImageJ software. The values were represented in the form of average optical intensity, which had been corrected for optical density, de-background, and finally normalized the staining intensity by the number of cell nuclei. A total of five images were randomly selected from each group for statistical analysis (n=5) (Supplementary information, page 17, line 426).

Results and discussion: Under normal physiological conditions, low expression of GAP-43 only maintains normal function of spinal cord, but GAP-43 is highly expressed in the process of nerve development and regeneration, which can regulate axonal extension, enhance nerve plasticity and release of neurotransmitters (Fig. 10C). Therefore, the fluorescence density of sham group was lower than that of other groups at the same time ($p<0.05$) (Fig. 10D). Meanwhile, the fluorescence density of GAP-43 in MSaP-aL/p group was significantly higher than that in other control groups, which proved that immunoregulation of local inflammatory response during acute spinal cord injury could promote nerve regeneration for a long time ($p<0.05$). In addition, aP group was lower than that in the microsol electrospun groups (MSaP, MSaP-aL, MSaP-aL/p) at the same time ($p<0.05$), suggesting that continuous supply of NGF could significantly promote nerve repair. Surprisingly, with the development of time, the expression of GAP-43 in all experimental groups except sham group at 4 weeks were significantly higher than those at 8 weeks ($p<0.05$). The results suggested that there is a limited time window for the repair of spinal cord injury. Immune regulation to reduce inflammatory injury in the stage of acute spinal cord injury can better promote nerve regeneration (page 22, line 613).

Figure 10C, D. Immunofluorescence staining of neuron cells and axons. (C), Immunofluorescence staining of axon with (D) its quantification analyzed with optical density. (n=5, GAP-43 intensity values were mean \pm std. dev., # $p<0.05$ when comparing 4w and 8w in blank control groups, aP, MSaP, MSaP-aL and MSaP-aL/p; @ $p<0.05$ when comparing sham group and other groups; & $p<0.05$ when comparing aP and microsol electrospun groups (MSaP, MSaP-aL and MSaP-aL/p), * $p<0.05$ when comparing MSaP-aL/p and other control groups via two-way analysis of variance (ANOVA) with a Tukey's post-hoc test, scale bars were 50 μ m)

(2). SMI31 reacts with a phosphorylated epitope in extensively phosphorylated neurofilament H, also serves as an evaluation criteria in studies of degenerative diseases such as multiple sclerosis (MS)²⁵, Alzheimer's disease (AD)²⁶ and stroke²⁷. Despite that

SMI31 was absent in current study, the NF-H protein in *in vitro* differentiated neuron-like axons was fluorescently labeled with its gene expression quantified by qRT-PCR in this study (Figure 6, in the red box below) which could also strengthen our result on neurogenic activity. Further study looking into the mechanism of biomimetic fiber membrane on promoting nerve regeneration would employ SMI31 for better characterization.

Figure 6. Neuron-specific marker immunofluorescence staining and gene detection assays of scaffolds. (A) Immunofluorescence staining of Tuj-1, NSE, NF-200, Tau, staining images on different fiber scaffolds with (B), (C), (D), (E) Corresponding fluorescence semi-quantitative analysis. (F), (G), (H), (I) mRNA expression analysis of Tuj-1, NSE, NF-200, Tau. (n=3, all values were mean \pm std. dev., * $p < 0.05$ when comparing microsol electrospun groups (MSaP, MSaP-aL, MSaP-aL/p) and control, aP group respectively via one-way analysis of variance (ANOVA) with a Tukey's post-hoc test., ns, not significant, scale bars were 100 μ m)

Line 601: I would add stainings against VWF to strengthen this result about neo-angiogenesis too.

Re : Thank you for the comment. We have added VWF labeled neovascularization research content *in vivo*, and made corresponding changes and highlighted in the revised manuscript.

Method: The primary antibody included rabbit anti-Tuj-1(Abcam, ab18207) for neuron staining, goat anti-nestin (R&D SYSTEMS, AF2736) for neural progenitor cells staining, mouse anti-GFAP (Servicebio, GB12096) for staining of astrocytes, mouse anti-NG2 (Abcam, ab50009) for staining of glial scars, mouse anti-CD31 (Servicebio, GB12063) for

staining of vascular endothelial cells , rabbit anti-VWF (Abcam, ab6994) for staining of neovascularization. Second antibody includes goat anti-rabbit IgG (Cy3, Servicebio, GB21303), goat anti-mouse IgG (Alexa Fluor 488, Servicebio, GB25301) and goat anti-rabbit IgG (Alexa Fluor 647, Abcam, ab150079) (Supplementary information, page 17, line 428).

Results and discussion: Coincidentally, the quantitative analysis of neovascularization labeled by Von Willebrand Factor (VWF) fluorescence showed that the intensity in MSaP-aL/p group was significantly higher than that in other control groups at the same time ($p < 0.05$) (Fig. S13C, D). Similarly, the number of neovascularization labeled by VWF at 4 weeks was significantly less than that at 8 weeks ($p < 0.05$), which indicated that biomimetic bundle could not only reduce the invasion of local inflammatory reaction, but also induce macrophages and microglia cells to be polarized to M2 subtype, which could secrete vascular endothelial growth factor (VEGF) and promote blood vessel formation, providing a superior platform for tissue regeneration for a long time (page 23, line 638).

Figure S13C, D. (C), Immunofluorescence staining of neovascularization and (D) their quantitative analysis with optical density. ($n=5$, all values were mean \pm std. dev., * $p < 0.05$ when comparing MSaP-aL/p and other control groups at the same time and # $p < 0.05$ when comparing 4w and 8w in MSaP-aL/p via two-way analysis of variance (ANOVA) with a Tukey's post-hoc test, scale bars were 50 μ m)

Figure 8: Behavioral recovery data are significant but of limited magnitude if compared to control groups.

Re : Thank you for the comment. The recovery of animal motor function is an important indicator for evaluating nerve repair. Sufficient sample size is the basis for ensuring the accuracy of the experimental results. However, I'm sorry that we had made some mistake

on setting sample sizes for scoring, since limited sample size (n=3) led to the inaccuracy of the comparison. At present, we increased the sample size (n=7) for re-statistical analysis. corresponding corrections were made and provided as follow:

Method: *The recovery of neuromotor function of hindlimb in six groups of rats was evaluated by BBB (Basso, Beattie, Bresnahan) and IPT (inclined plane test) scoring system. The score range of BBB scale was 0 to 21 with 0 indicating no hindlimb movement, and 21 indicating that motor function was normal. During each test, two unwitting examiners evaluated separately each week after the operation. As a supplementary study of BBB scoring test, IPT can improve the effectiveness and sensitivity of scoring. Using Rivlin method, the rats were placed on the inclined plate with rubber pad, the longitudinal axis of the rat was kept parallel to the longitudinal axis of the inclined plate, the head was raised to one side, and raised 5 °from 0 ° within 5 seconds. The highest angle of stay of the rats was recorded and the larger the angle indicated the stronger the load-bearing capacity of the lower extremities. The above two scoring methods are performed by two blinded observers and seven rats were randomly selected from each group at a fixed time point every week for evaluation (n=7) (Supplementary information, page 16, line 405).*

Results and discussion: *The in vivo performance of fiber bundle (Fig.7A (a, b, c, d) was evaluated by SD rat T9 spinal cord hemisection model. The recovery of hindlimb motor dysfunction after spinal cord injury was evaluated by BBB and IPT scores every week after operation. The results showed that the hindlimb motor function of rats in each group recovered in varied degrees. Superior recovery of motor function could be identified in MSaP-aL/p group compared with other groups, as indicated by significantly higher BBB score and IPT score at each time points from 4th week post-injury ($p<0.05$). At the 8th week, the BBB score of 13.71 ± 1.11 , and the IPT score of $55.86\pm 4.67^\circ$ could be found in MSaP-aL/p group, both of which were significantly higher than those of the other groups ($p<0.05$) (Fig. 7B, C). In addition, the minimum time required to achieve similar motor functions during recovery is more convincing in characterizing the functionality of neural repair materials. Therefore, in the comparison of 10 points (the highest average score of the negative control group in the BBB score) and 47° (the highest average angle of the negative control group in the IPT test), the two scores of the MSaP-aL/p group exceeded the control values at the 4th week after operation, reaching 11.14 ± 2.04 (BBB) and $52.43\pm 5.56^\circ$ (IPT) respectively. The results of motor function recovery suggested that immunoregulatory fiber bundle (MSaP-aL/p) implantation could reduce the risk of further damage to surviving motor neurons by effectively inhibiting the acute inflammatory response of spinal cord injury, and promote nerve repair subsequently (page 18, line 482).*

Figure 7B, C. (B) Evaluation of motor function recovery of lower limb by BBB score in rats assisted by (C) rats motor function IPT score. ($n=7$, all values were mean \pm std. dev., $*p < 0.05$ when comparing MSaP-aL/p and other groups via two-way analysis of variance (ANOVA) with a Tukey's post-hoc test)

Figure 11: I recommend to substitute “NSC” with neural progenitor cells

Re : Thank you for the excellent suggestion. We have changed the “NSC” involved in the manuscript to “neural progenitor cells”, and highlighted the changes in the manuscript.

Reviewer #3 (Remarks to the Author):

The manuscript by Xi et al. entitled " Microenvironment-Responsive Immunoregulatory Electrospun Fibers for Promoting Nerve Regeneration " presents results suggesting that treatment with a pH-responsive strategy based on the grafting of interleukin-4 (IL-4) plasmid-loaded and aldehydes-modified cationic liposome onto the nerve growth factor (NGF)-loaded electrospun fibrous scaffolds leads to immunomodulation and improves functional recovery and tissue repair after spinal cord injury.

Great thanks to the reviewer for your thoughtful reading of our manuscript. We have completely revised the original manuscript based on your constructive advices. Detailed point-by-point responses are provided below.

Points

1. The authors do not provide any clear evidence demonstrating that this strategy modulates inflammation. This is of key importance since their rationality to boost IL-4 levels after spinal cord injury was to modulate the inflammatory response. Were macrophage and microglia numbers

changed by the treatment? Did the treatment shift microglia and macrophage polarization towards an M2-like phenotype *in vivo*? These are key questions that the authors should have addressed. In addition, this must be assessed by using flow cytometry. This is the only methodology that allows to distinguish microglia from macrophages *in vivo*, since some new specific markers for microglia, such as TMEM119 or P2Y12, only discriminate microglia from macrophages in physiological but not in pathological conditions. The authors just showed that the treatment modulated cytokine levels in the spinal cord by using immunofluorescence. However, this is not a reliable technique to assess cytokine levels since these antibodies bind to various unspecific proteins. For these reasons, cytokine levels have to be assessed by ELISA (not WB) or Luminex assays. The authors need also to demonstrate that IL-4 protein levels are really increased after the treatment, since this therapeutic approach is based, in part, on increasing the levels of this immunomodulatory cytokine. Again, this needs to be done by ELISA or Luminex assays.

Re : Thank you for the thoughtful suggestion. According to your suggestion, we are deeply aware of the deficiency of immunomodulatory demonstration in *in vivo* research. In order to fix this significant problem, corresponding characterizations were supplemented. Both flow cytometry and ELISA tests were conducted 1w after implantation to evaluate the phenotypic changes of microglia and macrophages at the injured site *in vivo*. In addition, the changes in the serum level of inflammatory factors (IL-4, TNF- α , IL-1 β , IL-10, TGF- β) were also measured as described below to strengthen our results. Corresponding information was supplemented in the manuscript and provided as follows:

(1). Flow Cytometry method: 6.3.1 Identification of microglia/macrophages phenotype. After euthanasia, the thoracic cavity was cut open with the left ventricle and inferior vena cava fully exposed. 100ml 0.9% saline was perfused into the heart. Spinal cord samples for flow cytometry detection following the previous article¹⁶. Briefly, a 5mm section of the injured spinal cord tissue centered at the epicenter of the injury site was harvested and was immediately cut into small fragments and mechanically separated by a 100 μ m cell strainer. The cell suspensions of different groups were centrifuged at 300g for 10 minutes at 4 °C, then the antibodies were added and incubated at 4 °C for 30 minutes and fixed in 1% paraformaldehyde. The following rat conjugated antibodies were used: mouse anti-CD11b-APC (BD PharmingenTM, 562102), mouse anti-CD86-FITC (BD PharmingenTM, 561961), mouse anti-CD206-PE (Santa Cruz, sc-58986). Cells were analyzed on a flow cytometer (Merk Millipore, USA) and the results were analyzed by FlowJo7.6 software. For analysis, cells were first gated for CD11b to ensure that only microglia/macrophages were selected, then the following combination of specific markers were used to identify M1 (CD86) and M2 (CD206). In the assay, three samples were randomly selected from each group for detection (n=3) (Supplementary information, page 14, line 350).

Results and discussion: Macrophages and microglia reached the recruitment peak within 7 days after spinal cord injury, so immunomodulatory bundles should play the role of biological response regulation during this period. On the seventh day after the implantation of biomimetic immunomodulatory fiber bundles, we first tested whether they could guide changes in local immune cell subtypes (Fig. 8A). There was no significant

difference in the number of CD11b/CD86-positive macrophages between the non-immunomodulatory fiber bundle groups (aP, MSaP, MSaP-aL) and the blank control group ($p>0.05$) (Fig. 8B). However, due to the large porosity of oriented electrospinning fiber bundles, it could lead to the increase of CD11b/CD206-positive macrophages ($p<0.05$) (Fig. 8C)²⁸. In addition, the *in vivo* fiber bundles regulation of macrophage polarization was surprisingly consistent with the conclusion drawn in the *in vitro* study that MSaP-aL/p group not only significantly decreased the proportion of CD11b/CD86-positive macrophages, but also significantly increased CD11b/CD206-positive macrophages ($p<0.05$). The results of flow cytometry analysis showed that the implantation of immunomodulatory fiber bundles could promote the polarization of local microglia/macrophages to M2 phenotype *in vivo* (page 18, line 503).

Figure 8. Phenotypic identification of immune cells *in vivo*. (A), Flow cytometry analysis of CD11b/CD86-positive cells with (B) its quantification analyzed of the ratio and CD11b/CD206-positive cells with (C) its quantification analyzed of the ratio. ($n=3$, all values were mean \pm std. dev., # $p<0.05$ when comparing non-immunoregulatory fiber groups and blank control group and * $p<0.05$ when comparing MSaP-aL/p and other control groups via one-way analysis of variance (ANOVA) with a Tukey's post-hoc test., ns, not significant)

(2). **ELISA method:** The local severe inflammatory reaction of spinal cord injury produces a large number of immune factors into the systemic circulation with the blood. We used the way of clinical detection of inflammatory factors to initially evaluate the changes of inflammatory response after fiber bundles implantation. Briefly, three animals were randomly selected from each group, and the blood was collected with a centrifuge tube containing coagulant during cardiac perfusion ($n=3$). The blood was placed at room temperature for 30 minutes and then centrifuged at 3000rpm for 5 minutes. The upper serum was detected by ELISA kit (Multi Sciences, China) (Supplementary information, page 16, line 395).

Results and discussion: *The systemic inflammatory response of animals was evaluated by ELISA detection of serum to comprehensively assess the immunoregulatory function of fiber bundles. Due to the large amount of IL-4 secreted into the systemic circulation at the injury site, the serum IL-4 content was significantly increased in the MSaP-aL/p group compared with other groups ($p < 0.05$) (Fig. 9I). The results of pro-inflammatory and anti-inflammatory cytokines showed that, on the one hand, the serum inflammatory factors TNF- α and IL-1 β in the blank control, ap, MSaP, MSaP-aL groups were significantly higher than those in the MSaP-aL/p group ($p < 0.05$) (Fig. 9J, K). On the other hand, MSaP-aL/p group had higher levels of IL-10 and TGF- β ($p < 0.05$) (Fig. 9L, M), which highly suggested that the implantation of immune-regulating fiber bundles can not only improve the local immune environment, but also reduce systemic inflammation (page 20, line 543).*

Serum samples after centrifugation and **Figure 9I-M**. Serum levels of IL-4 (I), TNF- α (J), IL-1 β (K), IL-10 (L), and TGF- β (M) were assessed by ELISA. ($n=3$, ELISA values were mean \pm std. dev., * $p < 0.05$ when comparing MSaP-aL/p and other groups via one-way analysis of variance (ANOVA) with a Tukey's post-hoc test)

2. The title of the manuscript includes the words “Promoting Nerve Regeneration”. This is quite surprising since the author do not demonstrate that their experimental approach leads to axon regeneration. This is very important and needs to be assessed by the authors by means of tracer, viral plasmids or transgenic animals.

Re : Thank you for the comment. We hoped that by constructing a responsive composite fiber membrane, it could regulate the inflammation response of acute spinal cord injury and create a suitable microenvironment for nerve regeneration. Axon regeneration is an important criterion for evaluating the results of nerve repair, which we had been following during our research. Hence, for better assessment of axon regeneration, immunofluorescent

labeling of GAP-43, a marker for axon sprouting, was conducted *in vivo*, with MSaP-aL/p showed the highest level of GAP-43 expression. Corresponding results were provided as follow (Figure 10C, D). Apart from that, preliminarily evaluation on the ability of composite biological scaffolds to promote nerve regeneration were conducted using four kinds of nerve specific markers to label newborn neurons *in vitro*. Among them, β -III tubulin (Tuj-1) is an important component of neuronal skeleton and one of the markers of immature neurons²⁹. Neuronal specific enolase (NSE) is a cytosolic protein consistently expressed by neurons and cells of neuronal origin which is a specific marker used to label neurons in tissue engineering³⁰. Neurofilament H (NF-200) provides structural support for axons and regulates axon diameter³¹. Tau protein is expressed at the distal portions of the axons and can be involved in regulating stability of the cytoskeleton and providing axonal flexibility³². It has been confirmed from the results that the NGF-loaded microsol electrospinning groups (MSaP, MSaP-aL, MSaP-aL/p) can promote the neural differentiation of stem cells (Figure 6). *In vivo*, using nestin-labeled neural progenitor cells to assess immune regulation to inhibit glial scar formation and promote migration of neural progenitor cells (Figure S12). Tuj-1 was employed to label neuron cells, suggesting that the fibrous membrane had a sustained nerve regeneration function (Figure 10). Hence, we believed that the results *in vitro* and *in vivo* support the conclusion that biomimetic fiber membrane promotes nerve regeneration.

GAP-43 staining method: *As for the immunohistochemical evaluation, the antigen was repaired with 0.3% hydrogen peroxide and nonspecific antigen was blocked by 5% BSA. The primary antibody (rabbit anti GAP-43 (Abcam, ab75810) for axon staining) was added and incubated at 4 °C overnight, and the second antibody (goat anti-rabbit IgG (Alexa Fluor 647, Abcam, ab150079) was incubated at room temperature for 1 hour after PBS washing. The samples were observed and photographed under fluorescence microscope, with the fluorescence intensity analyzed with ImageJ software. The values were represented in the form of average optical intensity, which had been corrected for optical density, de-background, and finally normalized the staining intensity by the number of cell nuclei. A total of five images were randomly selected from each group for statistical analysis (n=5) (Supplementary information, page 17, line 426).*

Results and discussion: *Under normal physiological conditions, low expression of GAP-43 only maintains normal function of spinal cord, but GAP-43 is highly expressed in the process of nerve development and regeneration, which can regulate axonal extension, enhance nerve plasticity and release of neurotransmitters (Fig. 10C). Therefore, the fluorescence density of sham group was lower than that of other groups at the same time ($p < 0.05$) (Fig. 10D). Meanwhile, the fluorescence density of GAP-43 in MSaP-aL/p group was significantly higher than that in other control groups, which proved that immunoregulation of local inflammatory response during acute spinal cord injury could promote nerve regeneration for a long time ($p < 0.05$). In addition, aP group was lower than that in the microsol electrospun groups (MSaP, MSaP-aL, MSaP-aL/p) at the same time ($p < 0.05$), suggesting that continuous supply of NGF could significantly promote nerve repair. Surprisingly, with the development of time, the expression of GAP-43 in all experimental groups except sham group at 4 weeks were significantly higher than those at*

8 weeks ($p < 0.05$). The results suggested that there is a limited time window for the repair of spinal cord injury. Immune regulation to reduce inflammatory injury in the stage of acute spinal cord injury can better promote nerve regeneration (page 22, line 613).

Figure 10. Immunofluorescence staining of neuron cells and axons. (A), Immunofluorescence staining of neuron cells and (B) quantification analyzed with optical density. ($n=5$, Tuj-1 intensity values were mean \pm std. dev., $\%p < 0.05$ when comparing blank control and microsolvated groups (MSaP, MSaP-aL and MSaP-aL/p); $\&p < 0.05$ when comparing aP and microsolvated groups (MSaP, MSaP-aL and MSaP-aL/p); $\@p < 0.05$ when comparing 4w and 8w in MSaP; $\$p < 0.05$ when comparing 4w and 8w in MSaP-aL and $\#p < 0.05$ when comparing 4w and 8w in MSaP-aL/p; $*p < 0.05$ when comparing MSaP-aL/p and other control groups via two-way analysis of variance (ANOVA) with a Tukey's post-hoc test, scale bars are $50\mu\text{m}$) (C), Immunofluorescence staining of axon with (D) its quantification analyzed with optical density. ($n=5$, GAP-43 intensity values were mean \pm std. dev., $\#p < 0.05$ when comparing 4w and 8w in blank control groups, aP, MSaP, MSaP-aL and MSaP-aL/p; $\@p < 0.05$ when comparing sham group and other groups; $\&p < 0.05$ when comparing aP and microsolvated groups (MSaP, MSaP-aL and MSaP-aL/p), $*p < 0.05$ when comparing MSaP-aL/p and other control groups via two-way analysis of variance (ANOVA) with a Tukey's post-hoc test, scale bars were $50\mu\text{m}$)

Figure 6. Neuron-specific marker immunofluorescence staining and gene detection assays of scaffolds. (A) Immunofluorescence staining of Tuj-1, NSE, NF-200, Tau, staining images on different fiber scaffolds with (B), (C), (D), (E) Corresponding fluorescence semi-quantitative analysis. (F), (G), (H), (I) mRNA expression analysis of Tuj-1, NSE, NF-200, Tau. ($n=3$, all values were mean \pm std. dev., * $p < 0.05$ when comparing microsol electrospun groups (MSaP, MSaP-aL, MSaP-aL/p) and control, aP group respectively via one-way analysis of variance (ANOVA) with a Tukey's post-hoc test., ns, not significant, scale bars were 100 μ m)

*Figure S12. Immunofluorescence staining of neural stem cells and neuron cells. (A), Immunofluorescence staining of neural stem cells with (B) its quantification analyzed with optical density. (n=5, Nestin intensity values were mean ± std. dev., #p<0.05 when comparing sham group and other groups and *p<0.05 when comparing MSaP-aL/p and other control groups via two-way analysis of variance (ANOVA) with a Tukey's post-hoc test, scale bars were 50µm)*

3. The authors also applied NGF to stimulate mobilization and differentiation of stem cells into neurons. According to their data, the authors stated the this novel approach increased the stem cell counts and differentiation into neurons. However, this is based on immunohistochemistry against Nestin and Tuj1. These experiments are not appropriated to demonstrate or even to suggest their statements.

Re : Thank you for the comment. The misunderstanding about the mobilization of neural progenitor cells may be caused by the unclear description in the original manuscript. We had removed the content of endogenous stem cell mobilization and made corrections. Nestin fluorescence labeling test was used to evaluate whether immunoregulation during acute spinal cord injury could promote the appearance of neural progenitor cells at injured area in the later stage. The corresponding correction were made in the revised manuscript and provided as follow.

(1). **2.4 In vitro biological characteristics of nerve fiber membranes.** *In this study, BMSCs and bone marrow macrophages (BMMs) were co-cultured through Transwell to simulate the effects of fiber membranes on cell-to-cell interaction in the microenvironment of acute spinal cord injury, and to evaluate the biological characteristics of nerve fiber membranes (page 13, line 337).*

(2). **2.5.4 Histology and immunohistochemical evaluation of spinal cord repair in vivo.** *Nestin, Tuj-1 and GAP-43 were used to label neural progenitor cells, neuron cells and axonal sprouting respectively to evaluate the ability of biomimetic bundles to promote nerve regeneration in the middle and late stage of spinal cord injury. The distribution of nestin-labeled neural progenitor cells at the injured site (Fig. S12A) was observed in each bundle group at 4 and 8 weeks after injury. The neural progenitor cells in the sham group was lower than the other groups, which may be due to the low proliferation activity of spinal ependymal cells and the lack of immune response to nestin protein ($p < 0.05$), and the fluorescence quantitative display (Fig. S12B) at the same time point in the MSaP-aL/p group was significantly higher than that in the other control groups ($p < 0.05$). The results showed that biomimetic bundle could significantly inhibit the blocking effect of surrounding scar tissue on neural progenitor cells through immunomodulatory effect. At 4w and 8w time points, neural progenitor cells were continuously promoted by NGF released from microsol electrospun fibers (MSaP, MSaP-aL and MSaP-aL/p), and a large*

number of Tuj-1-labeled neurons appeared as shown in Fig. 10A (page 21, line 587).

4. The authors reveal that this novel approach improved functional outcomes based on the BBB and IPT. It is quite surprising to observe statistical difference between groups with such small difference. It is also surprising to observe this tiny standard deviation after hemisection injury. Highlight that the statistics used are not appropriate since there are two factors (experimental group and time). Two-way RM ANOVA needs to be used.

Re : Thank you for the valuable comment. The recovery of animal motor function is an important indicator for evaluating nerve repair. We had realized the mistake on using an unacceptably small sample size (n=3) for scoring when the original manuscript was first submitted, leading to inaccuracy of the comparison. For more accurate assessment, we had repeated and increased the sample size (n=7) for re-statistical analysis. Corresponding corrections were provided as follow:

Method: *The recovery of neuromotor function of hindlimb in six groups of rats was evaluated by BBB (Basso, Beattie, Bresnahan) and IPT (inclined plane test) scoring system. The score range of BBB scale was 0 to 21 with 0 indicating no hindlimb movement, and 21 indicating that motor function was normal. During each test, two unwitting examiners evaluated separately each week after the operation. As a supplementary study of BBB scoring test, IPT can improve the effectiveness and sensitivity of scoring. Using Rivlin method, the rats were placed on the inclined plate with rubber pad, the longitudinal axis of the rat was kept parallel to the longitudinal axis of the inclined plate, the head was raised to one side, and raised 5 ° from 0 ° within 5 seconds. The highest angle of stay of the rats was recorded and the larger the angle indicated the stronger the load-bearing capacity of the lower extremities. The above two scoring methods are performed by two blinded observers and seven rats were randomly selected from each group at a fixed time point every week for evaluation (n=7) (Supplementary information, page 16, line 405).*

Results and discussion: *The in vivo performance of fiber bundle (Fig. 7A (a, b, c, d) was evaluated by SD rat T9 spinal cord hemisection model. The recovery of hindlimb motor dysfunction after spinal cord injury was evaluated by BBB and IPT scores every week after operation. The results showed that the hindlimb motor function of rats in each group recovered in varied degrees. Superior recovery of motor function could be identified in MSaP-aL/p group compared with other groups, as indicated by significantly higher BBB score and IPT score at each time points from 4th week post-injury ($p < 0.05$). At the 8th week, the BBB score of 13.71 ± 1.11 , and the IPT score of $55.86 \pm 4.67^\circ$ could be found in MSaP-aL/p group, both of which were significantly higher than those of the other groups ($p < 0.05$) (Fig. 7B, C). In addition, the minimum time required to achieve similar motor functions during recovery is more convincing in characterizing the functionality of neural repair materials. Therefore, in the comparison of 10 points (the highest average score of the negative control group in the BBB score) and 47° (the highest average angle of the*

negative control group in the IPT test), the two scores of the MSaP-aL/p group exceeded the control values at the 4th week after operation, reaching 11.14 ± 2.04 (BBB) and $52.43 \pm 5.56^\circ$ (IPT) respectively. The results of motor function recovery suggested that immunoregulatory fiber bundle (MSaP-aL/p) implantation could reduce the risk of further damage to surviving motor neurons by effectively inhibiting the acute inflammatory response of spinal cord injury, and promote nerve repair subsequently (page 18, line 482).

Figure 7. Timeline of animal assay and motor function score. (A) Pictures of animal experiments. Image of (a) the responsive fiber bundle, (b) the fully exposed T9 spinal cord, (c) 3mm hemi-section made in the right site of the T9 spinal cord (the yellow arrow in the image indicated the injured site), (d) implanted fiber bundles (the yellow arrow in the image indicated the fiber bundle implantation position) along with (e), (f), (g), (h) post-operation condition of rat at 7 days, 4 weeks and 8 weeks. (B) Evaluation of motor function recovery of lower limb by BBB score in rats assisted by (C) rats motor function IPT score. ($n=7$, all values were mean \pm std. dev., $p < 0.05$ when comparing MSaP-aL/p and other groups via two-way analysis of variance (ANOVA) with a Tukey's post-hoc test)

5. English should be revised throughout the text.

Re : Thank you for the comment. The whole manuscript has been polished thoroughly, partial grammar has been corrected. All the changes have been highlighted in the revised manuscript.

6. The n should be included in all the experiments

Re : Thank you for raising the key point. We included the sample size (n) used for statistical evaluation to all data descriptions, and highlighted in the revised manuscript.

Reference

1. Metz, G. A. et al., Validation of the weight-drop contusion model in rats: a comparative study of human spinal cord injury. *J. Neurotrauma*. **17**, 1-17 (2000).
2. Wang, X. J. et al., Polysialic-Acid-Based Micelles Promote Neural Regeneration in Spinal Cord Injury Therapy. *Nano Lett.* **19**, 829-838 (2019).
3. D'Amato, A. R. et al., Vastly extended drug release from poly(pro-17beta-estradiol) materials facilitates in vitro neurotrophism and neuroprotection. *Nat. Commun.* **10**, 4830 (2019).
4. Li, G. et al., Graft of the NT-3 persistent delivery gelatin sponge scaffold promotes axon regeneration, attenuates inflammation, and induces cell migration in rat and canine with spinal cord injury. *Biomaterials*. **83**, 233-248 (2016).
5. Milich, L. M. et al., The origin, fate, and contribution of macrophages to spinal cord injury pathology. *Acta Neuropathol.* **137**, 785-797 (2019).
6. Cerqueira, S. R. et al., Decellularized peripheral nerve supports Schwann cell transplants and axon growth following spinal cord injury. *Biomaterials*. **177**, 176-185 (2018).
7. Rejman, J. et al., Size-dependent internalization of particles via the pathways of clathrin- and caveolae-mediated endocytosis. *Biochem J.* **377**, 159-169 (2004).
8. Zhou, L. et al., Microsol-electrospinning for controlled loading and release of water-soluble drugs in microfibrinous membranes. *RSC Adv.* **4**, 43220-43226 (2014).
9. Cossich, E. et al., Development of electrospun photocatalytic TiO₂-polyamide-12 nanocomposites. *Biotechnol. Adv.* **31**, 421-437 (2015).
10. Zhang, Y. Z. et al., Coaxial Electrospinning of (Fluorescein Isothiocyanate-Conjugated Bovine Serum Albumin)-Encapsulated Poly(ϵ -caprolactone) Nanofibers for Sustained Release. *Biomacromolecules*. **7**, 1049-1057 (2006).
11. Shi, Q. et al., Collagen scaffolds modified with collagen-binding bFGF promotes the neural regeneration in a rat hemisectioned spinal cord injury model. *Sci China Life Sci.* **57**, 232-240 (2014).
12. Veisoh, O. et al., Domesticating the foreign body response: Recent advances and applications. *Adv Drug Deliv Rev.* **144**, 148-161 (2019).
13. Li, F. et al., CCR5 blockade promotes M2 macrophage activation and improves locomotor recovery after spinal cord injury in mice. *Inflammation*. **38**, 126-133 (2015).
14. Chen, C. et al., Bioinspired Hydrogel Electrospun Fibers for Spinal Cord Regeneration. *Adv. Funct. Mater.* **29**, 1806899 (2019).

15. Wu, L. et al., Hierarchical micro/nanofibrous membranes of sustained releasing VEGF for periosteal regeneration. *Biomaterials*. **227**, 119555 (2020).
16. Francos-Quijorna, I. et al., IL-4 drives microglia and macrophages toward a phenotype conducive for tissue repair and functional recovery after spinal cord injury. *Glia*. **64**, 2079-2092 (2016).
17. Li, H. et al., Isolation and characterization of primary bone marrow mesenchymal stromal cells. *Ann N Y Acad Sci*. **1370**, 109-118 (2016).
18. Pawar, K. et al., Biomaterial bridges enable regeneration and re-entry of corticospinal tract axons into the caudal spinal cord after SCI: Association with recovery of forelimb function. *Biomaterials*. **65**, 1-12 (2015).
19. Xu, D. et al., Efficient Delivery of Nerve Growth Factors to the Central Nervous System for Neural Regeneration. *Adv. Mater.* **31**, e1900727 (2019).
20. Bellver-Landete, V. et al., Microglia are an essential component of the neuroprotective scar that forms after spinal cord injury. *Nat. Commun.* **10**, 518 (2019).
21. He, S. et al., Multiple release of polyplexes of plasmids VEGF and bFGF from electrospun fibrous scaffolds towards regeneration of mature blood vessels. *Acta Biomater.* **8**, 2659-2669 (2012).
22. Monteiro, N. et al., Antibacterial activity of chitosan nanofiber meshes with liposomes immobilized releasing gentamicin. *Acta Biomater.* **18**, 196-205 (2015).
23. Monteiro, N. et al., Instructive nanofibrous scaffold comprising runt-related transcription factor 2 gene delivery for bone tissue engineering. *ACS Nano*. **8**, 8082-8094 (2014).
24. Storer, P. D. et al., β II-tubulin and GAP 43 mRNA expression in chronically injured neurons of the red nucleus after a second spinal cord injury. *Exp Neurol*. **183**, 537-547 (2003).
25. Berghoff, S. A. et al., Dietary cholesterol promotes repair of demyelinated lesions in the adult brain. *Nat. Commun.* **8**, 14241 (2017).
26. Li, T. et al., The neuritic plaque facilitates pathological conversion of tau in an Alzheimer's disease mouse model. *Nat. Commun.* **7**, 12082 (2016).
27. Cui, X. et al., D-4F Decreases White Matter Damage After Stroke in Mice. *Stroke*. **47**, 214-220 (2016).
28. Garg, K. et al., Macrophage functional polarization (M1/M2) in response to varying fiber and pore dimensions of electrospun scaffolds. *Biomaterials*. **34**, 4439-4451 (2013).
29. Lee, M. K. et al., The expression and posttranslational modification of a neuron-specific beta-tubulin isotype during chick embryogenesis. *Cell Motil Cytoskeleton*. **17**, 118-132 (1990).
30. Portiansky, E. L. et al., Increased number of neurons in the cervical spinal cord of aged female rats. *PLoS ONE*. **6**, e22537 (2011).
31. Portier, M. M. et al., Peripherin and neurofilaments: expression and role during neural

development. *C R Acad Sci III*. **316**, 1124-1140 (1993).

32. Caprelli, M. T. et al., CNS Injury: Posttranslational Modification of the Tau Protein as a Biomarker. *Neuroscientist*. **25**, 8-21 (2019).

Reviewers' Comments:

Reviewer #1:

Remarks to the Author:

It is valued the effort of the authors to provide answers to the questions previously raised. However, most of the answers to the points raised previously do not satisfactorily address the problems identified. Thus, this manuscript should not be considered further for publication.

Reviewer #2:

Remarks to the Author:

Most of my questions were properly answered.

However, one last simple issue is still open: if the sample size for the behavioral tests was increased to $n=7$ per group why the total amount of animals used is the same? Did the authors performed new implantation surgeries or not? If not, did they select previous undisclosed data? If so, how did they do that? Was it a random choice? please explain.

Reviewer #3:

Remarks to the Author:

The authors have addressed my previous concerns.

Minor point:

Material and methods: line 370. "were first gated for CD11b to ensure that only microglia/macrophages were selected". Please, change microglia/macrophages for myeloid cells, since granulocytes are also CD11b+

Reviewers' comments :

The black text was copied from the editors' letter, and the blue indentation is our response.

Reviewer #1 (Remarks to the Author):

It is valued the effort of the authors to provide answers to the questions previously raised. However, most of the answers to the points raised previously do not satisfactorily address the problems identified. Thus, this manuscript should not be considered further for publication.

Re: Thank you for your time spent in the manuscript reviewing and acknowledging our efforts. As Alexandre Dumas said, "If the world has not become perfect all at once, at least it has become better than before." Please allow us to address current issues and further improve our manuscript. This paper is very important for us because of the first high level research in our group, and we spend long time for this research. Please tell us your opinions for this research.

Reviewer #2 (Remarks to the Author):

Most of my questions were properly answered.

However, one last simple issue is still open: if the sample size for the behavioral tests was increased to $n=7$ per group why the total amount of animals used is the same? Did the authors performed new implantation surgeries or not? If not, did they select previous undisclosed data? If so, how did they do that? Was it a random choice? please explain.

Re: Great thanks to the reviewer for the time and effort spent on the comprehensive and critical read of our manuscript. When the original manuscript was first submitted, we had realized the mistake on using an unacceptably small sample size ($n=3$) for scoring, which led to the inaccuracy of comparison. For more accurate assessment, we had performed new implantation surgeries. A total of 60 rats which underwent surgeries were supplemented in the motor function re-statistical analysis and randomly divided into six groups, from each of which, 7 rats were randomly selected from each group at a fixed time point ($n=7$). Hence, we've indeed made a mistake on putting a wrong total amount of animals, and we have to apologize for such mistake. Corrections have been made and corresponding content was highlighted in the revised manuscript and provided as follow.

A total of 330 female Sprague-Dawley (SD) rats (around 200g) were purchased from JOINN Laboratories (Soochow, China), and were randomly divided into six groups. Animal management and surgery were carried out according to the plan approved by the Ethics Committee of the First Affiliated Hospital of Soochow University (Supplementary information, page 13, line 325).

Reviewer #3 (Remarks to the Author):

The authors have addressed my previous concerns.

Minor point:

Material and methods: line 370. “were first gated for CD11b to ensure that only microglia/macrophages were selected”. Please, change microglia/macrophages for myeloid cells, since granulocytes are also CD11b+

Re: Thank you for your meticulous reading and suggestion. Corresponding corrections has been made according to your advice and highlighted in the revised manuscript and provided as follow.

For analysis, cells were first gated for CD11b to ensure that only myeloid cells were selected, then the following combination of specific markers were used to identified M1 (CD86) and M2 (CD206) (Supplementary information, page 14, line 361).

RESPONSE TO REVIEWERS COMMENTS

Reviewers' comments:

The black text was copied from the editors' letter, and the blue indentation is our response.

Reviewer #1 (Remarks to the Author):

The manuscript describes the preparation and characterization of aldehyde-modified cationic liposomes loaded with pDNA encoding for interleukin-4 (IL-4) grafted through Schiff base bond at the surface of amino-modified oriented electrospun fibrous scaffolds. These fibrous scaffolds were produced by a so-called microsolv electrospinning, a process previously proposed by the authors, where the core is loaded with nerve growth factor (NGF).

Thank you for your patient and thoughtful reading as well as the constructive comments and advices about our manuscript. We have completely revised the original manuscript based on your comments and suggestions. Detailed point-by-point responses are provided below.

The pH-responsive immunoregulation-assisted neural regeneration strategy seems to be able to release the pDNA under acidic environment, inducing some cell polarization and anti-inflammatory factors (IL-10 and TGF- β) secretion from macrophages and microglia. Indeed, the amino PLA microsolv electrospinning carrying aldehyde cationic liposomes loading IL-4 plasmid scaffold seems to reduce scar tissue formation, resulting in marginally enhanced motor function recovery when implanted at the spinal cord hemisection site. However, the histology results refer only to one animal analysed, are the histological results relative to the best animal? To have a clear perspective of the results, it is important to show results of the best, worst and more frequent result obtained. Without clarifying those results, it is impossible to derive useful conclusions.

Re : Thank you for the comment and suggestion.

(1). We consider three aspects when choosing animal: (I). Similar biological characteristics to humans; (II). Animal ethical requirements and prices. According to previous research results, Animal ethical requirements and prices. According to previous research results, rats and humans share high similarity on biological characteristics, immune response, behavioral manifestations, and electrophysiological parameters, as well as the process of axon regeneration and structural remodeling in the central nervous system ¹. In addition, SD rats are relatively inexpensive and we hold a license for experimental animal ethics research. (III). Meanwhile, we reviewed the literature of central nervous system regeneration in nerve tissue engineering which showed that the actual operation process of

rats is effective and the experimental results are feasible and reproducible ². After modeling *in vivo*, the loss of ipsilateral lower extremity motor function can be observed after surgery which confirmed the feasibility of SD rats for constructing spinal cord injury model (Figure 7A(e)).

(2). We selected representative images in the manuscript to show the results of each animal assay, and obtained the statistical analysis results by random sampling with several samples. Moreover, in order to show the results of the experiments more clearly, we have added the sample size (n) for each experiment to all the results descriptions and improved the description of the highlighted results in the manuscript. In addition, scatter points were added to all column charts to represent the distribution of all data results.

The developed fibrous scaffold with dual release capacity represents an attractive concept for the application envisaged. The novelty of the present approach is moderate and appears to be in the combination of the local immune regulation and sustained stimulation of neural tissue regeneration. The experimental design has important shortcomings to conclude about the results achieved, particularly the statistics and the cohort of animals considered in the reported results.

Re : Thank you for the comment. Following revision was conducted according to your advice.

(1). We have reconducted the statistical analysis of the data in the manuscript through GraphPad Prism 7.0 software. The specific measures are as follows: one-way ANOVA followed by Tukey's multiple comparison test for single variable data analysis. Two-way ANOVA followed by Tukey's multiple comparison test when analyzing two variable data. In addition, the sample size (n) is added to each result description as mentioned in the above question. We have corrected the statistical analysis part of the article as follows:

The data was presented in the form of means \pm standard deviations. The statistical analysis (Origin 9.1 or GraphPad Prism 7.0 software) was calculated by one-way or two-way analysis of variance by Tukey's multiple comparison test to evaluate the differences among the groups unless otherwise stated. The probability value (p) less than 0.05 was considered to be statistically significant (Supplementary information, page 17, line 446).

(2). The original manuscript in the animal experiment section on the cohort grouping was not appropriate, the correction was restated as follows:

a. *A total of 270 female Sprague-Dawley (SD) rats (around 200g) were purchased from JOINN Laboratories (Soochow, China). The animals were randomly divided into six groups with 45 rats in each group (Supplementary information, page 13, line 325).*

b. **6.3 Evaluation of the degree of local immune inflammatory response to spinal cord injury.** *In order to evaluate the local immunomodulatory effect of nerve bundles in response to spinal cord injury microenvironment, 15 rats were randomly selected from each group at 1 week after operation (Fig7. A(f)) (Supplementary information, page 14,*

line 346).

6.3.1 Identification of microglia/macrophages phenotype. *In the assay, three samples were randomly selected from each group for detection (n=3) (Supplementary information, page 14, line 364).*

6.3.2 Fluorescently labeled immune factors. *Photographing by fluorescence microscope, and semi-quantitative analysis of five randomly selected images which had been corrected for optical density, de-background, and finally normalized the staining intensity by the number of cell nuclei using ImageJ software (Supplementary information, page 15, line 377).*

6.3.3 Assessment of inflammatory genes in injured local tissues. *To evaluate the transfection performance and immunoregulatory function after fiber bundles implantation, three randomly selected spinal cord specimens were dehydrated in ethanol followed by xylene (n=3) (Supplementary information, page 15, line 382).*

6.3.4 Quantitative serum immune factors. *Briefly, three animals were randomly selected from each group, and the blood was collected with a centrifuge tube containing coagulant during cardiac perfusion (n=3) (Supplementary information, page 16, line 398).*

c. Evaluation of animal motor function. *Seven rats were randomly selected from each group at a fixed time point every week for BBB and IPT evaluation (n=7) (Supplementary information, page 16, line 417).*

d. Histological analysis of spinal cord. *Thirty rats in each group were randomly divided into 4w group and 8w group, and were euthanized at pre-designed time (Supplementary information, page 17, line 421).*

H&E staining evaluation: *Different fiber bundles were cut to the size of 10mm×10mm×0.1mm and implanted under the epidermis on the back of SD rats to evaluate the foreign body reaction in vivo. The specimens were harvested 2 weeks after operation. After fixed in 10% formalin solution, the specimens were sliced and stained with H&E to observe the inflammatory response zone, and randomly selected three fields of view for semi-quantitative and statistical analysis of inflammation area using ImageJ software (n=3) (Supplementary information, page 13, line 318). The spinal cord cavity area of each group was measured by ImageJ software, and three fields were randomly selected for statistical analysis (n=3) (Supplementary information, page 17, line 425).*

Immunohistochemical evaluation: *The primary antibody was added and incubated at 4 °C overnight, and the second antibody was incubated at room temperature for 1 hour after PBS washing. The samples were observed and photographed under fluorescence microscope, with the fluorescence intensity analyzed with ImageJ software. The values were represented in the form of average optical intensity, which had been corrected for optical density, de-background, and finally normalized by the number of cell nuclei. A total of five images were randomly selected from each group for statistical analysis (n=5) (Supplementary information, page 17, line 428).*

Questions regarding the concept of the implant:

- There is no evidence provided in the manuscript regarding the release of plasmid and of NGF over the time-span of the experiment (4-8 weeks). Thus, it is not clear if the implant is really operating in accordance with the projected performance.

Re : Thank you for the comment. This study aimed to construct a responsive biomimetic immunoregulatory electrospinning fiber scaffold that can respond and then immune-regulate the local microenvironment during the acute spinal cord injury and provide a differentiation platform for subsequent nerve regeneration. Therefore, demonstration of early rapid released IL-4 plasmid and sustained released NGF *in vivo* was of great importance in our study. Thus, we've further evaluated and discussed whether biomimetic composites can achieve expected performance in terms of biological effects in *in vivo* research. Additionally, we had referred to associated references to support our discussion³.

(1). We've established the model for the simulation of the *in vivo* pH-responsive release process of liposome according to the procedure previously reported⁴. Figure 4E showed that the biomimetic composite scaffold responded to the acidic environment and released cationic liposomes loading with IL-4 plasmid (approximately 90%) within 24 hours, which was consistent with the time window of blood-derived macrophages appearing at the spinal cord injury site⁵. In addition, the release kinetics of NGF under simulated pH environment was also measured in long time-span. As we could see in release kinetic of NGF measured using ELISA kits in Figure 4F, the acidic pH did have some effect on the release of NGF, reflected by the burst release of $40.06 \pm 2.29\%$ ($320.48 \pm 18.33\text{ng}$) under pH5.8, $36.59 \pm 3.19\%$ ($292.69 \pm 25.52\text{ng}$) under pH6.6 and $20.86 \pm 5.92\%$ ($166.88 \pm 47.35\text{ng}$) under pH7.4 in the first two days. However, owing to the effect of microsol spinning, which provided a relatively stable environment for the core part loaded with NGF, a generally controlled release of the factor could be identified, indicated by the slowly and stably release in the last 6 weeks under all three pH, verifying the effectiveness of our system. Specifically, NGF was allowed for a stable and sustained release for 40 days in an acidic environment of pH 5.8 ($87.99 \pm 1.90\%$, $703.89 \pm 15.24\text{ng}$). Our group have several years' experience on microsol spinning technique, which in current study, showed acceptable performance on protecting the factor loaded by core against harsh microenvironment (Figure 4F).

Specifically, the releasing rates of liposome was tested through immersing the liposome-grafted fiber scaffold in PBS solutions with different pH (Fig. 4E). In the acidic environments with pH of 5.8 and 6.6, the release of liposomes increased gradually with time. The release rate of liposomes under the pH of 5.8 reached $61.66 \pm 1.38\%$ within 10 hours, which was significantly higher than that of other pHs ($p < 0.05$). Therefore, quick respond to acidic environment of the responsive fiber scaffold could be verified in vitro which provided immunoregulatory signals for gathered immune cells during acute spinal cord injury in time (page 12, line 315).

Hence, the release characteristics of NGF from microsol electrospun fibers under different pH were also evaluated (Fig. 4F).The results showed that the release rate under the pH of

5.8 was slightly increased in acidic environment, reaching $71.64 \pm 1.82\%$ ($544.21 \pm 21.41\text{ng}$) on the 10th day, while $68.14 \pm 4.59\%$ ($545.09 \pm 36.72\text{ng}$) and $59.54 \pm 1.93\%$ ($476.29 \pm 15.41\text{ng}$) under pH of 6.6 and 7.4. All scaffolds under different pH exhibited sustained releases of NGF which lasted for more than 40 days (page 12, line 326).

Figure 4E, F. (E) Cumulative release curve of aliposomes in different pH environments. ($n=3$, aliposome release values were mean \pm std. dev., $p < 0.05$ when comparing pH5.8 and the other two pHs via two-way analysis of variance (ANOVA) with a Tukey's post-hoc test) (F) Cumulative release curve of NGF in microsol electrospinning in different pH environments. ($n=3$, NGF release values were mean \pm std. dev., statistical analysis evaluated by two-way analysis of variance (ANOVA) with a Tukey's post-hoc test)

(2). Apart from the release kinetics, *in vivo* biological effects of immune regulation in acute spinal cord injury was also characterized in order to verify the immunological effect of liposome. One week after surgery, immunofluorescence staining was employed to evaluate the localized levels of IL-4, TNF- α (M1), and IL-10 (M2) in rat (Figure 9A), which indicated significantly higher IL-4 and IL-10 fluorescence intensity in the MSaP-aL/p group ($p < 0.05$) compared with control groups (blank control, aP, MSaP, MSaP-aL). However, significantly lower TNF- α intensity was captured in MSaP-aL/p group ($p < 0.05$) (Figure 9B, C). In order to more accurately evaluate the transfection performance and the levels of pro-inflammatory and anti-inflammatory response, we collected spinal cord tissue for IL-4, TNF- α , IL-1 β , IL-10 and TGF- β gene expression detection using qRT-PCR assay one week after surgery (Figure 9D-H). The qRT-PCR test results were consistent with the fluorescence staining results, which also showed that biomimetic electrospun fibers have exerted immunoregulatory effects at the injured site as expected during acute spinal cord injury. The specific experimental method and results are shown below and highlighted in the revised manuscript.

Method: To evaluate the transfection performance and immunoregulatory function after fiber bundles implantation, three randomly selected spinal cord specimens were dehydrated in ethanol followed by xylene ($n=3$). Then total RNA of spinal cord was isolated by using the RNAprep Pure FFPE kit (TIANGEN BIOTECH, China) according to the manufacturer's instructions. cDNA was synthesized with 1 μ g of RNA using a Hiscript II 1st Strand cDNA Synthesis kit (Vazyme, Nanjing, China). Quantitative RT-PCR was carried out in a 20 μ L total volume ChamQ Universal SYBR qPCR Master Mix (Vazyme, Nanjing, China). The gene expression levels were normalized to GAPDH. Primers were

shown in Table 5 (Supplementary information, page 15, line 382).

Table 5. The primer sequence of immune factor for qRT-PCR in vivo.

Gene	Primer	Sequence	Tm (°C)
IL-4	Forward	GCAACAAGGAACACCACGG	60
	Reverse	AAGCACGGAGGTACATCACGT	57.8
TNF- α	Forward	TACTGAACTTCGGGGTGATTGGTCC	62.22
	Reverse	CAGCCTTGTCCTTGAAGAGAACC	62.11
IL-1 β	Forward	GAAAGACGGCACACCCACC	62
	Reverse	AAACCGCTTTCCATCTTCTTCT	54.86
IL-10	Forward	CCCTCTGGATACAGCTGCG	62
	Reverse	GCTCCACTGCCTTGCTTTTATT	56.26
TGF- β	Forward	CTTCAGCTCCACAGAGAAGAACTGC	62.22
	Reverse	CACGATCATGTTGGACAACCTGCTCC	62.22
GAPDH	Forward	AACTCCCATTCTTCCACCT	56
	Reverse	TTGCATACCAGGAAATGAGC	53.90

Results and discussion. Meanwhile, according to the results of the expression of IL-4 gene in MSaP-aL/p group was significantly higher than that of other groups ($p < 0.05$) (Fig. 9D). Combined with the above fluorescence labeled IL-4 results, it was further proved that it had played the function of transfection in response to the acidic environment of spinal cord injury. The expressions of pro-inflammatory genes TNF- α and IL-1 β in MSaP-aL/p, shown as Fig. 9E, F, were significantly decreased ($p < 0.05$). On the other hand, Fig. 9G, H showed the expressions of anti-inflammatory genes IL-10 and TGF- β were significantly higher than those in other control groups, respectively. The results showed that the immune bundles had played an immunomodulatory role during the acute spinal cord injury (page 19, line 533).

Figure 9D-H. Spinal cord specimens of IL-4 (D), TNF- α (E), IL-1 β (F), IL-10 (G) and TGF- β (H) gene expressions were assessed by qRT-PCR. ($n=3$, qRT-PCR values were mean \pm std. dev., * $p < 0.05$ when comparing MSaP-aL/p and other groups via one-way analysis of variance (ANOVA) with a Tukey's post-hoc test)

(3). The long-term biological effects of immune regulation on spinal cord injury was also characterized using immunofluorescence to label activated astrocytes and scar tissues at 4w and 8w after operation (Figure S11). We found significantly lower intensity from

labeled activated astrocytes and scar tissues in the MSaP-aL/p group compared with non-immunized control group (blank control, aP, MSaP, MSaP-aL) ($p < 0.05$). This result confirmed that immune intervention during acute spinal cord injury can significantly reduce astrocyte activation and scar tissues formation during chronic stage of spinal cord injury, which could pave the way for neural regeneration.

Figure S11. Immunofluorescence staining of activated astrocytes and glial scar. (A), (C) Immunofluorescence staining of astrocytes and quantitatively analyzed with optical density along with (B), (D) their quantification with optical density. ($n=5$, all values were mean \pm std. dev., $* p < 0.05$ when comparing MSaP-aL/p and other control groups via two-way analysis of variance (ANOVA) with a Tukey's post-hoc test, $\# p < 0.05$ when comparing material transplantation groups and blank control group via two-way analysis of variance (ANOVA) with a Tukey's post-hoc test, scale bars were $50 \mu\text{m}$)

(4). Neural regeneration was also fully characterized. As you stated in the question, it is important to explore whether biomimetic composite fibers can control the release of NGF and play a role in promoting neural differentiation *in vivo* as expected. Therefore, the assessment of biological effects on neural regeneration was conducted according to the previously reported study with the time point set for 4 and 8 weeks after surgery⁶. As shown in Figure 10A, B, the sustained release of NGF could be verified through significantly promoted neural differentiation of endogenous progenitor cells. In order to ensure the accuracy of the description of the experimental results, we have partially modified the original description as shown below and highlighted it in the revised manuscript.

At 4w and 8w time points, neural progenitor cells were continuously promoted by NGF released from microsolv electrospun fibers (MSaP, MSaP-aL and MSaP-aL/p), and a large number of Tuj-1-labeled neurons appeared as shown in Fig. 10A. It was significantly better than that in blank control and aP groups (Fig. 10B) ($p < 0.05$). Meanwhile, the

fluorescence intensity of each microsol electrospinning group at 8w was significantly higher than that at 4w respectively ($p < 0.05$). The result further proved that the biomimetic fiber bundle has the function of continuously promoting nerve regeneration. In the MSaP-aL/p group, due to the effect of responsive immunomodulation, the toxicity of astrocytes and the formation of scar tissues were significantly reduced, so at two time points, the amounts of neurons were significantly larger than that in other control groups ($p < 0.05$). The above results confirmed that the biomimetic bundle not only had the ability of immunomodulation in the area of spinal cord injury, but also consistent with the reported in the literature that the core-shell structure of microsol electrospun fibers has the ability to protect NGF from the adverse biological environment which could continuously provide differentiation information for nerve regeneration (page 21, line 598).

Figure 10A, B. (A), Immunofluorescence staining of neuron cells and (B) quantification analyzed with optical density. ($n=5$, Tuj-1 intensity values were mean \pm std. dev., $\%p < 0.05$ when comparing blank control and microsol electrospun groups (MSaP, MSaP-aL and MSaP-aL/p); $\&p < 0.05$ when comparing aP and microsol electrospun groups (MSaP, MSaP-aL and MSaP-aL/p); $\@p < 0.05$ when comparing 4w and 8w in MSaP; $\$p < 0.05$ when comparing 4w and 8w in MSaP-aL; and $\#p < 0.05$ when comparing 4w and 8w in MSaP-aL/p; $*p < 0.05$ when comparing MSaP-aL/p and other control groups via two-way analysis of variance (ANOVA) with a Tukey's post-hoc test, scale bars were $50 \mu\text{m}$)

- To be effective, the plasmid needs to be internalised by the target cells. No evidence is provided in the histology of the explants that the cells were effectively transfected by the plasmid. Indeed, the plasmid should have a reporter gene that should facilitate the evaluation of the transfection at the defect site.

Re : Thank you for the thoughtful suggestion. The material characterization experiments have initially demonstrated that the Schiff base bond responds to the local acidic microenvironment, releasing nearly 90% of the cationic liposomes loading with the IL-4 plasmid within 24h (Figure 4E). One week after surgery, we would like to prove the transfection effect of the IL-4 plasmid by paraffin section, but we could not observe the tissue fluorescence of eGFP expression due to the samples soaked with formalin or

hydrogen peroxide interference. Hence, in order to fully demonstrate the transfection and immunoregulatory effect of the cationic liposome loading with IL-4 plasmid *in vivo*, we added qRT-PCR technique to access IL-4, TNF- α , IL-1 β , IL-10, TGF- β gene expression in local tissue.

We have supplied the IL-4 plasmid profile to Fig. S1A, and the qRT-PCR experimental method and results have described as above (Figure 9D-H) and highlighted in the revised manuscript.

Figure S1A. IL-4-eGFP plasmid profile. GFP was designed for the report protein of IL-4 expression.

- The NGF is present in the nano fibres during the whole procedure for the grafting of the liposomes at the surface. How much of the NGF is preserved and to what extent it is bioactive after being exposed to this procedure?

Re.: Thank you for the significantly comment.

(1). Drug loading efficiency (LE) is an important parameter in electrospinning as a cytokine sustained release system. To clarify, we have added the experimental content of microsol electrospinning nanofiber related drug loading efficiency as shown below and highlighted in the original article.

Method: To characterize the cytokine content in the fibers, three prepared MSaP-aL/p scaffolds were placed in three 50ml centrifuge tubes containing 2g DCM, centrifuged at 15000rpm for 5 minutes, and the aPLA-containing supernatant was removed. This was repeated three times to ensure that all NGF in the organic phase precipitation. After adding 2ml PBS for resuspension, the ELISA kit was used to detect the NGF content, and the drug loading efficiency of electrospun fibers was calculated as follows: (Supplementary information, page 7, line 171)

$$LE (\%) = \frac{M (\text{total drug in fibers})}{\text{Total drug}} \times 100\%$$

M (total drug before spinning)

Results and discussion: *As measured, the drug loading efficiency (LE) of all NGF-loaded microsolv electrospun scaffolds were about $79.53 \pm 1.44\%$ in this study (page 12, line 322).*

(2). In order to ensure the biological activities of the IL-4 plasmid and NGF in the biomimetic immunoregulatory fibrous scaffold, we carried out relevant characterization immediately after preparation or stored at -20°C for subsequent experiments. In the study of biomimetic fiber membrane release, we also kept it under sterile conditions at -20°C after each sampling to maximally maintain the biological factor activity. We've used the previously collected release solutions which have been stored at -20°C for the 30th to 40th days to culture BMSCs *in vitro* to verify the long-term bioactivity released NGF. The specific methods and results are shown below and highlighted in the revised manuscript.

Method: *Rat bone marrow mesenchymal stem cells (BMSCs) were also seeded into 24-well plates at a density of 1×10^4 /well ($n=3$). When the cells were about 70% -80% fused, added MSaP-aLp release solution (day 30 to 40, pH5.8). Five days later, the neuronal marker Tuj-1 was used to label differentiated neuron-like cells and semi-quantitative fluorescence analysis was performed by imageJ software to evaluate the long-term biological activity of NGF released from the fibrous membrane (r-NGF group). Normally cultured bone marrow mesenchymal stem cells were used as the negative control group (NC), and rat β -NGF cytokines (20ng/ul) were added as the positive control group (PC). Three separate dissociations were carried out to achieve biological triplicate ($n=3$) (Supplementary information, page 12, line 296).*

Results and discussion : *During the neural regeneration process, it is necessary to provide long-term biological signal supply chains for neural progenitor cells. Therefore, it is very important to prove that the biological activity of NGF that is sustained released by biomimetic electrospun fibers. The neuron-specific marker Tuj-1 was still able to label neuron-like cells differentiated from bone marrow mesenchymal stem cells induced by release solution (Fig. S8A). At the same time, without the contact guidance of oriented electrospinning fibers, the cell morphology changed from spindle to long, but no neurite-like formation was observed. As shown in Fig. S8B, the PC (positive control group) is higher in the semi-quantitative fluorescent results than the r-NGF group, but there was no statistical difference between the two groups ($p>0.05$). However, the negative control group was significantly different from the r-NGF group and the PC group, respectively. This result combined with *in vitro* simulated release experiments proved that the biomimetic fibers can sustained release NGF, and that it can be protected by the core-shell structure to maintain its biological activity (page 16, line 446).*

Figure S8. Assessing long-term biological activity of NGF in biomimetic fibers. (A) Immunofluorescence staining of Tuj-1 staining images on cell culture plate (NC), release solution group (r-NGF) and cytokine NGF group (PC) with (B), Corresponding fluorescence semi-quantitative analysis. (n=3, all values were mean \pm std. dev., * $p < 0.05$ when comparing r-NGF, PC and NC group via one-way analysis of variance (ANOVA) with a Tukey's post-hoc test., ns, not significant, scale bars were 200 μ m)

- The size of the liposomes is quite large to enable an easy internalization. The authors should discuss this important aspect in the manuscript.

Re.: Thank you for the comment. As the reviewer said, the size of liposomes is indeed crucial for cell transfection, so we supplemented the discussion concerning the size effect of liposomes in the revised manuscript.

In addition, according to reports, liposomes with a diameter of 200nm-1 μ m can enter the cell slowly through endocytosis, so that DNA can be released from the liposome before reaching the lysosome, which can prevent its degradation in the lysosome ⁷. After screening the pDNA/aLiposome complex at various ratios in this study, it was found that the transfection efficiency was the highest at 1: 2.5, and the average size was 224.1 \pm 4.4nm, so it could be used in subsequent studies (page 8, line 191).

- No details are provided regarding the method used to obtain aligned nano fibrous constructs and its details (dimensions, porosity, degree of alignment, ...). Also the dose of plasmid and NGF effectively implanted and released should be provided.

Re : Thank you for the comment.

(1). We have perfected the description of each group of nano-electrospinning solution and the oriented electrospinning process. The details are as follows and highlighted in the original manuscript:

a. Conventional electrospinning solution : *Aminated polylactic acid electrospinning solution was prepared by dissolving 0.5g amino polylactic acid (aPLA, Mw=100kDa, Mw/Mn=2.06, Ruixi, Xi'an, China) in 4g dichloromethane (DCM, Sinopharm, Beijing, China) and 2g N, N-dimethylformamide (DMF, Lingfeng, Shanghai, China) to obtain uniform and stable solution under magnetic stirring at room temperature. The preparation method of polylactic acid (PLA, Mw=100kDa, Mw/Mn=1.61, Daigang, Jinan, China) electrospinning solution is the same as above (Supplementary information, page 4, line 91).*

b. Microsol electrospinning solution: *The preparation of microsol electrospinning solution as described in the literature⁸. Briefly, started with the preparation of sodium hyaluronate hydrosol (1wt%) (HA, Mw=0.5MDa, Yuancheng, Wuhan, China), which was obtained by dissolving 0.1g HA in 9.9g deionized water and rotating to complete dissolution at room temperature. Rat β -NGF (R&D system, USA) was resuspended in 0.1wt% bovine serum albumin (BSA, Solarbio, Beijing, China) solution to achieve the final concentration of 100 μ g/ml. The uniform HA- β -NGF hydrosol was obtained by mixing 10 μ l resuspended β -NGF with 50 μ l hyaluronic acid solution, after which 0.01g Span-80 (Sigma, USA) and 4g DCM were added into the mixture and stirred at high speed for 30 minutes at room temperature to obtain a homogeneous and stable water-in-oil (W/O) emulsion containing β -NGF microsol particles. The microsol (MS) spinning solution was obtained eventually by adding 0.5g aPLA and 2g DMF into the emulsion. Preparation of different oriented electrospinning fibers (Supplementary information, page 4, line 97).*

c. Oriented electrospinning fibers: *As for the preparation of different oriented fiber scaffolds, the aforementioned electrospinning solutions were loaded in a 10ml syringe with a length of 10cm and an inner needle diameter of 0.9mm. The electrospinning process was conducted with a propulsion pump speed of 70 μ l/min, at a voltage of 15 to 18kV, and a distance between needle tip and the parallel electrode receiver of 15cm. The parallel oriented fiber scaffolds can be collected between the electrode rods. In order to remove the residual solvent from the obtained fibers, all prepared fiber scaffolds were dried under vacuum overnight. Traditional electrospun directional fiber (PLA, aP) and microsol electrospun oriented fibers (MSaP) were also obtained. In the following studies, the oriented fiber membrane was used in vitro study, while the fiber bundle was used in vivo research (Supplementary information, page 5, line 110).*

(2). We have performed scanning electron microscopy (SEM), transmission electron microscopy (TEM), X-ray photoelectron spectroscopy (XPS), atomic mechanical microscopy (AFM) and diameter analysis on traditional oriented electrostatic spinning (aP), oriented microsol electrospinning (MSaP) and oriented responsive nerve fibers (MSaP-aL/p) for material morphological characterization, and representative images were shown in Figure 3 to show the differences among groups. In addition, we supplemented the

characterization of the fibers orientation and porosity with corresponding experimental methods and results were added in the article.

Figure 3. The morphology of different fiber scaffolds. (A) Schematic diagram of amino poly(lactic acid) microsolv electrospun fiber and aldehyde cationic liposomes connected with Schiff base. (B), (C), (D) SEM images of different fiber scaffolds. (E), (F), (G) Histogram of frequency distribution of different fiber scaffolds diameters. ($n=100$ fibers from each sample) (H), (I), (J) TEM images of different fiber scaffolds. (K), (L), (M) XPS indicating chemical elements on different fiber scaffold surfaces. (N), (O), (P) AFM images of scaffolds. The yellow arrows in the images indicate the location of liposome. ($n=3$, all SEM scale bars were $5\ \mu\text{m}$, all TEM scale bars were $0.2\ \mu\text{m}$)

Orientation method: To characterize the orientation of the fiber scaffold, we randomly selected three SEM images from each group, and the overall direction of the fiber membrane was set to 0° . Orientation analysis through the orientation plugin of ImageJ software (Supplementary information, page 5, line 137).

Results and discussion: Nano-topology of neural tissue engineering biomaterials is one of the important factors affecting cell behavior. Biomimetic oriented electrospinning fiber scaffold can provide contact guidance *in vivo* or *in vitro* for the growth and extension of

axons of *in situ* neurons or induced differentiated neuron-like cells. As shown in Fig. S2C, the orientations of the three groups of scaffold fibers at 0° were 10799.59±1152.65, 30435.12±3440.99, 49274.01±2292.76, which indicates that the fiber orientation of each group is highly consistent (page 9, line 221).

Figure S2C. Analysis of orientation of different fiber scaffolds. ($n=3$, orientation values were mean \pm std. dev)

Porosity method: The degree of porosity of the each group scaffolds was determined by the pycnometer method following the procedure described elsewhere⁹. Briefly, the weight of the pycnometer filled with ethanol was measured and labeled as W_1 ; the sample with a weight of W_s , and was immersed in ethanol. Subsequently, the sample was saturated by ethanol; additional ethanol was added to complete the volume of the pycnometer. Then, the pycnometer was weighted and labeled as W_2 ; the sample filled with ethanol was taken out of the pycnometer and the residual weight of the ethanol and the pycnometer was labeled as W_3 . The porosity of the membrane was calculated according to:

$$\mathcal{E} = \frac{W_2 - W_3 - W_s}{W_1 - W_3} \times 100\%$$

The porosity of each scaffold was obtained as the mean value of the porosity determined in three samples (Supplementary information, page 8, line 197).

Results and discussion: One of the advantages of electrospinning fiber scaffolds becoming a hot spot in tissue engineering research is that it has large porosity, which can provide more growth space for cells. As mentioned above, there is no statistically significant difference in the diameter of the three groups of fiber scaffolds ($p > 0.05$), so the factors that change the porosity due to different diameters can be excluded. The experimental group MSaP-aL/p compared with aP and MSaP two groups, the porosity increased significantly ($p < 0.05$) (Fig. S3C). This phenomenon can also be explained by the lipophilicity of aminated PLA and phospholipid, during the grafting process, some adjacent fibers fused, and the internal bulk density decreased, resulting in increased porosity (page 11, line 278).

Figure 3C. Porosity of different fiber scaffolds. ($n=3$, porosity values were mean \pm std. dev and p values were determined by one-way analysis of variance (ANOVA) with a Tukey's post-hoc test)

(3). In this study, we hoped that the IL-4 plasmid loaded with aldehyde cationic liposomes will play a transfection role during the acute early stage of spinal cord injury to secrete IL-4, and immunoregulatory the local microenvironment of spinal cord injury to pave the way for long-term neural regeneration. In this article, we discussed the encapsulation efficiency, release characteristics, and biological activity of the IL-4 plasmid from the following three aspects:

a. In the process of characterizing the biomimetic fiber scaffold, we used a dsDNA quantification kit to assess the concentration change of the IL-4 plasmid before and after grafting. The initial concentration of the IL-4 plasmid was $1.16 \pm 0.06 \mu\text{g}/\mu\text{l}$. The concentration was $0.83 \pm 0.01 \mu\text{g}/\mu\text{l}$ (approximately 71.49%) after forming the pDNA/aLiposome complex, with $0.23 \pm 0.01 \mu\text{g}/\mu\text{l}$ (approximately 19.38%) grafted on the fiber membrane by Schiff base bond (Figure 4D). Subsequently, the response characteristics of the biomimetic composite fiber membrane showed that $0.19 \pm 0.02 \mu\text{g}/\mu\text{l}$ (approximately 85.72%) was released within 24 hours in an acidic environment at pH 5.8 (Figure 4E). This result indicates that the composite fiber scaffold rapidly responds to release a large number of liposomes for immunoregulation in an acidic environment, and as mentioned above, it coincides with the time window of the occurrence of blood-derived macrophages in the spinal cord injury.

Figure 4D, E. (D) Concentration of pDNA in the aliposome solutions before and after grafting at the surface of scaffold. (E) Cumulative release curve of aliposomes in different pH environments. ($n=3$, aliposome release values were mean \pm std. dev., $p < 0.05$ when comparing pH5.8 and the other two pHs via two-way analysis of variance (ANOVA) with a Tukey's post-hoc test)

b. In the *in vitro* cell assay, macrophages were stimulated by IL-4 cytokines after transfection of the IL-4 plasmid in the co-culture environment at pH 5.8. Compared with other groups, MSaP-aL/p significantly decreased the pro-inflammatory genes expression while increased the anti-inflammatory genes expression at day 3 ($p < 0.05$) (Figure 5A, D and Figure 5 G, J), and similar trend could be found on the secretion of pro-inflammatory and anti-inflammatory factors (Figure S6A, D and Figure S6G, J).

Figure 5A, D, G, J. Evaluation of pro-inflammatory and anti-inflammatory factor gene expression of BMMs in different pH culture environments to simulate the microenvironment of spinal cord injury. The expression of IL-1 β mRNA (A), TNF- α mRNA (D), IL-10 mRNA (G), TGF- β mRNA (J) under pH5.8. ($n=3$, all values were mean \pm std. dev., * $p < 0.05$ and ** $p < 0.01$ when comparing MSaP-aL/p and other groups via two-way analysis of variance (ANOVA) with a Tukey's post-hoc test)

Figure S6. Evaluation of pro-inflammatory and anti-inflammatory factors secreted by BMM at different pHs by ELISA. IL-1 β (A), TNF- α (D), IL-10 (G), TGF- β (J) under pH5.8. ($n=3$, all values were mean \pm std. dev., Comparison between groups via two-way analysis of variance (ANOVA) with a Tukey's post-hoc test)

c. *In vivo*, we collected 1w postoperative spinal cord samples for immunofluorescence labeling of IL-4, TNF- α (M1 macrophages/microglia-specific cytokines), and IL-10 (M2-type macrophages/microglia-specific cytokines) (Figure 9A, B, C) and flow cytometry (FCM) to evaluate macrophages/microglia cell subtype changes in damaged tissues (Figure 8). Also, qRT-PCR was employed to test the expression of immune inflammation-related genes in damaged tissues (Figure 9D-H), followed by ELISA quantification of IL-4, pro-inflammatory and anti-inflammatory factors production in animal serum (Figure 9I-M). The above experiments have proved that the IL-4 plasmid plays an immunoregulatory role after implantation and improves the local environment of spinal cord injury.

Figure 8. Phenotypic identification of immune cells in vivo. (A), Flow cytometry analysis of CD11b/CD86-positive cells with (B) its quantification analyzed of the ratio and CD11b/CD206-positive cells with (C) its quantification analyzed of the ratio. (n=3, all values were mean \pm std. dev., #p<0.05 when comparing non-immunoregulatory fiber groups and blank control group and *p<0.05 when comparing MSaP-aL/p and other control groups via one-way analysis of variance (ANOVA) with a Tukey's post-hoc test., ns, not significant)

Figure 9. Evaluation of immune regulation 7 days after operation. (A) IL-4 immunofluorescence single labeling and TNF- α , IL-10 immunofluorescence double labeling. Quantitatively analysis of (B) IL-4 immunofluorescence, (C) IL-10 immunofluorescence, (D) IL-4 mRNA Expression, (E) TNF- α mRNA Expression, (F) IL-10 mRNA Expression, (G) IL-4 mRNA Expression, (H) TNF- α mRNA Expression, (I) Serum IL-4, (J) Serum TNF- α , (K) Serum IL-10, (L) Serum IL-10, (M) Serum TGF- β 1.

immunofluorescence and TNF- α immunofluorescence. ($n=5$, all values were mean \pm std. dev., * $p < 0.05$ when comparing MSaP-aL/p and other groups via two-way analysis of variance (ANOVA) with a Tukey's post-hoc test, scale bars are 50 μ m) Spinal cord specimens of IL-4 (D), TNF- α (E), IL-1 β (F), IL-10 (G) and TGF- β (H) gene expressions were assessed by qRT-PCR. Serum levels of IL-4 (I), TNF- α (J), IL-1 β (K), IL-10 (L), and TGF- β (M) were assessed by ELISA. ($n=3$, qRT-PCR and ELISA values were mean \pm std. dev., * $p < 0.05$ when comparing MSaP-aL/p and other groups via one-way analysis of variance (ANOVA) with a Tukey's post-hoc test)

Based on the above three points, we could conclude the IL-4 plasmid released by the biomimetic responsive immunoregulatory fiber scaffolds during acute spinal cord injury participates in the immune response process, timely regulates blood-derived macrophages and in situ microglia subtypes, and improves the inflammatory environment to provide favorable conditions for subsequent long-term neural regeneration.

(4). As mentioned in the manuscript, the microsol electrospinning technology is based on the fact that the organic solvent evaporates faster than water during the spinning process. HA hydrosol moves and stretches inside the PLA to form a core-shell structure which could be able to protect the internal drug or cytokine activity from interference and destruction from external environment¹⁰.

a. In this study, the NGF loading efficiency was 79.53 \pm 1.44%, and we simulated the release process of *in vivo* by characterizing the sustained release properties of fiber scaffolds in different pH environments *in vitro* (Figure 4F). The cumulative release results were 320.48 \pm 18.33ng (40.06 \pm 2.29%, pH5.8), 292.69 \pm 25.52ng (36.59 \pm 3.19%, pH6.6) and 166.88 \pm 47.35ng (20.86 \pm 5.92%, pH7.4), which showed that in different acidic environments, MSaP-aL/p had a sudden release in the first two days. The NGF content in the release solution from day 38 to day 40 of each group can still be detected by the ELISA kit, which were 7.89 \pm 2.01ng (Rc (cumulative release)=703.89 \pm 15.24ng, pH5.8), 7.11 \pm 1.39ng (Rc=678.3 \pm 35.33ng) , PH6.6) and 10.24 \pm 1.63ng (Rc=669.87 \pm 38.14ng, pH7.4) respectively.

Figure 4F. Cumulative release curve of NGF in microsol electrospinning in different pH environments. ($n=3$, NGF release values were mean \pm std. dev., statistical analysis evaluated by two-way analysis of variance (ANOVA) with a Tukey's post-hoc test)

b. In order to further characterize the biological activity of NGF released by MSaP-aL/p, as

mentioned above, BMSCs were cultured with the release solution from day 30 to day 40, and the results showed that differentiated neuron-like cells could be labeled by neuronal specific marker tuj-1 after 5 days (Figure S8) indicating the bioactivity of controlled released NGF. The above results proved that the biomimetic microsolv electrospun fibers could sustained release NGF and provide neural differentiation biological information for stem cells.

Figure S8. Assessing long-term biological activity of NGF in biomimetic fibers. (A) Immunofluorescence staining of Tuj-1 staining images on cell culture plate (NC), release solution group (r-NGF) and cytokine NGF group (PC) with (B), Corresponding fluorescence semi-quantitative analysis. (n=3, all values were mean ± std. dev., *p<0.05 when comparing r-NGF, PC and NC group via one-way analysis of variance (ANOVA) with a Tukey's post-hoc test., ns, not significant, scale bars were 200 μm)

Questions and comments raised by the animal study and its outcomes:

- The spinal cord implant is insufficiently detailed in the manuscript, being very difficult to understand its size, the shape, how it is fixed in the defect site,, how it is ensured a good and reproducible coverage of the hemisection defect created, the alignment of the fibres in the implant, the confinement at the defect site.

Re. Thank you for the comment. We modified and supplemented the surgical procedure for rat spinal cord injury model as follows :

The spinal cord hemi-section model was established following the procedure described previous article ¹¹. Briefly, the rats were anesthetized by intraperitoneal injection of 2% Pentobarbital (50mg/kg). After anesthesia, a 3cm longitudinal incision centered on T9 was made on the back of the rat, and the paraspinal muscles were separated. After removal the

T9 lamina was fully exposed to the spinal canal, under the accurate measurement of the scale, the right half of the spinal cord was cut off to create a 3mm spinal cord hemi-section defect. Followed by saline irrigation, the fiber bundles (2mm×2mm×3mm) placed inside the defect (Fig7.A(b, c, d)). After sewing the fascia and skin layer by layer with sutures each rat was intramuscular injected with 2×10^5 units of antibiotics every day for 5 days and manually emptied the bladder every 12 hours. The bare laminectomy was conducted in sham operation group, and spinal cord hemi-section without the implantation of bundle was introduced as the blank control group. Models implanted with aP, MSaP and MSaP-aL were used as the negative control group (Supplementary information, page 13, line 330).

- How many researchers were involved in creating the defects? How was the training of those researchers performed? How consistent is the reproducibility of the defects created in the animal model?

Re : Thank you for the comment. The technique of rat hemisected spinal cord injury model is well-established in our group¹¹. All animal models were constructed by a spine surgeon from the First Affiliated Hospital of Soochow University with extensive clinical experience and solid surgical skills. During the modeling process, the operator used a scale to accurately measure the defect of each rat, and strived to make the spinal cord resection range of each rat the same size in order to minimize the operation error.

- The results reported do not identify the number of animals that were considered for the data reported in the results (both the histology and the functional evaluation of the recovery). Indeed, given the large number of animals tested, if the analysis does not include either all the animals tested or comparable subsets of randomly selected animals, it is very difficult to have a clear perception of the efficacy of this implant.

Re : Thank you for the comment. The histology and functional recovery assessment in the original manuscript listed representative images. At present, we have added the number of samples (n) for testing in each data description, and also corrected the methodological description as shown above.

- Much more details are needed to ensure that the procedures used to obtain the safety and efficacy profile of the implant are robust and representative of the spectra of outcomes obtained for each condition.

Re : Thank you for the comment.

(1). Although liposomes benefited from their low toxicity, low immunogenicity and

genomic safety during transfection, their low encapsulation efficiency and transfection efficiency limit their development. To overcome this bottleneck limitation, we added stearylamine to endow liposomes a positive charge and the stability of liposomes were evaluated using a phospholipid detection kit at different time points within 24 hours after preparation (Figure S1B), which is crucial during the transfection. We then optimized the ratio of pDNA / aLiposome as 1:2.5 for subsequent experiments due to the best transfection efficiency according to the results of immunofluorescence labeling and ELISA (Figure 2).

Figure S1B. Detection of stability of aldehyde cationic liposomes. ($n=3$, all values were mean \pm std. dev)

Figure 2. Evaluation of transfection efficiency of aldehyde cationic liposomes loading different ratio of pDNA. (A) GFP expression indicated by green fluorescence in BMSCs after transfecting 72h. (B) Analysis of nuclear and eGFP fluorescence overlap coefficient rate. (C) Quantification of the IL-4 secretion after 1, 3, 5, and 7 days of transfection. The red marks represented the highest group of transfection efficiency. ($n=3$, all values were mean \pm std. dev., * $p<0.05$ when comparing 1:2.5 and other ratios of pDNA/aLiposome via one-way analysis of variance (ANOVA) with a Tukey's post-hoc test, scale bars were

1000 μm)

(2). We kept all grafted biomimetic composite fiber membranes at $-20\text{ }^{\circ}\text{C}$ to maintain the bioactivity of cytokines and plasmids. The results of simulated release studies in different pH environments *in vitro* showed that the composite fiber scaffold had a sensitive acid response (rapid release of cationic liposomes loaded with IL-4 plasmid and sustained release of NGF), which can provide experimental support for subsequent cell and animal research (Figure 4E, F). *In vitro* experiments, we evaluated the proliferation and adhesion properties of fiber scaffolds using live/dead staining, CCK8 and Integrin β 1 immunofluorescence staining labeling methods, respectively (Figure S4 and Figure S5). Both results showed that fiber scaffolds have favorable biocompatibility. In addition, we implanted each group of fiber membranes into the subcutaneous tissue of the rat's back to further evaluate the foreign body reaction in the animal (Figure S9), and proved the safety of the implantation of fibrous scaffolds.

Figure S4. Cell survival and proliferation assays of membranes. (A) Live/dead (green/red) fluorescence staining images. (all scale bars are 1000 μm) (B) Fluorescence semi-quantitative analysis of living cells. ($n=3$, fluorescence density values were mean \pm std. dev., * $p<0.05$ when comparing control group and four other groups via one-way analysis of variance (ANOVA) with a Tukey's post-hoc test., ns, not significant) (C) Detection of cell proliferation by CCK8 kit at 1, 3 and 5 days. ($n=6$, absorbance values were mean \pm std. dev., * $p<0.05$ when comparing control group and four other groups via two-way analysis of variance (ANOVA) with a Tukey's post-hoc test., ns, not significant)

Figure S5. Cell adhesion assays of scaffolds. (A) Integrin $\beta 1$ immunofluorescence staining images of BMSCs inoculated with fiber scaffolds for 1 day. (B) Fluorescence semi-quantitative analysis of Integrin $\beta 1$. (C), (D), (E), (F) SEM images of BMSCs cultured on different fiber scaffolds for three days. ($n=3$, integrin $\beta 1$ intensity values were mean \pm std. dev., * $p < 0.05$ when comparing control group and four other groups via one-way analysis of variance (ANOVA) with a Tukey's post-hoc test., ns, not significant, fluorescent images scale bar were 100 μm and 50 μm in SEM images)

Figure S9. Characterization of foreign body reaction of fiber scaffolds. (A) General observation and corresponding H&E staining after subcutaneous implantation on the back of rats for 2 weeks. The yellow arrow refers to the inflammatory response area around the fiber scaffolds. (B) Analysis of inflammatory band area of HE staining by ImageJ software. (C) Schematic diagram of fibrous scaffolds implanted subcutaneously on the back of rats. ($n=3$, all values were mean \pm std. dev., * $p < 0.05$ when comparing

MSaP-aL/p and other groups via one-way analysis of variance (ANOVA) with a Tukey's post-hoc test., ns, not significant, scale bars were 50µm)

Other issues of the manuscript:

- The authors described similar information on the Introduction and on the Results and Discussion sections. Please avoid duplication of information along the manuscript, by providing relevant background information in the Introduction section.

Re : Thank you for the comment. We have made the following modifications to the duplicates and highlighted in the revised manuscript.

(1). **Results and discussion opening remarks.** *The bioinspired microenvironment-responsive fiber has been constructed step-by-step as shown in scheme 1. In order to verify the effect of materials, physiochemical property, in vitro and in vivo biological characteristics of different combination were characterized comprehensively in this study with the group setting depicted in Table 1 (page 6, line 138).*

Table 1. Material grouping.

Group	Denoted	Function
PLA electrospinning	PLA	Control
Amino PLA electrospinning	aP	Control
Amino PLA microsolv electrospinning	MSaP	Control
Amino PLA microsolv electrospinning carrying blank cationic liposome	MSaP-L	Control
Amino PLA microsolv electrospinning carrying blank aldehyde cationic liposomes	MSaP-aL	Control
Amino PLA microsolv electrospinning carrying aldehyde cationic liposomes loading IL-4 plasmid	MSaP-aL/p	Test

(2). **Biomimetic composite fiber simulates the release of liposomes in vitro.** *Severe injury of local blood vessels caused by external mechanical damage of spinal cord will lead to a sharp decrease of pH, resulting in an acidic microenvironment in the injured area. Based on this pathological characteristic, the focal pH was simulated in vitro for the characterization of microenvironment-responsive releasing functionality. Specifically, the releasing rates of liposome was tested through immersing the liposome-grafted fiber scaffold in PBS solutions with different pH (Fig. 4E). In the acidic environments with pH of 5.8 and 6.6, the release of liposomes increased gradually with time. The release rate of liposomes under the pH of 5.8 reached 61.66±1.38% within 10 hours, which was significantly higher than that of other pHs (p<0.05). Therefore, quick respond to acidic environment of the responsive fiber scaffold could be verified in vitro which provided basis for further study (page 12, line 311).*

Figure 4E. Cumulative release curve of aliposomes in different pH environments. ($n=3$, aliposome release values were mean \pm std. dev., * $p<0.05$ when comparing pH5.8 and the other two pHs via two-way analysis of variance (ANOVA) with a Tukey's post-hoc test)

(3). **Gene expression profiles of BMMs *in vitro*.** The expression of pro-inflammatory and anti-inflammatory genes in BMMs cultured in different pH was detected after co-culture for 1, 3, 5 and 7 days. In the culture medium with pH of 5.8, the expression of pro-inflammatory genes *IL-1 β* and *TNF- α* in MSaP-aL/p group decreased gradually as time went on, reaching the lowest expression of 9.63 ± 0.80 and 0.30 ± 0.03 fold (Fig. 5A, D) on the 7th day respectively. At the same time, the expression of anti-inflammatory gene *IL-10* and *TGF- β* were 3.42 ± 0.07 , 11.59 ± 0.09 , 27.02 ± 0.45 , 29.30 ± 0.51 fold and 2.85 ± 0.06 , 7.99 ± 0.07 , 22.52 ± 0.37 , 27.47 ± 0.59 fold on the 1st, 3rd, 5th and 7th day, respectively. Expression of all anti-inflammatory gene showed an upward trend (Fig. 5G, J), which was significantly different from that of groups ($p<0.01$). Under the pH of 6.6, MSaP-aL/p group also showed response to acidic environment. However, due to the decreased acidity, the response of Schiff base to environment decreased, followed by relatively less significant changes on pro-inflammatory and anti-inflammatory gene expression, compared with those under pH of 5.8 (Fig. 5B, E, H, K). Under the pH of 7.4, the differences on pro-inflammatory and anti-inflammatory genes expression was smaller and showed insignificant difference between different groups at each time point ($p>0.05$). Surprisingly, pro-inflammatory genes, *IL-1 β* and *TNF- α* , showed a downward trend while anti-inflammatory genes, *IL-10* and *TGF- β* , showed an upward trend (Fig. 5C, F, I, L). This could be explained that the lower chamber BMSCs had a certain degree of immunoregulation function, and could shift M1 macrophages into M2 macrophages, and inhibit the expression of inflammatory genes (page 14, line 377).

Figure 5. Evaluation of pro-inflammatory and anti-inflammatory factor gene expression of BMMs in different pH culture environments to simulate the microenvironment of spinal cord injury. The expression of (A), (B), (C) IL-1 β mRNA, (D), (E), (F) TNF- α mRNA, (G), (H), (I) IL-10 mRNA and (J), (K), (L) TGF- β mRNA at different pHs of BMMs. ($n=3$, all values were mean \pm std. dev., * $p < 0.05$ and ** $p < 0.01$ when comparing MSaP-aL/p and other groups via two-way analysis of variance (ANOVA) with a Tukey's post-hoc test)

(4). **BMSCs differentiate into neuron-like cells *in vitro*.** Endogenously secreted NGF was responsible for the maintenance of multiple neurophysiological functions. BMSCs was introduced to mimic endogenous stem cell and investigate the biological effect of NGF-loaded fibers. After cultured on different fiber membranes for 10 days, immunofluorescence staining and qRT-PCR technique were employed to detect the expression of neuron-specific markers in neuron-like cells, including neurofilament protein (NF-200), neuron specific enolase (NSE), nerve cytoskeleton tubulin (Tau protein) and neuron cell specific differentiation tubulin (Tuj-1), in order to evaluate the neurogenic activity of fiber membranes (Figure 6) (page 16, line 422).

Figure 6. Neuron-specific marker immunofluorescence staining and gene detection assays of scaffolds. (A) Immunofluorescence staining of Tuj-1, NSE, NF-200, Tau, staining images on different fiber scaffolds with (B), (C), (D), (E) Corresponding fluorescence semi-quantitative analysis. (F), (G), (H), (I) mRNA expression analysis of Tuj-1, NSE, NF-200, Tau. (n=3, all values were mean ± std. dev., * $p < 0.05$ when comparing microsol electrospun groups (MSaP, MSaP-aL, MSaP-aL/p) and control, aP group respectively via one-way analysis of variance (ANOVA) with a Tukey's post-hoc test., ns, not significant, scale bars were 100 μm)

- The achieved results are poorly compared to the achievements of others already described in the literature.

Re : Thank you for the comment. Studies have shown that *in vitro* biomaterials carrying neural-promoting cytokines can induce neural differentiation of stem cells. However, when the cytokine-loaded biomaterials were implanted into the spinal cord injury site of animals, it was found that the immune inflammatory response in situ and the foreign body reaction brought by the implant significantly compromised the regeneration of neural tissue, and even the loaded cytokines would lose bioactivity in the malignant pathological environment of spinal cord injury¹². Therefore, researchers had focused on reducing the local immune response to spinal cord injury for better prognosis¹³. However, the single use of immune regulation measures to suppress the inflammatory response can only prevent the further development of spinal cord injury. It cannot provide biological signals that promote neuronal regeneration during the later-stage repair and regeneration process of central nervous tissue, bringing the bottleneck in the research of neural tissue engineering treating spinal cord injury. In our previous research, it was found that oriented

electrospinning scaffolds can contact guidance nerve cells well ¹⁴, and microsol electrospinning technology had excellent sustained-release properties and the function of core-shell structure could protect the biological activity of cytokines ¹⁵. Hence, in order to verify the immunoregulating effect of responsive transfection system, we grafted the cationic liposome onto this oriented microsol fibers which had been thoroughly studied and well recognized by our group. Since the pH-responsive immunoregulating function played the key role in this study, the oriented microsol fiber basically acted as a platform instead of the key point of the study. Despite that our study may not be comparable to those based on other platforms and materials, we've made our point on confirming the satisfying effect of pH-responsive immunoregulating system in this study. In order to achieve better outcome in the treatment of spinal cord injury, platforms made of better biomaterials and endowed with different attributes would be explored in our future experiment.

- Approximately 90% of IL-4 plasmid release is achieved in 24h at pH 5.8 (Figure 4 E). How relevant the action of IL-4 when kept under culture or implanted for longer time periods?

Re : Thank you for the comment. In situ microglia activation and macrophage infiltration appeared within 24 hours after acute spinal cord injury, reached the peak within 3 days and 7 days respectively ⁵, followed by differentiated into type M1 under the regulation of local environment, which can release inflammatory factors to aggravate local inflammatory response, promote apoptosis of surviving tissue cells, induce astrocyte activation and expand the scope of lesions, resulting in secondary spinal cord injury. However, the infiltrating macrophages could be induced to differentiate into M2, secreting biological factors such as anti-inflammation and promoting tissue regeneration ¹⁶. Based on the above pathological basis of spinal cord injury, this study hoped that cationic liposomes loaded with IL-4 plasmid could play an immunomodulatory role in a short time, promote local tissue to secrete IL-4, so that local and infiltrating related immune cells can shorten M1 action phase and differentiate into M2 at earlier stage, so as to reduce acute inflammatory reaction and reduce the release of toxic substances and retain a glimmer of vitality for later nerve regeneration. Hence, according to the physiological features, the effect of plasmid release at later stage after implantation was not our interest when characterizing immunoregulating capability.

In the release assay, the acid-sensitive Schiff base bond rapidly responded to the pH5.8 environment and released the 90% IL-4 plasmid within 24 hours(Figure 4E), and the results of *in vitro* BMMs polarization study (Figure 5, Figure S6, Figure S7) together with the results of *in vivo* immunoregulation effect characterization 1 week post-operation (Figure 8 and Figure 9) all supported the conclusion that the biomimetic composite fiber had an immunomodulatory function during acute spinal cord injury.

- Why was the effect on the pro-inflammatory and anti-inflammatory expression only observed on

the 3rd day of BBM culture (Figure 6 G and J)

Re : Thank you for the comment. As mentioned above, the expectation of this study is that immunoregulation during acute spinal cord injury provides a favorable microenvironment for subsequent neural regeneration. *In vitro* cell assay, biomimetic responsive immunoregulatory fiber membranes in response to pH5.8 culture environment released IL-4 plasmid loaded cationic liposomes transfected with mesenchymal stem cells in the lower chamber of Transwell plate, and secreted IL-4 to exert immune regulation on bone marrow macrophages (BMMs) in the upper compartment. Figure 5G, J showed that there was no significant difference in the expression of anti-inflammatory gene (IL-10, TGF- β) among the groups on the first day ($p > 0.05$). We speculate that this stage is in the response phase and cell transfection stage of the material. However, on the 3rd, 5th and 7th day, MSaP-aL/p anti-inflammatory genes were highly expressed, and there was significant difference compared with other groups ($p < 0.05$). The above *in vitro* experimental results supported that the biomimetic composites we constructed could respond positively to the acidic environment and induce BMMs to differentiate into M2.

Figure 5G, J. Evaluation of pro-inflammatory and anti-inflammatory factor gene expression of BMMs in different pH culture environments to simulate the microenvironment of spinal cord injury. The expression of IL-10 mRNA (G), TGF- β mRNA (J) under pH5.8. ($n=3$, all values were mean \pm std. dev., * $p < 0.05$ and ** $p < 0.01$ when comparing MSaP-aL/p and other groups via two-way analysis of variance (ANOVA) with a Tukey's post-hoc test)

- Since approximately 50% of IL-4 plasmid is released in 24h at pH 6.6 (Figure 4 E), how you explain the few differences regarding IL-10 and TGF-b expression at pH 5.8 and pH 6.6 (Figure 6 G and J vs H and K)?

Re : Thank you for the comment. Figure 6 in the original manuscript is now placed in Figure 5 as shown below. The simulated release showed that about 90% loaded IL-4 plasmid cationic liposomes were released under pH 5.8 environment within 24 hours,

while the cumulative release under pH6.6 was about 50% (Figure 4E), which had a significant difference in response and release between the two. Therefore, under the condition of co-culture of pH5.8 and pH6.6, there was a significant difference in the concentration of cationic liposomes for transfection of stem cells. According to the experimental results, Figure 5G showed that the expression of IL-10 gene in MSaP-aL/p group increased significantly on the 3rd day (11.59 ± 0.09 fold), 27.02 ± 0.45 fold on the 5th day and 29.30 ± 0.51 fold on the 7th day. In the pH6.6 culture condition, Figure 5H showed that the expression of IL-10 gene was 4.37 ± 0.04 , 17.09 ± 0.10 and 22.29 ± 0.04 fold on the 1st, 3rd and 5th day respectively. However, under pH 5.8 culture conditions (Figure 5J), the gene expression of TGF- β was 2.85 ± 0.06 , 7.99 ± 0.07 , 22.52 ± 0.37 , 27.47 ± 0.59 fold on the 1st, 3rd, 5th, and 7th days, and the pH was 6.6, the expression of TGF- β on the 1st, 3rd, 5th, and 7th days (2.60 ± 0.05 , 3.64 ± 0.03 , 14.24 ± 0.08 , 18.57 ± 0.03 fold) had significant differences (Figure 5K). As shown in the figure below, at the same time in the MSaP-aL/p group, the expression levels of the anti-inflammatory genes IL-10 and TGF- β at pH5.8 were significantly higher than the expression levels at pH6.6 ($p < 0.05$).

Measurement of IL-10 and TGF- β gene expression in BMM of MSaP-aL/p group in different pH environments

Figure 5. Evaluation of pro-inflammatory and anti-inflammatory factor gene expression of BMMs in different pH culture environments to simulate the microenvironment of spinal cord injury. The expression of (A), (B), (C) IL-1 β mRNA, (D), (E), (F) TNF- α mRNA, (G), (H), (I) IL-10 mRNA and (J), (K), (L) TGF- β mRNA at different pHs of BMMs. ($n=3$, all values were mean \pm std. dev., * $p < 0.05$ and ** $p < 0.01$ when comparing MSaP-aL/p and other groups via two-way analysis of variance (ANOVA) with a Tukey's post-hoc test)

- The authors claim that the microsol electrospinning allows for a prolonged release of protein molecules (6 weeks) (based on a previous authors' work on chloroquine-based DDS). However, approximately 80% of NGF is released in 25 days (Figure 4 F). Please revise the claims regarding the microsol electrospinning advantages having in consideration the present results. The results should be also presented in terms of the effective amount of protein released over time and also its bioactivity.

Re : Thank you for the comment. We've revised the claim according to your advice. Also, the release of NGF from both quantitative and qualitative aspects was thoroughly characterized.

(1). As described above, the NGF *in vitro* simulated release solution was quantified by ELISA technology at each predetermined time point (Figure 4F).

(2). Currently, we had added the characterization of NGF concerning its biological activity after the release. As shown above, we used fiber membranes release solution from the 30th to the 40th day (the longest period of time) to induce the differentiation of bone marrow mesenchymal stem cells for five days. As a result, NGF-induced neuron-like could be

identified through fluorescently labeled by Tuj-1 (Figure S8), and the results of semi-quantitative analysis also showed significant statistical difference from the negative control group.

- How did the authors acquire the BMSCs? Is the procedure previously approved by the Ethical Committee of the certified institution for animal experimentation? Are those isolated from a single donor or pooled? Did the authors assess BMSCs putative stem cell potential?

Re : Thank you for the comment and reminding. Rat bone marrow mesenchymal stem cells (BMSCs) were provided by the Institute of Orthopaedics of the First Affiliated Hospital of Soochow University, with corresponding extracting procedure certified and approved by the Ethics Committee.

The BMSCs used in the study were isolated from pooled donors. Before employed in *in vitro* assessment, BMSCs were subjected for flow cytometry to identify the surface specific markers and confirm their purity. Flow cytometry method and results are described below.

Method : 3rd generation bone marrow mesenchymal stem cells were collected, after washed with PBS, cells were treated with American Hamster anti-rat-PE CD29 (Invitrogen, 12-0291-81), mouse Brilliant Violet 650TM anti-rat CD90 (Biolegend, 202533), mouse anti-rat-FITC CD34 (Novus, NBP2-47911F) and Alexa Fluor[®] mouse anti-rat CD45 (Biolegend, 202212) flow cytometry antibodies (1: 100). One negative control group and three separate experimental groups were set for each antibody. Putative identification by flow cytometry after incubation at 4 °C for 30 minutes.

Results: In order to confirm the purity of BMSCs employed for *in vitro* study, the molecular markers on the cell surface were detected by flow cytometry which showed high expression of CD29 and CD90, and low expression of CD34 and CD45, indicating the high purity of bone marrow mesenchymal stem cells as well as the feasibility of employing these cells in following studies ¹⁷.

Identification of BMSCs purity by flow cytometry

- The neurogenic activity of the testing condition was assessed by culturing the BMSCs onto the NGF-loaded fibrous scaffolds for 10 days. The authors used qRT-PCR technique to assess the expression of earlier neuron-specific markers in neuron-like cells. To increase the significance of these results, it is suggested to assess the expression of lately expressing neuronal markers.

Re : Thank you for the suggestion. Spinal cord injury repair occurs in a complex pathological environment, the study of biomimetic fiber membranes *in vitro* can only preliminarily simulate the early processing of nerve regeneration *in vivo*. In order to more accurately evaluate the function of biomimetic fibers in promoting nerve regeneration, we have added the *in vivo* immunofluorescence staining of growth associated protein-43 (GAP-43), a lately expressing neuronal marker, 4w and 8w after surgery, which could reflect the neurogenic activity at later stage. The supplemented content in Method and Result & Discussion section were provided as follow ¹⁸:

Method: *As for the immunohistochemical evaluation, the antigen was repaired with 0.3% hydrogen peroxide and nonspecific antigen was blocked by 5% BSA. The primary antibody (rabbit anti GAP-43 (Abcam, ab75810) for axon staining) was added and incubated at 4 °C overnight, and the second antibody (goat anti-rabbit IgG (Alexa Fluor 647, Abcam, ab150079) was incubated at room temperature for 1 hour after PBS washing. The samples were observed and photographed under fluorescence microscope, with the fluorescence intensity analyzed with ImageJ software. The values were represented in the form of average optical intensity, which had been corrected for optical density, de-background, and finally normalized the staining intensity by the number of cell nuclei. A total of five images were randomly selected from each group for statistical analysis (n=5) (Supplementary information, page 17, line 426).*

Results and discussion: *Under normal physiological conditions, low expression of GAP-43 only maintains normal function of spinal cord, but GAP-43 is highly expressed in the process of nerve development and regeneration, which can regulate axonal extension, enhance nerve plasticity and release of neurotransmitters (Fig. 10C). Therefore, the fluorescence density of sham group was lower than that of other groups at the same time ($p < 0.05$) (Fig. 10D). Meanwhile, the fluorescence density of GAP-43 in MSaP-aL/p group was significantly higher than that in other control groups, which proved that immunoregulation of local inflammatory response during acute spinal cord injury could promote nerve regeneration for a long time ($p < 0.05$). In addition, aP group was lower than that in the microsol electrospun groups (MSaP, MSaP-aL, MSaP-aL/p) at the same time ($p < 0.05$), suggesting that continuous supply of NGF could significantly promote nerve repair. Surprisingly, with the development of time, the expression of GAP-43 in all experimental groups except sham group at 4 weeks were significantly higher than those at 8 weeks ($p < 0.05$). The results suggested that there is a limited time window for the repair of spinal cord injury. Immune regulation to reduce inflammatory injury in the stage of*

acute spinal cord injury can better promote nerve regeneration (page 22, line 613).

Figure 10C, D. Immunofluorescence staining of neuron cells and axons. (C), Immunofluorescence staining of axon with (D) its quantification analyzed with optical density. ($n=5$, GAP-43 intensity values were mean \pm std. dev., # $p<0.05$ when comparing 4w and 8w in blank control groups, aP, MSaP, MSaP-aL and MSaP-aL/p; @ $p<0.05$ when comparing sham group and other groups; & $p<0.05$ when comparing aP and microsol electrospun groups (MSaP, MSaP-aL and MSaP-aL/p), * $p<0.05$ when comparing MSaP-aL/p and other control groups via two-way analysis of variance (ANOVA) with a Tukey's post-hoc test, scale bars were $50\mu\text{m}$)

- During neuronal differentiation, neurite extension and outgrowth occurs. Therefore, it is important to assess the neurite outgrowth by counting the Neurite-bearing cells.

Re : Thank you for the comment. In neurorepair studies, it is necessary to count the number of neurites during neuron regeneration. The literature reports that after induction of spinal dorsal root ganglion (DRG) and PC12, the neurite extension and growth can be clearly observed in the differentiated neuronal cell morphology^{3, 19}. However, in this study, in order to explore the function of biomimetic fibers in promoting nerve regeneration, bone marrow mesenchymal stem cells were selected as seed cells to differentiate into neuron-like cells induced by NGF. Due to the limitation of cellular characteristics, neuron-like cells differentiated from mesenchymal stem cells are similar to neuronal cells in protein molecule and gene expression, but the morphological changes are not completely consistent. To the end, for better assessment, after the neural differentiation of bone marrow mesenchymal stem cells were induced by microsol electrospinning groups *in vitro*, higher expression of neuron specific marker gene and corresponding protein fluorescence semi-quantitative were obviously observed (Figure 6). *In vivo*, Tuj-1-labeled neurons and GAP-43-labeled axon sprouting were observed at 4 and 8 weeks (Figure 10). Based on the current results of *in vitro* and *in vivo*, it has been preliminarily confirmed that biomimetic composite fibers possess the capacity of continuously promoting nerve regeneration. We will use nervous system-specific cell lines to assess the neurite outgrowth in the further study of the mechanism of biomimetic fibrous membrane promoting nerve

regeneration.

- From the results of Figure 8 B and C, it is important to denote the minimum time needed to achieve similar motor function. Please discuss it accordingly.

Re : Thank you for the thoughtful comment. As the reviewer mentioned earlier, to have a clear result, it is necessary to show the best, worst and more frequent experimental results. When the original manuscript was first submitted, we realized that the sample size in the animal motor function scoring part was inappropriate (n=3), so we repeated this part of assay and increased the sample size (n=7) so as to highlight the significance of our study. We are sorry for this mistake. Figure 8 in the original manuscript is placed in Figure 7 and the revised version in different sections and figure caption were provided as follows:

Method: *The recovery of neuromotor function of hindlimb in six groups of rats was evaluated by BBB (Basso, Beattie, Bresnahan) and IPT (inclined plane test) scoring system. The score range of BBB scale was 0 to 21 with 0 indicating no hindlimb movement, and 21 indicating that motor function was normal. During each test, two unwitting examiners evaluated separately each week after the operation. As a supplementary study of BBB scoring test, IPT can improve the effectiveness and sensitivity of scoring. Using Rivlin method, the rats were placed on the inclined plate with rubber pad, the longitudinal axis of the rat was kept parallel to the longitudinal axis of the inclined plate, the head was raised to one side, and raised 5° from 0° within 5 seconds. The highest angle of stay of the rats was recorded and the larger the angle indicated the stronger the load-bearing capacity of the lower extremities. The above two scoring methods are performed by two blinded observers and seven rats were randomly selected from each group at a fixed time point every week for evaluation (n=7) (Supplementary information, page 16, line 405).*

Results and discussion: *The in vivo performance of fiber bundle (Fig.7A (a, b, c, d) was evaluated by SD rat T9 spinal cord hemisection model. The recovery of hindlimb motor dysfunction after spinal cord injury was evaluated by BBB and IPT scores every week after operation. The results showed that the hindlimb motor function of rats in each group recovered in varied degrees. Superior recovery of motor function could be identified in MSaP-aL/p group compared with other groups, as indicated by significantly higher BBB score and IPT score at each time points from 4th week post-injury ($p < 0.05$). At the 8th week, the BBB score of 13.71 ± 1.11 , and the IPT score of $55.86 \pm 4.67^\circ$ could be found in MSaP-aL/p group, both of which were significantly higher than those of the other groups ($p < 0.05$) (Fig. 7B, C). In addition, the minimum time required to achieve similar motor functions during recovery is more convincing in characterizing the functionality of neural repair materials. Therefore, in the comparison of 10 points (the highest average score of the negative control group in the BBB score) and 47° (the highest average angle of the negative control group in the IPT test), the two scores of the MSaP-aL/p group exceeded the control values at the 4th week after operation, reaching 11.14 ± 2.04 (BBB) and 52.43 ± 5.56 (IPT) respectively. The results of motor function recovery suggested that*

immunoregulatory fiber bundle (MSaP-aL/p) implantation could reduce the risk of further damage to surviving motor neurons by effectively inhibiting the acute inflammatory response of spinal cord injury, and promote nerve repair subsequently (page 18, line 482).

Figure 7. (B) Evaluation of motor function recovery of lower limb by BBB score in rats assisted by (C) rats motor function IPT score. (n=7, all values were mean \pm std. dev., * $p < 0.05$ when comparing MSaP-aL/p and other groups via two-way analysis of variance (ANOVA) with a Tukey's post-hoc test)

- In Figure 8B, please clarify the significant differences between the conditions MSaP-aL/P and MSaP-aL. They seem very similar, namely when considering the SD at each time point.

Re : Thank you for the comment. In order to clearly observe the recovery of animal motor function, we've increased the sample size for analysis (n=7), and highlighted the corrected content in the manuscript as described above.

In terms of composition, MSaP-aL/p and MSaP-aL represent loaded or unloaded IL-4 plasmid (Table 1). Functionally, MSaP-aL/p can protect part of surviving motor neurons and inhibit the formation of glial scar tissue by regulating the polarization of infiltrating macrophages and in situ microglia during the acute stage (Figure 7-10), and improve neural progenitor cell effect (Figure S12).

- In Figure 8C, no significant differences were clearly observed. Please revise the statistical analysis, namely testing the normality and variance homogeneity.

Re : Thank you for the comment. Figure 8 in the original manuscript is now placed in Figure 7. In order to observe the statistical differences among groups more clearly, we had reorganized and corrected this part of content as follows.

Figure 7C. rats motor function IPT score. (n=7, all values were mean \pm std. dev., * $p < 0.05$ when comparing MSaP-aL/p and other groups via two-way analysis of variance (ANOVA) with a Tukey's post-hoc test)

- In Figure 9 please provide the results for the same time points (i.e. 4 and 8 weeks), as for other in vivo results (i.e. Figures 10 and 11).

Re : Thank you for the comment. As we've mentioned in the manuscript, the immunofluorescence labeling of IL-4, TNF- α and IL-10 were conducted to evaluate the immunoregulating effect of biomimetic fibers on macrophage polarization at early stage which usually means within 1w after surgery. Similar set of time point on this topic could be found in previous literature concerning immunoregulation^{16, 20}. While the later time points (4w and 8w) for the labeling of activated astrocyte (GFAP) and glial scar tissue (NG2) were employed to observe the long-term effect after regeneration.

(1). In the original manuscript, Figure 9 was the result of evaluating the immunomodulatory function of biomimetic fiber bundles during acute spinal cord injury, so immunofluorescence labeled IL-4, TNF- α (M1 specific marker) and IL-10 (M2 specific marker) in spinal cord specimens 1 week after operation.

(2). Biomimetic fiber bundles will produce long-term biological effects after reducing inflammatory response in the period of acute spinal cord injury. In the previous manuscript, Figure 10A and Figure 10B were activated astrocyte (GFAP) and glial scar tissue (NG2) labeled by immunofluorescence at the same time point (4 weeks and 8 weeks), respectively, and were separated by black dotted lines at different time points. In addition, in order to clearly show the fluorescence quantitative difference among groups at the same time point, we draw the results of each experimental group separately according to the time point,

such as the abscissa display in Figure 10C and Figure 10D (“4w” and “8w”). In order to show the results at the same point more clearly, we have made improvements as follows and the original Figure 10 was placed in Figure S11 in the revised manuscript:

Figure S11. Immunofluorescence staining of activated astrocytes and glial scar. (A), (B) Immunofluorescence staining of astrocytes and quantitatively analyzed with optical density along with (C), (D) their quantification with optical density. (n=5, all values were mean \pm std. dev., * $p < 0.05$ when comparing MSaP-aL/p and other control groups via two-way analysis of variance (ANOVA) with a Tukey's post-hoc test, # $p < 0.05$ when comparing material transplantation groups and blank control group via two-way analysis of variance (ANOVA) with a Tukey's post-hoc test, scale bars were 50 μ m)

(3). Activated astrocytes and scar tissue after spinal cord injury will directly affect nerve regeneration. Following the long-term biological effects of the immune regulation of the above-mentioned fiber bundles, we labeled the migrated neural progenitor cells and differentiated neuron-like neurons at the injured site at the same time point, as shown in Figure S12 and Figure 10A, B. In order to observe the results at the same time more clearly, we have also improved it.

Figure S12. Immunofluorescence staining of neural progenitor cells. (A), Immunofluorescence staining of neural progenitor cells with (B) its quantification analyzed with optical density. ($n=5$, intensity values were mean \pm std. dev., $\#p<0.05$ when comparing sham group and other groups and $*p<0.05$ when comparing MSaP-aL/p and other control groups via two-way analysis of variance (ANOVA) with a Tukey's post-hoc test, scale bars were $50\mu\text{m}$)

Figure 10A, B. (A), Immunofluorescence staining of neuron cells and (B) quantification analyzed with optical density. ($n=5$, Tuj-1 intensity values were mean \pm std. dev., $\%p<0.05$ when comparing blank control and microsol electrospun groups (MSaP, MSaP-aL and MSaP-aL/p); $\&p<0.05$ when comparing aP and microsol electrospun groups (MSaP, MSaP-aL and MSaP-aL/p); $@p<0.05$ when comparing 4w and 8w in MSaP; $\$p<0.05$ when

comparing 4w and 8w in MSaP-aL; and [#] $p < 0.05$ when comparing 4w and 8w in MSaP-aL/p; * $p < 0.05$ when comparing MSaP-aL/p and other control groups via two-way analysis of variance (ANOVA) with a Tukey's post-hoc test, scale bars were 50 μm)

- The authors assess the pro-inflammatory (IL-1 β , TNF- α and IL-10) and anti-inflammatory (TGF- β) cytokines expression of BMM in in vitro assays. However, the authors only assessed the pro-inflammatory cytokines TNF- α and IL-10 in vivo. Please be consistent on the presentation of results along the manuscript.

Re: Thank you for raising the key point. The characterization of pro-inflammatory and anti-inflammatory biological effects *in vitro* and *in vivo* are very important to evaluate the immunomodulatory function of biomimetic fiber membrane. TNF- α and IL-10 are specific secretions of M1 and M2 microglia/macrophages respectively ⁵. As usual, IL-10 is regarded as the anti-inflammatory cytokine. Therefore, in the previous manuscript, Figure 9A listed the representative images of TNF- α and IL-10 in spinal cord specimens of rats in each group with immunofluorescence double labeling one week after operation, as well as the results of semi-quantitative analysis (Figure 9C). The results showed that the fluorescence density of IL-10 in MSaP-aL/p group was significantly higher than that in other groups ($p < 0.05$), while the fluorescence density of TNF- α was significantly lower ($p < 0.05$). Combined with the experimental results *in vitro* and *in vivo*, it was confirmed that the biomimetic composite fiber membrane plays an immunoregulatory function in response to the local acidic microenvironment.

Figure 9A-C. Evaluation of immune regulation 7 days after operation. (A) IL-4 immunofluorescence single labeling and TNF- α , IL-10 immunofluorescence double labeling. Quantitatively analysis of (B) IL-4 immunofluorescence, (C) IL-10 immunofluorescence and TNF- α immunofluorescence. ($n=5$, all values were mean \pm std. dev., * $p < 0.05$ when comparing MSaP-aL/p and other groups via two-way analysis of variance (ANOVA) with a Tukey's post-hoc test, scale bars were 50 μm)

- The Figure 9 quantitative results are based on immunofluorescence images. Following the same presentation approach depicted in Figure 7, please provide mRNA expression results for the in vivo assay.

Re : Thank you for the comment. In order to ensure the consistency of the experimental results *in vivo* and *in vitro*, we added qRT-PCR to evaluate the expression of inflammatory genes in spinal cord specimens of animals one week after operation. The methods and results of gene detection in animal samples are as described above. The results showed that the implantation of biomimetic immunomodulatory fiber bundle could significantly inhibit the expression of pro-inflammatory gene and promote the expression of anti-inflammatory gene.

Figure 9D-H. Spinal cord specimens of IL-4 (D), TNF- α (E), IL-1 β (F), IL-10 (G) and TGF- β (H) gene expressions were assessed by qRT-PCR. (n=3, qRT-PCR values were mean \pm std. dev., * $p < 0.05$ when comparing MSaP-aL/p and other groups via one-way analysis of variance (ANOVA) with a Tukey's post-hoc test)

-For all the immunofluorescence images quantification, it is suggested to normalize the staining intensity by the number of cells nuclei.

Re : Thank you for the comment. As recommended, we have recalibrated all semi-quantitative analyses of immunofluorescence images and corrected them in the manuscript.

- The authors should refer previous works in which liposomes carrying plasmids were immobilised at the surface of electrospun meshes.

Re : Thank you for the comment.

We searched the literatures based on the keywords "electrospun meshes", "liposomes" and "plasmids" in the comment. Three representative literatures were selected for comparison

and discussion.

(1). Monteiro, N. et al., Antibacterial activity of chitosan nanofiber meshes with liposomes immobilized releasing gentamicin. *Acta Biomater.* 18. 196-205 (2015).

In the first article, gentamicin-loaded liposomes were grafted on the surface of chitosan fiber membrane with Maleimide, which proved to have significant antibacterial activity against *Escherichia coli*, *Pseudomonas aeruginosa* and *Staphylococcus aureus*. It was pointed out that it was feasible and advantageous for liposomes to be immobilized on the surface of electrospun meshes to construct nanostructured composite delivery system.

(2). Monteiro, N. et al., Instructive nanofibrous scaffold comprising runt-related transcription factor 2 gene delivery for bone tissue engineering. *ACS Nano.* 8, 8082-8094 (2014).

In the second article, the liposome loaded with RUNX2 plasmid was immobilized on the surface of polycaprolactone fiber membrane to promote bone gene expression for a long time, which was helpful to enhance the tissue regeneration induction of biomaterials.

(3). He, S. H. et al., Multiple release of polyplexes of plasmids VEGF and bFGF from electrospun fibrous scaffolds towards regeneration of mature blood vessels. *Acta Biomater.* 8. 2659-2669 (2012).

In the third article, basic fibroblast growth factor plasmid (pbFGF) and vascular endothelial growth factor plasmid (pVEGF) were loaded into the core-shell structure of emulsion electrospun fiber membrane at the same time. After 4 weeks of sustained release, it was proved that the multiple delivery strategy of biomaterials could significantly promote tissue regeneration.

Corresponding discussion regarding these representative literatures were supplemented in the manuscript and provided as follow:

Studies have shown that combining electrospinning with cationic liposomes would be more effective in antibacterial, inducing bone tissue regeneration and promoting vascular repair^{21, 22, 23}. However, few studies were reported to employ similar strategy to simultaneously control the inflammatory response of acute spinal cord injury and promote nerve regeneration continuously (page 4, line 91).

Reviewer #2 (Remarks to the Author):

The authors provide a refined strategy to target both immune reaction and neural regeneration in spinal cord injuries. The approach is interesting and rather solid.

I recommend to address the following points to facilitate the comprehension of the paper as well as to strengthen the obtained results.

Thank you for your thoughtful reading of our manuscript as well as your constructive comments. We have carefully revised the manuscript, with some new experiments supplemented to ensure the integrity of all the conclusions of this study. Detailed point-by-point responses are provided below.

A lot of the data are described in the manuscript, but I am afraid that 12 figures and 28 pages of manuscript are way too many for a communication. The authors should condense data presentation and panels: some of less significant data could be moved to the supplementary material section.

Re : Thank you for the comment. We considered to place some figures into the supplementary information. There were some changes in these figures as follows:

(1). Figure 5 in the original manuscript is now placed in Figure S4.

Figure S4. Cell survival and proliferation assays of membranes. (A) Live/dead (green/red) fluorescence staining images. (all scale bars were 1000 μm) (B) Fluorescence semi-quantitative analysis of living cells. (n=3, fluorescence density values were mean ± std. dev., * $p < 0.05$ when comparing control group and four other groups via one-way analysis of variance (ANOVA) with a Tukey's post-hoc test., ns, not significant) (C) Detection of cell proliferation by CCK8 kit at 1, 3 and 5 days. (n=6, absorbance values were mean ± std. dev., * $p < 0.05$ when comparing control group and four other groups via two-way analysis of variance (ANOVA) with a Tukey's post-hoc test., ns, not significant)

(2). Figure 10 in the original manuscript is now placed in Figure S11

Figure S11. Immunofluorescence staining of activated astrocytes and glial scar. (A), (B) Immunofluorescence staining of astrocytes and quantitatively analyzed with optical density along with (C), (D) their quantification with optical density. ($n=5$, all values were mean \pm std. dev., * $p < 0.05$ when comparing MSaP-aL/p and other control groups via two-way analysis of variance (ANOVA) with a Tukey's post-hoc test, # $p < 0.05$ when comparing material transplantation groups and blank control group via two-way analysis of variance (ANOVA) with a Tukey's post-hoc test, scale bars were $50\mu\text{m}$)

(3). The Figure 11A of the original manuscript is placed independently in Figure S12.

Figure S12. Immunofluorescence staining of neural stem cells and neuron cells. (A), Immunofluorescence staining of neural stem cells with (B) its quantification analyzed with optical density. ($n=5$, Nestin intensity values were mean \pm std. dev., # $p<0.05$ when comparing sham group and other groups and * $p<0.05$ when comparing MSaP-aL/p and other control groups via two-way analysis of variance (ANOVA) with a Tukey's post-hoc test, scale bars were $50\mu\text{m}$)

(4). Figure 12 in the original manuscript is now in Figure S13, and VWF fluorescently labeled neovascularization was added to this section.

Figure S13. (A), Immunofluorescence staining of vascular endothelial cells and (B) their quantitatively analysis with optical density. (C), Immunofluorescence staining of neovascularization and (D) their quantitatively analysis with optical density. (n=5, all values were mean \pm std. dev., * $p < 0.05$ when comparing MSaP-aL/p and other control groups at the same time and # $p < 0.05$ when comparing 4w and 8w in MSaP-aL/p via two-way analysis of variance (ANOVA) with a Tukey's post-hoc test, scale bars were 50 μ m)

For a better understanding of the impact of their work the authors should state clearly that the present work deals with acute spinal cord injuries.

Re : Thank you for the thoughtful suggestion. We have made corresponding changes in the article to highlight the main purpose of this study is acute spinal cord injury. The specific modifications are as follows.

(1). **Abstract:** *The strategies concerning modification of complex immune pathological inflammatory environment during acute spinal cord injury remain oversimplified and superficial. Inspired by the acidic microenvironment at acute injury site, a functional pH-responsive immunoregulation-assisted neural regeneration strategy was constructed. With the capability of directly responding to acidic microenvironment at focal area followed by triggered release of the IL-4 plasmid-loaded liposomes within few hours to*

suppress the release of inflammatory cytokines and promote neural differentiation of mesenchymal stem cells in vitro, the microenvironment-responsive immunoregulatory electrospun fibers were implanted into acute spinal cord injury rats. Together with slow and sustained release of NGF achieved by microsolv core-shell structure, the immunological fiber scaffolds were revealed to bring significantly shifted immune cells subtype to down-regulate acute inflammation response, reduce scar tissue formation, promote angiogenesis as well as neural differentiation at injury site, and enhance functional recovery in vivo. Overall, this strategy provided a novel delivery system through microenvironment-responsive immunological regulation effect so as to break through the current dilemma from the contradiction between immune response and nerve regeneration, which providing an alternative for the treatment of acute spinal cord injury. (page 2, line 24).

(2). **Introduction:** In the stage of acute spinal cord injury, the balance between M1 and M2 subtypes could be one of the key factors in determining the prognosis of spinal cord injury (page 3, line 56).

In the study of conventional tissue engineering using biomaterials to transmit biological information to promote neural differentiation of stem cells, nerve repair is at risk of failure under the influence of severe immune inflammation exerted by acute spinal cord injury. Therefore, the construction of the bioinspired material that can rapidly respond to local microenvironment during the acute stage and accurately regulate macrophages and microglia cells polarization to reduce inflammatory response, and provide sustained neurogenic platform for stem cells in the later stage may break through the bottleneck of the current treatment of acute spinal cord injury (page 3, line 60).

In this work, inspired by the inflammation and acid-enriched feature of the microenvironment after acute spinal cord injury, a biomimetic fiber scaffold with both immunoregulation function and neurogenic potential was designed to satisfy the specific demand during the acute stage suppression of inflammation and later-stage neural regeneration in scenario of spinal cord injury. (page 5, line 117).

(3). **Conclusion:** The responsive fiber scaffold was endowed with capability of local immune regulation which could respond to the acid microenvironment after the acute spinal cord injury, as well as the sustained promoting effect on neural tissue regeneration (page 23, line 651).

Line 568: there is actually no trustable specific marker for NSC in vivo, Nestin can provide hints of PROGENITOR cells, but not neural stem cells. Tuj-1 marks immature neurons. Therefore I would remove any statement regarding endogenous stem cell mobilization and I will focus on nervous regeneration with newly formed nervous fibers.

Re : Thank you for the thoughtful suggestion. We have removed the content of endogenous stem cell mobilization and made the following changes in the revised manuscript.

(1). **2.4 In vitro biological characteristics of nerve fiber membranes.** In this study, BMSCs and bone marrow macrophages (BMMs) were co-cultured through Transwell to simulate the effects of fiber membranes on cell-to-cell interaction in the microenvironment of acute spinal cord injury, and to evaluate the biological characteristics of nerve fiber membranes (page 13, line 337).

(2). **2.5.4 Histology and immunohistochemical evaluation of spinal cord repair in vivo.** Nestin, Tuj-1 and GAP-43 were used to label neural progenitor cells, neuron cells and axonal sprouting respectively to evaluate the ability of biomimetic bundles to promote nerve regeneration in the middle and late stage of spinal cord injury. The distribution of nestin-labeled neural progenitor cells at the injured site (Fig. S12A) was observed in each bundle group at 4 and 8 weeks after injury. The neural progenitor cells in the sham group was lower than the other groups, which may be due to the low proliferation activity of spinal ependymal cells and the lack of immune response to nestin protein ($p < 0.05$), and the fluorescence quantitative display (Fig. S12B) at the same time point in the MSaP-aL/p group was significantly higher than that in the other control groups ($p < 0.05$). The results showed that biomimetic bundle could significantly inhibit the blocking effect of surrounding scar tissue on neural progenitor cells through immunomodulatory effect. At 4w and 8w time points, neural progenitor cells were continuously promoted by NGF released from microsol electrospun fibers (MSaP, MSaP-aL and MSaP-aL/p), and a large number of Tuj-1-labeled neurons appeared as shown in Fig. 10A (page 21, line 587).

Line 595: I would recommend to add stainings against GAP-43 and SMI31 to better strengthen the findings about the neurogenic potential of the scaffold.

Re : Thank you for the suggestion.

(1). GAP-43 and SMI31 play an indispensable role in nervous system research. Growth associated protein-43 (GAP-43) is a major component of the tips of elongating axon, as well as a suitable marker for labeling axonal sprouting, playing an important role in nerve regeneration²⁴. At present, we supplied GAP-43 fluorescence labeling to visualize regenerated nerve axon sprouting *in vivo*. Corresponding supplemented content in **Method** and **Results and discussion** section was provided as follows:

Method: As for the immunohistochemical evaluation, the antigen was repaired with 0.3% hydrogen peroxide and nonspecific antigen was blocked by 5% BSA. The primary antibody (rabbit anti GAP-43 (Abcam, ab75810) for axon staining) was added and incubated at 4 °C overnight, and the second antibody (goat anti-rabbit IgG (Alexa Fluor 647, Abcam, ab150079) was incubated at room temperature for 1 hour after PBS washing. The samples were observed and photographed under fluorescence microscope, with the fluorescence intensity analyzed with ImageJ software. The values were represented in the form of average optical intensity, which had been corrected for optical density, de-background, and finally normalized the staining intensity by the number of cell nuclei. A total of five images were randomly selected from each group for statistical analysis (n=5)

(Supplementary information, page 17, line 426).

Results and discussion: Under normal physiological conditions, low expression of GAP-43 only maintains normal function of spinal cord, but GAP-43 is highly expressed in the process of nerve development and regeneration, which can regulate axonal extension, enhance nerve plasticity and release of neurotransmitters (Fig. 10C). Therefore, the fluorescence density of sham group was lower than that of other groups at the same time ($p<0.05$) (Fig. 10D). Meanwhile, the fluorescence density of GAP-43 in MSaP-aL/p group was significantly higher than that in other control groups, which proved that immunoregulation of local inflammatory response during acute spinal cord injury could promote nerve regeneration for a long time ($p<0.05$). In addition, aP group was lower than that in the microsol electrospun groups (MSaP, MSaP-aL, MSaP-aL/p) at the same time ($p<0.05$), suggesting that continuous supply of NGF could significantly promote nerve repair. Surprisingly, with the development of time, the expression of GAP-43 in all experimental groups except sham group at 4 weeks were significantly higher than those at 8 weeks ($p<0.05$). The results suggested that there is a limited time window for the repair of spinal cord injury. Immune regulation to reduce inflammatory injury in the stage of acute spinal cord injury can better promote nerve regeneration (page 22, line 613).

Figure 10C, D. Immunofluorescence staining of neuron cells and axons. (C), Immunofluorescence staining of axon with (D) its quantification analyzed with optical density. ($n=5$, GAP-43 intensity values were mean \pm std. dev., $\#p<0.05$ when comparing 4w and 8w in blank control groups, aP, MSaP, MSaP-aL and MSaP-aL/p; $@p<0.05$ when comparing sham group and other groups; $&p<0.05$ when comparing aP and microsol electrospun groups (MSaP, MSaP-aL and MSaP-aL/p), $*p<0.05$ when comparing MSaP-aL/p and other control groups via two-way analysis of variance (ANOVA) with a Tukey's post-hoc test, scale bars were $50\mu\text{m}$)

(2). SMI31 reacts with a phosphorylated epitope in extensively phosphorylated neurofilament H, also serves as an evaluation criteria in studies of degenerative diseases such as multiple sclerosis (MS)²⁵, Alzheimer's disease (AD)²⁶ and stroke²⁷. Despite that SMI31 was absent in current study, the NF-H protein in *in vitro* differentiated neuron-like axons was fluorescently labeled with its gene expression quantified by qRT-PCR in this study (Figure 6, in the red box below) which could also strengthen our result on neurogenic activity. Further study looking into the mechanism of biomimetic fiber

membrane on promoting nerve regeneration would employ SMI31 for better characterization.

Figure 6. Neuron-specific marker immunofluorescence staining and gene detection assays of scaffolds. (A) Immunofluorescence staining of Tuj-1, NSE, NF-200, Tau, staining images on different fiber scaffolds with (B), (C), (D), (E) Corresponding fluorescence semi-quantitative analysis. (F), (G), (H), (I) mRNA expression analysis of Tuj-1, NSE, NF-200, Tau. (n=3, all values were mean \pm std. dev., $p < 0.05$ when comparing microsol electrospun groups (MSaP, MSaP-aL, MSaP-aL/p) and control, aP group respectively via one-way analysis of variance (ANOVA) with a Tukey's post-hoc test., ns, not significant, scale bars were 100 μ m)

Line 601: I would add stainings against VWF to strengthen this result about neo-angiogenesis too.

Re : Thank you for the comment. We have added VWF labeled neovascularization research content *in vivo*, and made corresponding changes and highlighted in the revised manuscript.

Method: The primary antibody included rabbit anti-Tuj-1(Abcam, ab18207) for neuron staining, goat anti-nestin (R&D SYSTEMS, AF2736) for neural progenitor cells staining, mouse anti-GFAP (Servicebio, GB12096) for staining of astrocytes, mouse anti-NG2 (Abcam, ab50009) for staining of glial scars, mouse anti-CD31 (Servicebio, GB12063) for staining of vascular endothelial cells, rabbit anti-VWF (Abcam, ab6994) for staining of neovascularization. Second antibody includes goat anti-rabbit IgG (Cy3, Servicebio, GB21303), goat anti-mouse IgG (Alexa Fluor 488, Servicebio, GB25301) and goat anti-rabbit IgG (Alexa Fluor 647, Abcam, ab150079) (Supplementary information, page 17, line 428).

Results and discussion: Coincidentally, the quantitative analysis of neovascularization labeled by Von Willebrand Factor (VWF) fluorescence showed that the intensity in MSaP-aL/p group was significantly higher than that in other control groups at the same time ($p < 0.05$) (Fig. S13C, D). Similarly, the number of neovascularization labeled by VWF at 4 weeks was significantly less than that at 8 weeks ($p < 0.05$), which indicated that biomimetic bundle could not only reduce the invasion of local inflammatory reaction, but also induce macrophages and microglia cells to be polarized to M2 subtype, which could secrete vascular endothelial growth factor (VEGF) and promote blood vessel formation, providing a superior platform for tissue regeneration for a long time (page 23, line 638).

Figure S13C, D. (C), Immunofluorescence staining of neovascularization and (D) their quantitatively analysis with optical density. ($n=5$, all values were mean \pm std. dev., $p < 0.05$ when comparing MSaP-aL/p and other control groups at the same time and $\#p < 0.05$ when comparing 4w and 8w in MSaP-aL/p via two-way analysis of variance (ANOVA) with a Tukey's post-hoc test, scale bars were 50 μ m)

Figure 8: Behavioral recovery data are significant but of limited magnitude if compared to control groups.

Re : Thank you for the comment. The recovery of animal motor function is an important indicator for evaluating nerve repair. Sufficient sample size is the basis for ensuring the accuracy of the experimental results. However, I'm sorry that we had made some mistake on setting sample sizes for scoring, since limited sample size ($n=3$) led to the inaccuracy of the comparison. At present, we increased the sample size ($n=7$) for re-statistical analysis. corresponding corrections were made and provided as follow:

Method: The recovery of neuromotor function of hindlimb in six groups of rats was evaluated by BBB (Basso, Beattie, Bresnahan) and IPT (inclined plane test) scoring system. The score range of BBB scale was 0 to 21 with 0 indicating no hindlimb movement, and 21 indicating that motor function was normal. During each test, two unwitting examiners evaluated separately each week after the operation. As a supplementary study of BBB scoring test, IPT can improve the effectiveness and sensitivity of scoring. Using Rivlin

method, the rats were placed on the inclined plate with rubber pad, the longitudinal axis of the rat was kept parallel to the longitudinal axis of the inclined plate, the head was raised to one side, and raised 5° from 0° within 5 seconds. The highest angle of stay of the rats was recorded and the larger the angle indicated the stronger the load-bearing capacity of the lower extremities. The above two scoring methods are performed by two blinded observers and seven rats were randomly selected from each group at a fixed time point every week for evaluation (n=7) (Supplementary information, page 16, line 405).

Results and discussion: The *in vivo* performance of fiber bundle (Fig.7A (a, b, c, d)) was evaluated by SD rat T9 spinal cord hemisection model. The recovery of hindlimb motor dysfunction after spinal cord injury was evaluated by BBB and IPT scores every week after operation. The results showed that the hindlimb motor function of rats in each group recovered in varied degrees. Superior recovery of motor function could be identified in MSaP-aL/p group compared with other groups, as indicated by significantly higher BBB score and IPT score at each time points from 4th week post-injury ($p < 0.05$). At the 8th week, the BBB score of 13.71 ± 1.11 , and the IPT score of $55.86 \pm 4.67^\circ$ could be found in MSaP-aL/p group, both of which were significantly higher than those of the other groups ($p < 0.05$) (Fig. 7B, C). In addition, the minimum time required to achieve similar motor functions during recovery is more convincing in characterizing the functionality of neural repair materials. Therefore, in the comparison of 10 points (the highest average score of the negative control group in the BBB score) and 47° (the highest average angle of the negative control group in the IPT test), the two scores of the MSaP-aL/p group exceeded the control values at the 4th week after operation, reaching 11.14 ± 2.04 (BBB) and 52.43 ± 5.56 (IPT) respectively. The results of motor function recovery suggested that immunoregulatory fiber bundle (MSaP-aL/p) implantation could reduce the risk of further damage to surviving motor neurons by effectively inhibiting the acute inflammatory response of spinal cord injury, and promote nerve repair subsequently (page 18, line 482).

Figure 7B, C. (B) Evaluation of motor function recovery of lower limb by BBB score in rats assisted by (C) rats motor function IPT score. (n=7, all values were mean \pm std. dev., * $p < 0.05$ when comparing MSaP-aL/p and other groups via two-way analysis of variance (ANOVA) with a Tukey's post-hoc test)

Figure 11: I recommend to substitute “NSC” with neural progenitor cells

Re : Thank you for the excellent suggestion. We have changed the “NSC” involved in the manuscript to “neural progenitor cells”, and highlighted the changes in the manuscript.

Reviewer #3 (Remarks to the Author):

The manuscript by Xi et al. entitled " Microenvironment-Responsive Immunoregulatory Electrospun Fibers for Promoting Nerve Regeneration " presents results suggesting that treatment with a pH-responsive strategy based on the grafting of interleukin-4 (IL-4) plasmid-loaded and aldehydes-modified cationic liposome onto the nerve growth factor (NGF)-loaded electrospun fibrous scaffolds leads to immunomodulation and improves functional recovery and tissue repair after spinal cord injury.

Great thanks to the reviewer for your thoughtful reading of our manuscript. We have completely revised the original manuscript based on your constructive advices. Detailed point-by-point responses are provided below.

Points

1. The authors do not provide any clear evidence demonstrating that this strategy modulates inflammation. This is of key importance since their rationality to boost IL-4 levels after spinal cord injury was to modulate the inflammatory response. Were macrophage and microglia numbers changed by the treatment? Did the treatment shift microglia and macrophage polarization towards an M2-like phenotype *in vivo*?. These are key questions that the authors should have addressed. In additions, this must be assessed by using flow cytometry. This is the only methodology that allows to distinguish microglia from macrophages *in vivo*, since some new specific markers for microglia, such as TMEM119 or P2Y12, only discriminate microglia from macrophages in physiological but not in pathological conditions. The authors just showed that the treatment modulated cytokine levels in the spinal cord by using immunofluorescence. However, this is not a reliable technique to assess cytokine levels since these antibodies bind to various unspecific proteins. For these reasons, cytokine levels have to be assessed by ELISA (not WB) or Luminex assays. The authors need also to demonstrate that IL-4 protein levels is really increased after the treatment, since this therapeutic approach is based, in part, in increasing the levels of this immunomodulatory cytokine. Again, this needs to be done by ELISA or Luminex assays.

Re : Thank you for the thoughtful suggestion. According to your suggestion, we are deeply aware of the deficiency of immunomodulatory demonstration in *in vivo* research. In order to fix this significant problem, corresponding characterization were supplemented. Both flow cytometry and ELISA test were conducted 1w after implantation to evaluate the phenotypic changes of microglia and macrophages at the injured site *in vivo*. In addition,

the changes on the serum level inflammatory factors (IL-4, TNF- α , IL-1 β , IL-10, TGF- β) were also measured as described below to strengthen our results. Corresponding information was supplemented in manuscript and provided as follows:

(1). **Flow Cytometry method: 6.3.1 Identification of microglia/macrophages phenotype.** After euthanasia, the thoracic cavity was cut open with the left ventricle and inferior vena cava fully exposed. 100ml 0.9% saline was perfused into the heart. Spinal cord samples for flow cytometry detection following the previous article¹⁶. Briefly, a 5mm section of the injured spinal cord tissue centered at the epicenter of the injury site was harvested and was immediately cut into small fragments and mechanically separated by a 100 μ m cell strainer. The cell suspensions of different groups were centrifuged at 300g for 10 minutes at 4 $^{\circ}$ C, then the antibodies were added and incubated at 4 $^{\circ}$ C for 30 minutes and fixed in 1% paraformaldehyde. The following rat conjugated antibodies were used: mouse anti-CD11b-APC (BD PharmingenTM, 562102), mouse anti-CD86-FITC (BD PharmingenTM, 561961), mouse anti-CD206-PE (Santa Cruz, sc-58986). Cells were analyzed on a flow cytometer (Merk Millipore, USA) and the results were analyzed by FlowJo7.6 software. For analysis, cells were first gated for CD11b to ensure that only microglia/macrophages were selected, then the following combination of specific markers were used to identified M1 (CD86) and M2 (CD206). In the assay, three samples were randomly selected from each group for detection (n=3) (Supplementary information, page 14, line 350).

Results and discussion: Macrophages and microglia reached the recruitment peak within 7 days after spinal cord injury, so immunomodulatory bundles should play the role of biological response regulation during this period. On the seventh day after the implantation of biomimetic immunomodulatory fiber bundles, we first tested whether they could guide changes in local immune cell subtypes (Fig. 8A). There was no significant difference in the number of CD11b/CD86-positive macrophages between the non-immunomodulatory fiber bundle groups (ap, MSaP, MSaP-aL) and the blank control group ($p>0.05$) (Fig. 8B). However, due to the large porosity of oriented electrospinning fiber bundles, it could lead to the increase of CD11b/CD206-positive macrophages ($p<0.05$) (Fig. 8C)²⁸. In addition, the in vivo fiber bundles regulation of macrophage polarization was surprisingly consistent with the conclusion drawn in the in vitro study that MSaP-aL/p group not only significantly decreased the proportion of CD11b/CD86-positive macrophages, but also significantly increased CD11b/CD206-positive macrophages ($p<0.05$). The results of flow cytometry analysis showed that the implantation of immunomodulatory fiber bundles could promote the polarization of local microglia/macrophages to M2 phenotype in vivo (page 18, line 503).

Figure 8. Phenotypic identification of immune cells in vivo. (A), Flow cytometry analysis of CD11b/CD86-positive cells with (B) its quantification analyzed of the ratio and CD11b/CD206-positive cells with (C) its quantification analyzed of the ratio. (n=3, all values were mean \pm std. dev., #p<0.05 when comparing non-immunoregulatory fiber groups and blank control group and *p<0.05 when comparing MSaP-aL/p and other control groups via one-way analysis of variance (ANOVA) with a Tukey's post-hoc test., ns, not significant)

(2). **ELISA method:** The local severe inflammatory reaction of spinal cord injury produces a large number of immune factors into the systemic circulation with the blood. We used the way of clinical detection of inflammatory factors to initially evaluate the changes of inflammatory response after fiber bundles implantation. Briefly, three animals were randomly selected from each group, and the blood was collected with a centrifuge tube containing coagulant during cardiac perfusion (n=3). The blood was placed at room temperature for 30 minutes and then centrifuged at 3000rpm for 5 minutes. The upper serum was detected by ELISA kit (Multi Sciences, China) (Supplementary information, page 16, line 395).

Results and discussion: The systemic inflammatory response of animals was evaluated by ELISA detection of serum to comprehensively assess the immunoregulatory function of fiber bundles. Due to the large amount of IL-4 secreted into the systemic circulation at the injury site, the serum IL-4 content was significantly increased in the MSaP-aL/p group compared with other groups (p<0.05) (Fig. 9I). The results of pro-inflammatory and anti-inflammatory cytokines showed that, on the one hand, the serum inflammatory factors TNF- α and IL-1 β in the blank control, aP, MSaP, MSaP-aL groups were significantly higher than those in the MSaP-aL/p group (p <0.05) (Fig. 9J, K). On the other hand, MSaP-aL/p group had higher levels of IL-10 and TGF- β (p<0.05) (Fig. 9L, M), which highly suggested that the implantation of immune-regulating fiber bundles can not only improve the local immune environment, but also reduce systemic inflammation (page 20, line 543).

Serum samples after centrifugation and **Figure 9I-M**. Serum levels of IL-4 (I), TNF- α (J), IL-1 β (K), IL-10 (L), and TGF- β (M) were assessed by ELISA. ($n=3$, ELISA values were mean \pm std. dev., * $p<0.05$ when comparing MSaP-aL/p and other groups via one-way analysis of variance (ANOVA) with a Tukey's post-hoc test)

2. The title of the manuscript includes the words “Promoting Nerve Regeneration”. This is quite surprising since the author do not demonstrate that their experimental approach leads to axon regeneration. This is very important and needs to be assessed by the authors by means of tracer, viral plasmids or transgenic animals.

Re : Thank you for the comment. We hoped that by constructing a responsive composite fiber membrane, it could regulate the inflammation response of acute spinal cord injury and create a suitable microenvironment for nerve regeneration. Axon regeneration is an important criterion for evaluating the results of nerve repair, which we had been following during our research. Hence, for better assessment of axon regeneration, immunofluorescent labeling of GAP-43, a marker for axon sprouting, was conducted *in vivo*, with MSaP-aL/p showed the highest level of GAP-43 expression. Corresponding results were provided as follow (Figure 10C, D). Apart from that, preliminarily evaluation on the ability of composite biological scaffolds to promote nerve regeneration were conducted using four kinds of nerve specific markers to label newborn neurons *in vitro*. Among them, β -III tubulin (Tuj-1) is an important component of neuronal skeleton and one of the markers of immature neurons²⁹. Neuronal specific enolase (NSE) is a cytosolic protein consistently expressed by neurons and cells of neuronal origin which is a specific marker used to label neurons in tissue engineering³⁰. Neurofilament H (NF-200) provides structural support for axons and regulates axon diameter³¹. Tau protein is expressed at the distal portions of the axons and can be involved in regulating stability of the cytoskeleton and providing axonal flexibility³². It has been confirmed from the results that the NGF-loaded microsol electrospinning groups (MSaP, MSaP-aL, MSaP-aL/p) can promote the neural

differentiation of stem cells (Figure 6). *In vivo*, using nestin-labeled neural progenitor cells to assess immune regulation to inhibit glial scar formation and promote migration of neural progenitor cells (Figure S12). Tuj-1 was employed to label neuron cells, suggesting that the fibrous membrane had a sustained nerve regeneration function (Figure 10). Hence, we believed that the results *in vitro* and *in vivo* support the conclusion that biomimetic fiber membrane promotes nerve regeneration.

GAP-43 staining method: *As for the immunohistochemical evaluation, the antigen was repaired with 0.3% hydrogen peroxide and nonspecific antigen was blocked by 5% BSA. The primary antibody (rabbit anti GAP-43 (Abcam, ab75810) for axon staining) was added and incubated at 4 °C overnight, and the second antibody (goat anti-rabbit IgG (Alexa Fluor 647, Abcam, ab150079) was incubated at room temperature for 1 hour after PBS washing. The samples were observed and photographed under fluorescence microscope, with the fluorescence intensity analyzed with ImageJ software. The values were represented in the form of average optical intensity, which had been corrected for optical density, de-background, and finally normalized the staining intensity by the number of cell nuclei. A total of five images were randomly selected from each group for statistical analysis (n=5) (Supplementary information, page 17, line 426).*

Results and discussion: *Under normal physiological conditions, low expression of GAP-43 only maintains normal function of spinal cord, but GAP-43 is highly expressed in the process of nerve development and regeneration, which can regulate axonal extension, enhance nerve plasticity and release of neurotransmitters (Fig. 10C). Therefore, the fluorescence density of sham group was lower than that of other groups at the same time ($p<0.05$) (Fig. 10D). Meanwhile, the fluorescence density of GAP-43 in MSaP-aL/p group was significantly higher than that in other control groups, which proved that immunoregulation of local inflammatory response during acute spinal cord injury could promote nerve regeneration for a long time ($p<0.05$). In addition, aP group was lower than that in the microsol electrospun groups (MSaP, MSaP-aL, MSaP-aL/p) at the same time ($p<0.05$), suggesting that continuous supply of NGF could significantly promote nerve repair. Surprisingly, with the development of time, the expression of GAP-43 in all experimental groups except sham group at 4 weeks were significantly higher than those at 8 weeks ($p<0.05$). The results suggested that there is a limited time window for the repair of spinal cord injury. Immune regulation to reduce inflammatory injury in the stage of acute spinal cord injury can better promote nerve regeneration (page 22, line 613).*

Figure 10. Immunofluorescence staining of neuron cells and axons. (A), Immunofluorescence staining of neuron cells and (B) quantification analyzed with optical density. ($n=5$, Tuj-1 intensity values were mean \pm std. dev., $\%p < 0.05$ when comparing blank control and microsol electrospun groups (MSaP, MSaP-aL and MSaP-aL/p); $\&p < 0.05$ when comparing aP and microsol electrospun groups (MSaP, MSaP-aL and MSaP-aL/p); $\@p < 0.05$ when comparing 4w and 8w in MSaP; $\$p < 0.05$ when comparing 4w and 8w in MSaP-aL and $\#p < 0.05$ when comparing 4w and 8w in MSaP-aL/p; $*p < 0.05$ when comparing MSaP-aL/p and other control groups via two-way analysis of variance (ANOVA) with a Tukey's post-hoc test, scale bars are $50\mu\text{m}$) (C), Immunofluorescence staining of axon with (D) its quantification analyzed with optical density. ($n=5$, GAP-43 intensity values were mean \pm std. dev., $\#p < 0.05$ when comparing 4w and 8w in blank control groups, aP, MSaP, MSaP-aL and MSaP-aL/p; $\@p < 0.05$ when comparing sham group and other groups; $\&p < 0.05$ when comparing aP and microsol electrospun groups (MSaP, MSaP-aL and MSaP-aL/p), $*p < 0.05$ when comparing MSaP-aL/p and other control groups via two-way analysis of variance (ANOVA) with a Tukey's post-hoc test, scale bars were $50\mu\text{m}$)

***Figure 6.*** *Neuron-specific marker immunofluorescence staining and gene detection assays of scaffolds. (A) Immunofluorescence staining of Tuj-1, NSE, NF-200, Tau, staining images on different fiber scaffolds with (B), (C), (D), (E) Corresponding fluorescence semi-quantitative analysis. (F), (G), (H), (I) mRNA expression analysis of Tuj-1, NSE, NF-200, Tau. (n=3, all values were mean ± std. dev.,* $^{*}p < 0.05$ *when comparing microsol electrospun groups (MSaP, MSaP-aL, MSaP-aL/p) and control, aP group respectively via one-way analysis of variance (ANOVA) with a Tukey's post-hoc test., ns, not significant, scale bars were 100 μm)*

Figure S12. Immunofluorescence staining of neural stem cells and neuron cells. (A), Immunofluorescence staining of neural stem cells with (B) its quantification analyzed with optical density. (n=5, Nestin intensity values were mean \pm std. dev., [#]p<0.05 when comparing sham group and other groups and *p<0.05 when comparing MSaP-aL/p and other control groups via two-way analysis of variance (ANOVA) with a Tukey's post-hoc test, scale bars were 50 μ m)

3. The authors also applied NGF to stimulate mobilization and differentiation of stem cells into neurons. According to their data, the authors stated the this novel approach increased the stem cell counts and differentiation into neurons. However, this is based on immunohistochemistry against Nestin and Tuj1. These experiments are not appropriated to demonstrate or even to suggest their statements.

Re : Thank you for the comment. The misunderstanding about the mobilization of neural progenitor cells may be caused by the unclear description in the original manuscript. We had removed the content of endogenous stem cell mobilization and made corrections. Nestin fluorescence labeling test was used to evaluate whether immunoregulation during acute spinal cord injury could promote the appearance of neural progenitor cells at injured area in the later stage. The corresponding correction were made in the revised manuscript and provided as follow.

(1). **2.4 In vitro biological characteristics of nerve fiber membranes.** In this study, BMSCs and bone marrow macrophages (BMMs) were co-cultured through Transwell to simulate the effects of fiber membranes on cell-to-cell interaction in the microenvironment of acute spinal cord injury, and to evaluate the biological characteristics of nerve fiber membranes (page 13, line 337).

(2). **2.5.4 Histology and immunohistochemical evaluation of spinal cord repair in vivo.** Nestin, Tuj-1 and GAP-43 were used to label neural progenitor cells, neuron cells and axonal sprouting respectively to evaluate the ability of biomimetic bundles to promote nerve regeneration in the middle and late stage of spinal cord injury. The distribution of nestin-labeled neural progenitor cells at the injured site (Fig. S12A) was observed in each bundle group at 4 and 8 weeks after injury. The neural progenitor cells in the sham group was lower than the other groups, which may be due to the low proliferation activity of spinal ependymal cells and the lack of immune response to nestin protein (p <0.05), and the fluorescence quantitative display (Fig. S12B) at the same time point in the MSaP-aL/p group was significantly higher than that in the other control groups (p<0.05). The results showed that biomimetic bundle could significantly inhibit the blocking effect of surrounding scar tissue on neural progenitor cells through immunomodulatory effect. At 4w and 8w time points, neural progenitor cells were continuously promoted by NGF released from microsol electrospun fibers (MSaP, MSaP-aL and MSaP-aL/p), and a large number of Tuj-1-labeled neurons appeared as shown in Fig. 10A (page 21, line 587).

4. The authors reveal that this novel approach improved functional outcomes based on the BBB and IPT. It is quite surprising to observe statistical difference between groups with such small difference. It is also surprising to observe this tiny standard deviation after hemisection injury. Highlight that the statistics used are not appropriate since there are two factors (experimental group and time). Two-way RM ANOVA needs to be used.

Re : Thank you for the valuable comment. The recovery of animal motor function is an important indicator for evaluating nerve repair. We had realized the mistake on using an unacceptably small sample size (n=3) for scoring when the original manuscript was first submitted, leading to inaccuracy of the comparison. For more accurate assessment, we had repeated and increased the sample size (n=7) for re-statistical analysis. Corresponding corrections were provided as follow:

Method: *The recovery of neuromotor function of hindlimb in six groups of rats was evaluated by BBB (Basso, Beattie, Bresnahan) and IPT (inclined plane test) scoring system. The score range of BBB scale was 0 to 21 with 0 indicating no hindlimb movement, and 21 indicating that motor function was normal. During each test, two unwitting examiners evaluated separately each week after the operation. As a supplementary study of BBB scoring test, IPT can improve the effectiveness and sensitivity of scoring. Using Rivlin method, the rats were placed on the inclined plate with rubber pad, the longitudinal axis of the rat was kept parallel to the longitudinal axis of the inclined plate, the head was raised to one side, and raised 5° from 0° within 5 seconds. The highest angle of stay of the rats was recorded and the larger the angle indicated the stronger the load-bearing capacity of the lower extremities. The above two scoring methods are performed by two blinded observers and seven rats were randomly selected from each group at a fixed time point every week for evaluation (n=7) (Supplementary information, page 16, line 405).*

Results and discussion: *The in vivo performance of fiber bundle (Fig.7A (a, b, c, d) was evaluated by SD rat T9 spinal cord hemisection model. The recovery of hindlimb motor dysfunction after spinal cord injury was evaluated by BBB and IPT scores every week after operation. The results showed that the hindlimb motor function of rats in each group recovered in varied degrees. Superior recovery of motor function could be identified in MSaP-aL/p group compared with other groups, as indicated by significantly higher BBB score and IPT score at each time points from 4th week post-injury ($p < 0.05$). At the 8th week, the BBB score of 13.71 ± 1.11 , and the IPT score of $55.86 \pm 4.67^\circ$ could be found in MSaP-aL/p group, both of which were significantly higher than those of the other groups ($p < 0.05$) (Fig. 7B, C). In addition, the minimum time required to achieve similar motor functions during recovery is more convincing in characterizing the functionality of neural repair materials. Therefore, in the comparison of 10 points (the highest average score of the negative control group in the BBB score) and 47° (the highest average angle of the negative control group in the IPT test), the two scores of the MSaP-aL/p group exceeded the control values at the 4th week after operation, reaching 11.14 ± 2.04 (BBB) and 52.43 ± 5.56 (IPT) respectively. The results of motor function recovery suggested that immunoregulatory fiber bundle (MSaP-aL/p) implantation could reduce the risk of further*

damage to surviving motor neurons by effectively inhibiting the acute inflammatory response of spinal cord injury, and promote nerve repair subsequently (page 18, line 482).

Figure 7. Timeline of animal assay and motor function score. (A) Pictures of animal experiments. Image of (a) the responsive fiber bundle, (b) the fully exposed T9 spinal cord, (c) 3mm hemi-section made in the right site of the T9 spinal cord (the yellow arrow in the image indicated the injured site), (d) implanted fiber bundles (the yellow arrow in the image indicated the fiber bundle implantation position) along with (e), (f), (g), (h) post-operation condition of rat at 7 days, 4 weeks and 8 weeks. (B) Evaluation of motor function recovery of lower limb by BBB score in rats assisted by (C) rats motor function IPT score. (n=7, all values were mean \pm std. dev., * $p < 0.05$ when comparing MSaP-aL/p and other groups via two-way analysis of variance (ANOVA) with a Tukey's post-hoc test)

5. English should be revised throughout the text.

Re : Thank you for the comment. The whole manuscript has been polished thoroughly, partial grammar has been corrected. All the changes have been highlighted in the revised manuscript.

6. The n should be included in all the experiments

Re : Thank you for raising the key point. We included the sample size (n) used for statistical evaluation to all data descriptions, and highlighted in the revised manuscript.

Reference

1. Metz, G. A. et al., Validation of the weight-drop contusion model in rats: a comparative study of human spinal cord injury. *J. Neurotrauma*. **17**, 1-17 (2000).
2. Wang, X. J. et al., Polysialic-Acid-Based Micelles Promote Neural Regeneration in Spinal Cord Injury Therapy. *Nano Lett.* **19**, 829-838 (2019).
3. D'Amato, A. R. et al., Vastly extended drug release from poly(pro-17beta-estradiol) materials facilitates in vitro neurotrophism and neuroprotection. *Nat. Commun.* **10**, 4830 (2019).
4. Li, G. et al., Graft of the NT-3 persistent delivery gelatin sponge scaffold promotes axon regeneration, attenuates inflammation, and induces cell migration in rat and canine with spinal cord injury. *Biomaterials*. **83**, 233-248 (2016).
5. Milich, L. M. et al., The origin, fate, and contribution of macrophages to spinal cord injury pathology. *Acta Neuropathol.* **137**, 785-797 (2019).
6. Cerqueira, S. R. et al., Decellularized peripheral nerve supports Schwann cell transplants and axon growth following spinal cord injury. *Biomaterials*. **177**, 176-185 (2018).
7. Rejman, J. et al., Size-dependent internalization of particles via the pathways of clathrin- and caveolae-mediated endocytosis. *Biochem J.* **377**, 159-169 (2004).
8. Zhou, L. et al., Microsol-electrospinning for controlled loading and release of water-soluble drugs in microfibrinous membranes. *RSC Adv.* **4**, 43220-43226 (2014).
9. Cossich, E. et al., Development of electrospun photocatalytic TiO₂-polyamide-12 nanocomposites. *Biotechnol. Adv.* **31**, 421-437 (2015).
10. Zhang, Y. Z. et al., Coaxial Electrospinning of (Fluorescein Isothiocyanate-Conjugated Bovine Serum Albumin)-Encapsulated Poly(ϵ -caprolactone) Nanofibers for Sustained Release. *Biomacromolecules*. **7**, 1049-1057 (2006).
11. Shi, Q. et al., Collagen scaffolds modified with collagen-binding bFGF promotes the neural regeneration in a rat hemisectioned spinal cord injury model. *Sci China Life Sci.* **57**, 232-240 (2014).
12. Veisoh, O. et al., Domesticating the foreign body response: Recent advances and applications. *Adv Drug Deliv Rev.* **144**, 148-161 (2019).
13. Li, F. et al., CCR5 blockade promotes M2 macrophage activation and improves locomotor recovery after spinal cord injury in mice. *Inflammation*. **38**, 126-133 (2015).
14. Chen, C. et al., Bioinspired Hydrogel Electrospun Fibers for Spinal Cord Regeneration. *Adv. Funct. Mater.* **29**, 1806899 (2019).
15. Wu, L. et al., Hierarchical micro/nanofibrous membranes of sustained releasing VEGF for periosteal regeneration. *Biomaterials*. **227**, 119555 (2020).
16. Francos-Quijorna, I. et al., IL-4 drives microglia and macrophages toward a phenotype conducive for tissue repair and functional recovery after spinal cord injury. *Glia*. **64**, 2079-2092 (2016).

17. Li, H. et al., Isolation and characterization of primary bone marrow mesenchymal stromal cells. *Ann N Y Acad Sci.* **1370**, 109-118 (2016).
18. Pawar, K. et al., Biomaterial bridges enable regeneration and re-entry of corticospinal tract axons into the caudal spinal cord after SCI: Association with recovery of forelimb function. *Biomaterials.* **65**, 1-12 (2015).
19. Xu, D. et al., Efficient Delivery of Nerve Growth Factors to the Central Nervous System for Neural Regeneration. *Adv. Mater.* **31**, e1900727 (2019).
20. Bellver-Landete, V. et al., Microglia are an essential component of the neuroprotective scar that forms after spinal cord injury. *Nat. Commun.* **10**, 518 (2019).
21. He, S. et al., Multiple release of polyplexes of plasmids VEGF and bFGF from electrospun fibrous scaffolds towards regeneration of mature blood vessels. *Acta Biomater.* **8**, 2659-2669 (2012).
22. Monteiro, N. et al., Antibacterial activity of chitosan nanofiber meshes with liposomes immobilized releasing gentamicin. *Acta Biomater.* **18**, 196-205 (2015).
23. Monteiro, N. et al., Instructive nanofibrous scaffold comprising runt-related transcription factor 2 gene delivery for bone tissue engineering. *ACS Nano.* **8**, 8082-8094 (2014).
24. Storer, P. D. et al., β II-tubulin and GAP 43 mRNA expression in chronically injured neurons of the red nucleus after a second spinal cord injury. *Exp Neurol.* **183**, 537-547 (2003).
25. Berghoff, S. A. et al., Dietary cholesterol promotes repair of demyelinated lesions in the adult brain. *Nat. Commun.* **8**, 14241 (2017).
26. Li, T. et al., The neuritic plaque facilitates pathological conversion of tau in an Alzheimer's disease mouse model. *Nat. Commun.* **7**, 12082 (2016).
27. Cui, X. et al., D-4F Decreases White Matter Damage After Stroke in Mice. *Stroke.* **47**, 214-220 (2016).
28. Garg, K. et al., Macrophage functional polarization (M1/M2) in response to varying fiber and pore dimensions of electrospun scaffolds. *Biomaterials.* **34**, 4439-4451 (2013).
29. Lee, M. K. et al., The expression and posttranslational modification of a neuron-specific beta-tubulin isotype during chick embryogenesis. *Cell Motil Cytoskeleton.* **17**, 118-132 (1990).
30. Portiansky, E. L. et al., Increased number of neurons in the cervical spinal cord of aged female rats. *PLoS ONE.* **6**, e22537 (2011).
31. Portier, M. M. et al., Peripherin and neurofilaments: expression and role during neural development. *C R Acad Sci III.* **316**, 1124-1140 (1993).
32. Caprelli, M. T. et al., CNS Injury: Posttranslational Modification of the Tau Protein as a Biomarker. *Neuroscientist.* **25**, 8-21 (2019).

Reviewers' Comments:

Reviewer #3:

Remarks to the Author:

The authors have addressed my concerns

Reviewer #4:

Remarks to the Author:

The two key questions raised by Reviewer 1 was not addressed satisfactorily. These deal with whether there is in vivo release of the IL-4-plasmid and of NGF over the timespan of the experiment (4-8 weeks).

The authors responded by discussing the release of plasmid and NGF in their in vitro assays. They showed under these in vitro conditions the release of IL-4 plasmid and NGF under acidic conditions. But the in vivo release of NGF or plasmid in the injured spinal cord was not demonstrated.

On page 8 of the Response letter, they state that they were unable to detect eGFP staining of the IL-4 plasmid in formalin fixed paraffin sections either due to the embedding conditions or the need to use hydrogen peroxide. They could have used paraformaldehyde fixed frozen section stained for immunofluorescence to detect the plasmid. Although the indirect evidence showing increase in IL-4 expression by q-PCR and immunofluorescence is good, they should provide some direct evidence of the release of the plasmid in vivo.

They also do not have direct evidence of in vivo release of NGF but state that the sustained release of NGF could be verified indirectly through significantly promoted neural differentiation of endogenous progenitor cells. The latter may be due to the effects of NGF or some other effects mediated for example by the change in the levels of pro and anti-inflammatory cytokines. Moreover, the increase in Nestin+ cells does not mean that they differentiate into neurons. In fact, in vivo evidence indicates that in the adult spinal cord, progenitor cells differentiate mainly into astroglia. The increase in Tuj-1+ neurons in Fig. 10A could be likely due to reduction in secondary damage and sparing of intact tissue after SCI. The latter is likely to be mediated by the change in the pro and anti-inflammatory milieu. The role of inflammation on secondary damage after SCI is widely discussed in the literature. The possibility of secondary damage was not mentioned in the manuscript.

On page 7 of the Response letter, the authors state that "Neural regeneration was also fully characterized." This is an overstatement. Although the analysis done is good it is still very superficial. There is no assessment of axon growth or regeneration. This would need tracing or labeling of axons and viewed under high magnification as is done in the SCI field. High magnification examination of the implant area and the region of the interface is needed to clearly understand what effect the implant has with the host tissue. All the images shown are extremely small. The BBB analysis of locomotor recovery was developed for spinal cord contusion injury and is not ideal for hemisection injury. Other tests such as the catwalk would be better.

Reviewers' comments:

The black text was copied from the editors' letter, and the blue indentation is our response.

Reviewer #3 (Remarks to the Author):

The authors have addressed my concerns .

Thank you for your consideration of this work. We appreciate your effort in helping us improve our manuscript!

Reviewer #4 (Remarks to the Author):

Thank you for your constructive comments. We have carefully revised the manuscript according to your comments and supplemented some new results to ensure the integrity of the research conclusions. Detailed point-by-point responses are provided below.

The two key questions raised by Reviewer 1 was not addressed satisfactorily. These deal with whether there is *in vivo* release of the IL-4-plasmid and of NGF over the timespan of the experiment (4-8 weeks).

The authors responded by discussing the release of plasmid and NGF in their *in vitro* assays. They showed under these *in vitro* conditions the release of IL-4 plasmid and NGF under acidic conditions. But the *in vivo* release of NGF or plasmid in the injured spinal cord was not demonstrated.

On page 8 of the Response letter, they state that they were unable to detect eGFP staining of the IL-4 plasmid in formalin fixed paraffin sections either due to the embedding conditions or the need to use hydrogen peroxide. They could have used paraformaldehyde fixed frozen section stained for immunofluorescence to detect the plasmid. Although the indirect evidence showing increase in IL-4 expression by q-PCR and immunofluorescence is good, they should provide some direct evidence of the release of the plasmid *in vivo*.

Re : Thank you for the comment and suggestion. We should indeed provide some direct evidence on the release of the plasmid *in vivo*. As shown in Fig. 9 below, we had indirectly demonstrated that the aldoxylated cationic liposomes loading IL-4 plasmid regulated subtypes of local microglia and infiltrating macrophage to improve the microenvironment of nerve regeneration after implantation at injured site with the help of qRT-PCR, immunofluorescence and serum ELISA detection. In order to more intuitively characterize the effect of responsive plasmid transfection, rat spinal cord samples were collected after operation, and the tissue transfected with IL-4-eGFP plasmid was labeled *in vivo* by frozen section. At the same time, M2 macrophages polarized by IL-4 plasmid were labeled with

CD206 antibody. Corresponding supplemented contents are shown below and highlighted in the revised manuscript.

(1)

Figure 9. Evaluation of immune regulation 7 days after operation. (A) IL-4 immunofluorescence single labeling and TNF- α , IL-10 immunofluorescence double labeling. Quantitatively analysis of (B) IL-4 immunofluorescence, (C) IL-10 immunofluorescence and TNF- α immunofluorescence. (n=5, all values were mean \pm std. dev., * $p < 0.05$ when comparing MSaP-aL/p and other gmls via two-way analysis of

variance (ANOVA) with a Tukey's post-hoc test, scale bars are 50 μ m) Spinal cord specimens of IL-4 (D), TNF- α (E), IL-1 β (F), IL-10 (G) and TGF- β (H) gene expressions were assessed by qRT-PCR. Serum levels of IL-4 (I), TNF- α (J), IL-1 β (K), IL-10 (L), and TGF- β (M) were assessed by ELISA. (n=3, qRT-PCR and ELISA values were mean \pm std. dev., * $p < 0.05$ when comparing MSaP-aL/p and other groups via one-way analysis of variance (ANOVA) with a Tukey's post-hoc test)

(2) Method: 6.3.5 Evaluation of IL-4 plasmid transfection and macrophage phenotype. Spinal cord specimens were collected 7 days after operation for frozen sections following procedure reported previously (MSaP-aL/p group was set as the experimental group, the unloaded plasmid group (MSaP-aL) and the eGFP plasmid loaded group (MSaP-aL/g) were used as the control group)¹. Briefly, after intraperitoneal anesthesia, 4% paraformaldehyde and saline were perfused into the left ventricle of rat respectively, and the spinal cord specimens were quickly collected and immersed in 4% paraformaldehyde overnight. After subsequently immersed in 30% sucrose cryoprotectant overnight, the spinal cord samples were frozen for section with a longitudinal thickness of 15 μ m for following immunofluorescent staining with CD206 antibody (Abcam, ab195192) coupled with co-staining of nucleus with DAPI. (page 40, line 1084)

Results and discussion: In addition, as shown in Fig.S10A. Strong green fluorescence of eGFP could be observed in spinal cord tissue of MSaP-aL/g and MSaP-aL/p groups, showing no significant difference in semi-quantitative between these two groups ($p > 0.05$). On the other hand, no green fluorescence could be detected in MSaP-aL group compared with other two groups ($p < 0.05$) (Fig.S10B). Meanwhile, M2 macrophages labeled against CD206 with red fluorescence were found in all three groups to distribute around the cells. Corresponding semi-quantitative analysis showed significantly lower CD206 levels in MSaP-aL and MSaP-aL/g groups without IL-4 plasmid when compared with plasmid-containing MSaP-aL/p group ($p < 0.05$) (Fig.S10C). Taking results of IL-4 immunofluorescent staining into consideration, which showed significantly higher IL-4 expression in MSaP-aL/p group (Fig 9.A, B), it could be confirmed that the cationic liposomes loading with IL-4 plasmid have been responsively transfected into the surrounding tissue to regulate the polarization of macrophages. (page 20, line 542)

Figure S10. IL-4-eGFP transfection (green) and staining of M2 macrophages (red) of spinal cord tissues in MSaP-aL, MSaP-aL/g and MSaP-aL/p at Day 7. (A) EGFP expression and CD206 immunofluorescence staining in each group, scale bars are 50µm. (B) Semi-quantitatively analysis of eGFP. (n=5, all values were mean ± std. dev., *p < 0.05 when comparing MSaP-aL and other groups via one-way analysis of variance (ANOVA) with a Tukey's post-hoc test., ns, not significant) (C) Semi-quantitatively analysis of CD206. (n=5, all values were mean ± std. dev., *p < 0.05 when comparing MSaP-aL/p and other groups via one-way analysis of variance (ANOVA) with a Tukey's post-hoc test., ns, not significant)

They also do not have direct evidence of in vivo release of NGF but state that the sustained release of NGF could be verified indirectly through significantly promoted neural differentiation of endogenous progenitor cells. The latter may be due to the effects of NGF or some other effects mediated for example by the change in the levels of pro and anti-inflammatory cytokines. Moreover, the increase in Nestin+ cells does not mean that they differentiate into neurons. In fact, in vivo evidence indicates that in the adult spinal cord, progenitor cells differentiate mainly into astroglia. The increase in Tuj-1+ neurons in Fig. 10A could be likely due to reduction in secondary damage and sparing of intact tissue after SCI. The latter is likely to be mediated by the change in

the pro and anti-inflammatory milieu. The role of inflammation on secondary damage after SCI is widely discussed in the literature. The possibility of secondary damage was not mentioned in the manuscript.

Re : Thank you for the comment. For the comment concerning NGF release, direct evidence on NGF sustained release should indeed be provided to consolidate our conclusion. Hence, we have supplemented the experiment on the *in vivo* release of NGF and made the following highlighted modifications in the revised manuscript.

(1) Quantification of NGF concentration in spinal cord tissue by ELISA.

Method : 6.3.6 Pharmacokinetics of NGF in spinal cord. *Spinal cord specimens centered on the injury site were collected on 1, 6, 11, 16, 21 days after operation. After weighing, specimens were immersed in PBS buffer containing 1% PMSF with 5 times mass volume of sample, and cut into small pieces. The tissue homogenate was prepared and centrifuged at 12000g for 10 minutes to obtain supernatant. The NGF level in spinal cord was then determined by ELISA (Solarbio Life Sciences, SEKR-0015) with a standard curve established as manuscript's instruction. (page 40, line 1096)*

Results and discussion : *The content of NGF in spinal cord tissue of each group was detected at the predetermined time point after operation (Fig. S11). NGF in spinal cord injury group (blank control, aP, MSaP, MSaP-aL, MSaP-aL/p) increased in varying degrees, indicating certain degrees of self-repair activities after spinal cord injury. Meanwhile, the concentration of NGF in MSaP, MSaP-aL and MSaP-aL/p groups were significantly higher than that in aP and blank control groups on the first day after operation indicating that loaded cytokines have been partially released from microsol fiber groups ($p < 0.05$). At each time point after day 6, the content of NGF in each group generally showed a downward trend, but the concentration of NGF in MSaP-aL/p group was significantly higher than that in other groups, and on the 21st day ($p < 0.05$), MSaP-aL/p group exhibited a concentration twice as much as that in other microsol fiber groups, indicating the process of endogenous NGF consumption in local tissue to maintain the homeostasis of surviving nerve tissue after spinal cord injury. Such difference could further indicate that the regulation of inflammatory response in acute spinal cord injury is beneficial to inhibit local tissue necroptosis², reduce the leakage of cell content enzymes to interfere and decompose endogenous and exogenous NGF³. (page 21, line 572)*

Figure S11. Pharmacokinetics of NGF in spinal cord after operation ($n=3$, all values were mean \pm std. dev., # $p < 0.05$ when comparing microsolv fiber groups and other groups, * $p < 0.05$ when comparing MSaP-aL/p and other groups via one-way analysis of variance (ANOVA) with a Tukey's post-hoc test)

(2) We totally agree with the reviewers that a large number of *in vivo* studies have shown that neural progenitor cells mainly differentiate into astrocytes. We used Nestin fluorescence staining in current study to evaluate the immune regulation of composite fiber scaffolds in response to local pathological environment in acute spinal cord injury. The scaffold was found to effectively alleviate acute inflammation and subsequently reduce glial scar formation (as shown in Fig. S13 below), reduce the blocking effect of physical barrier on the migration of endogenous stem cells with neurogenic potential to the injured site. In addition, the sustained release of NGF from the composite fiber scaffold can enhance neuronal regeneration and reduce the area of tissue cavity⁴. The process of nerve tissue repair is maintained cooperatively by the biological effect of composite materials.

Figure S13. Immunofluorescence staining of activated astrocytes and glial scar. (A), (C) Immunofluorescence staining of astrocytes and quantitatively analyzed with optical density along with (B), (D) their quantification with optical density. ($n=5$, all values were mean \pm std. dev., $*p < 0.05$ when comparing MSaP-aL/p and other control groups, $\#p < 0.05$ when comparing material transplantation groups and blank control group via two-way analysis of variance (ANOVA) with a Tukey's post-hoc test, scale bars were $50\mu\text{m}$)

(3) The occurrence of primary spinal cord injury leads to the destruction of local blood vessels, resulting in hypoxia, oxidative stress and increased vascular permeability. Inflammatory cells such as neutrophils and macrophages reach the injury site through the blood-spinal cord barrier, and in situ microglia secrete inflammatory factors such as IL-1 β , IL-6 and TNF- α , resulting in necrosis or/and apoptosis of surviving neurons and glial cells, which would enhance the activation of astrocytes and promote the excessive deposition of chondroitin sulfate proteoglycans to form dense glial scar tissue that hinders axonal regeneration and extension. The above series of biochemical cascade reactions progressed

to secondary spinal cord injury, which seriously affected the prognosis⁵. Our previous studies showed that regulating local macrophage subtypes in acute spinal cord injury could significantly inhibit the apoptosis of nearby surviving neurons². IL-4 can bind to IL-4Ra on the surface of macrophages and microglia to activate STAT6 to promote tissue repair and reduce immune inflammatory response⁶, thus protecting the central nervous system⁷. Hence, the responsive composite fiber scaffold to promote the release of IL-4 cytokines in the acute phase of spinal cord injury is beneficial to reduce the degree and duration of inflammatory reaction. To sum up, in the study, through the evaluation of animal motor function, the composite material inhibited inflammation and neuroprotective function by IL-4 in the acute phase, while staining of GFAP and NG2 labeled reactive astrocytes and glial scar tissue respectively to evaluate the degree of secondary spinal cord injury. We have added and highlighted the regulation of the effect of acute inflammation on secondary spinal cord injury in the manuscript to improve the integrity of the discussion.

a. In addition, acute inflammatory storm causes glial cells to deposit chondroitin sulfate proteoglycan (CSPGs)⁸, resulting in the formation of glial scar tissue in the process of secondary spinal cord injury to hinder nerve repair. In order to further study the effect of biomimetic immunoregulatory bundles on anti-scar formation, we employed antibody against GFAP and NG2 to label the activated astrocytes and glial scar respectively, as shown in Fig. S13A, C. Both the activated astrocytes and glial scar tissues of aP, MSaP, MSaP-aL and MSaP-aL/p bundles at 4 and 8 weeks were significantly less than those in the blank control group ($p < 0.05$), it was inferred that the oriented electrospun fibers structure simulated the morphological distribution of nerve tissue, which could reduce the formation of central traumatic neuroma after nerve injury. In addition, the fluorescence of semi-quantitative analysis showed (Fig. S13B, D), the MSaP-aL/p bundle produced significantly lower signal intensity than blank control group and three other negative control groups ($p < 0.05$). It was confirmed that the biomimetic bundle has a superior ability to regulate the local immune microenvironment during acute spinal cord injury and further effect on reducing the formation of scar tissue in the stage of secondary spinal cord injury, paving the way for endogenous neural progenitor cells to migrate to the injured site and differentiate into nerve tissue. (page 22, line 603)

b. After spinal cord injury, endogenous stem cells have the potential to migrate to the injured site and differentiate into nerves. However, due to the occurrence of secondary spinal cord injury, the formation of scar tissue hinders cell migration. The distribution of nestin-labeled stem cells at the injured site (Fig. S14A) was observed in each bundle group at 4 and 8 weeks after injury. The neural progenitor cells in the sham group was lower than the other groups, which may be attributed to the low proliferation activity of spinal ependymal cells and the lack of immune response to nestin protein ($p < 0.05$). The corresponding fluorescence quantitative study displayed that at the same time point (Fig. S14B), MSaP-aL/p group could bring significantly more nestin-labeled stem cells than other control groups ($p < 0.05$). The results indicated that biomimetic bundle could significantly inhibit the blocking effect of surrounding scar tissue on endogenous stem cells through immunomodulatory effect in acute phase. (page 22, line 621)

c. The results of motor function recovery suggested that immunoregulatory fiber bundle

(MSaP-aL/p) implantation could reduce the risk of further damage to surviving motor neurons by effectively inhibiting the acute inflammatory response of spinal cord injury, and promoted nerve repair in the subsequent stage of secondary spinal cord injury. (page 18, line 496)

On page 7 of the Response letter, the authors state that “Neural regeneration was also fully characterized.” This is an overstatement. Although the analysis done is good it is still very superficial. There is no assessment of axon growth or regeneration. This would need tracing or labeling of axons and viewed under high magnification as is done in the SCI field. High magnification examination of the implant area and the region of the interface is needed to clearly understand what effect the implant has with the host tissue. All the images shown are extremely small. The BBB analysis of locomotor recovery was developed for spinal cord contusion injury and is not ideal for hemisection injury. Other tests such as the catwalk would be better.

Re : Thank you for pointing out our shortcomings and suggestions for improvement.

(1) Nerve repair after spinal cord injury is a complex and extremely extensive process. The responsive immunoregulatory electrospun fibers constructed in current study involves a representative part concerning the regeneration of spinal cord tissue. We sincerely apologize for the previous exaggerated description.

(2) Axon regeneration and growth are important evaluation indexes of nerve repair, so NF-200 and Tau protein antibodies were used to fluorescently label the axons and detect related gene expression in co-culture systems (as shown in Fig. 6 below). *In vivo* repair is not only a more diversified microenvironment, but also a more stringent test for the function of nerve-promoting repair biomaterials. Therefore, in the process of revising the manuscript, we adopted the previous reviewer's suggestion that GAP-43 should be introduced to fluorescently label the axon sprouts in tissue (as shown in Fig. 10C, D below). The above *in vitro* and *in vivo* results indicated that immunoregulatory electrospun fibers have the function of inhibiting the inflammatory response in acute spinal cord injury and promoting nerve regeneration.

Figure 6. Neuron-specific marker immunofluorescence staining and gene detection assays of scaffolds. (A) Immunofluorescence staining of Tuj-1, NSE, NF-200, Tau, staining images on different fiber scaffolds with (B), (C), (D), (E) Corresponding fluorescence semi-quantitative analysis. (F), (G), (H), (I) mRNA expression analysis of Tuj-1, NSE, NF-200, Tau. ($n=3$, all values were mean \pm std. dev., $*p < 0.05$ when comparing microsol electrospun groups (MSaP, MSaP-aL, MSaP-aL/p) and control, aP group respectively via one-way analysis of variance (ANOVA) with a Tukey's post-hoc test., ns, not significant, scale bars were $100\mu\text{m}$)

Figure 10C, D. (C), Immunofluorescence staining of axon with (D) its quantification analyzed with optical density. ($n=5$, GAP-43 intensity values were mean \pm std. dev., $^{\#}p<0.05$ when comparing 4w and 8w in blank control groups, aP, MSaP, MSaP-aL and MSaP-aL/p; $^{\textcircled{a}}p<0.05$ when comparing sham group and other groups; $^{\textcircled{b}}p<0.05$ when comparing aP and microsol electrospun groups (MSaP, MSaP-aL and MSaP-aL/p), $^*p<0.05$ when comparing MSaP-aL/p and other control groups via two-way analysis of variance (ANOVA) with a Tukey's post-hoc test, scale bars were $50\mu\text{m}$)

(3) The improvement of animal motor function can directly reflect the recovery of spinal cord nerve function, thus in the study of motor function, with reference to the methods reported in the literature, we found that BBB score could evaluate motor function^{9,10}. However, considering that the single research method may have some defects such as one-sidedness, we used BBB score method combined with inclined plate test (IPT) to

preliminarily evaluate the effect of immunoregulatory fiber scaffolds on promoting nerve function recovery (Fig. 7B, C). Thanks to the reviewers for the suggestions on conducting more comprehensive evaluation of the motor function of spinal cord injury, we've planned to conduct more systematic evaluation on motor function with employment of high-level experimental animals in the future studies for the exploration of better biomaterial design and underlying mechanism.

Figure 7B, C. (B) Evaluation of motor function recovery of lower limb by BBB score in rats assisted by (C) rats motor function IPT score. ($n=7$, all values were mean \pm std. dev., $*p < 0.05$ when comparing MSaP-aL/p and other groups via two-way analysis of variance (ANOVA) with a Tukey's post-hoc test)

Reference

1. McMahon, S. S. et al., Engraftment, migration and differentiation of neural stem cells in the rat spinal cord following contusion injury. *Cytotherapy*. **12**, 313-325 (2010).
2. Peng, Z. et al., Promotion of neurological recovery in rat spinal cord injury by mesenchymal stem cells loaded on nerve-guided collagen scaffold through increasing alternatively activated macrophage polarization. *J Tissue Eng Regen Med*. **12**, e1725-e1736 (2018).
3. Tanzer, M. C. et al., Quantitative and Dynamic Catalogs of Proteins Released during Apoptotic and Necroptotic Cell Death. *Cell Rep*. **30**, 1260-1270.e5 (2020).
4. Lee, K. et al., Growth factor delivery-based tissue engineering: general approaches and a review of recent developments. *J R Soc Interface*. **8**, 153-170 (2011).
5. Hulsebosch, C. E. Recent advances in pathophysiology and treatment of spinal cord injury. *Adv Physiol Educ*. **26**, 1-4 (2002).
6. Spiller, K. L. et al., Macrophage-based therapeutic strategies in regenerative medicine. *Adv Drug Deliv Rev*. **122**, 74-83 (2017).
7. Dooley, D. et al., Immunopharmacological intervention for successful neural stem cell

therapy: New perspectives in CNS neurogenesis and repair. *Pharmacol Ther.* **141**, 21-31 (2014).

8. Properzi, F. et al., Chondroitin sulphate proteoglycans in the central nervous system: changes and synthesis after injury. *Biochem Soc Trans.* **31**, 335-336 (2003).

9. Liu, D. et al., Biodegradable Spheres Protect Traumatically Injured Spinal Cord by Alleviating the Glutamate-Induced Excitotoxicity. *Adv Mater.* **30**, e1706032 (2018).

10. Shi, Y. et al., Effective repair of traumatically injured spinal cord by nanoscale block copolymer micelles. *Nat Nanotechnol.* **5**, 80-87 (2010).

Reviewers' Comments:

Reviewer #5:

None